# IN-CONTEXT TIME SERIES PREDICTOR

**Jiecheng Lu, Yan Sun, Shihao Yang**
Georgia Institute of Technology
{jlu414,yansun}@gatech.edu, shihao.yang@isye.gatech.edu

## ABSTRACT

Recent Transformer-based large language models (LLMs) demonstrate in-context learning ability to perform various functions based solely on the provided context, without updating model parameters. To fully utilize the in-context capabilities in time series forecasting (TSF) problems, unlike previous Transformer-based or LLM-based time series forecasting methods, we reformulate "time series forecasting tasks" as input tokens by constructing a series of (lookback, future) pairs within the tokens. This method aligns more closely with the inherent in-context mechanisms, and is more parameter-efficient without the need of using pre-trained LLM parameters. Furthermore, it addresses issues such as overfitting in existing Transformer-based TSF models, consistently achieving better performance across full-data, few-shot, and zero-shot settings compared to previous architectures [1].

## 1 INTRODUCTION

Transformer-based large language models (LLMs) have significantly impacted various research and application areas (Brown et al., 2020). Their inherent in-context learning (ICL) capabilities highlighted by previous studies (Müller et al., 2022; Min et al., 2022; Wei et al., 2023; Xie et al., 2022) allow them to adapt and generalize from context examples provided in input prompts without any parameter updates. This enables LLMs to effectively handle few-shot and zero-shot tasks. Recent research on ICL (Zhang et al., 2023; Garg et al., 2022; Akyürek et al., 2023; Li et al., 2023; Dai et al., 2023; Bai et al., 2023) has shown that Transformers can adaptively learn to perform various functions including linear predictors, shallow MLPs, gradient descent, algorithm selection, etc., based on a series of (input, label) pairs as input tokens, as shown in Figure 1 (a).

Time series forecasting (TSF) is critical in fields like epidemiology, finance, and traffic (Kaastra & Boyd, 1996; Lana et al., 2018; Yang et al., 2015; Ma et al., 2022), predicting future values from historical data. Temporal-wise Transformers, which build input tokens from series values at each timestep, have been widely researched in TSF (Li et al., 2019; Zhou et al., 2021; Wu et al., 2021; Nie et al., 2022). Some studies have

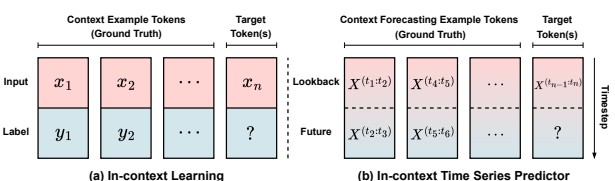

Figure 1: Overview of in-context TSF learning in our setup.

identified several issues with such Transformers, like timestep mixing and permutation invariance (Zeng et al., 2023), leading to overfitting and underperformance on real-world datasets compared to simpler models like linear predictors. Previously proposed solutions include channel independence (Nie et al., 2022), random channel dropout (Lu et al., 2024), and using Series-wise Transformers (Liu et al., 2024a; Wang et al., 2024), which consider each series as a token. Yet, their underlying mechanisms are not well explained. Additionally, the fixed structure of input series of these existing Transformers restricts their adaptability in few-shot and zero-shot learning for multivariate TSF.

Moreover, recent research have expanded the application of Transformer-based LLMs to TSF (Zhang et al., 2024), achieving improvements in few-shot and zero-shot generalization. They use the methods like prompt engineering (Gruver et al., 2023), fine-tuning and embedding inversion (Zhou et al., 2023;

---
[1]Code implementation is available at: https://anonymous.4open.science/r/ICTSP-C995

Jin et al., 2024) to integrate time series context into prompts, improving forecasting accuracy for new data. However, these methods are designed to adapt LLMs to TSF tasks rather than directly addressing the core aspects of TSF problems. This adaptation leads to inefficient use of LLM parameters and substantially increased computational costs.

In this study, we apply the most fundamental ICL settings in TSF problems to construct the In-context Time Series Predictor (ICTSP). This structure allows us to leverage the in-context generalization capabilities of Transformers efficiently without relying on large-scale LLM parameters. In ICTSP, we generate a sequence of (lookback, future) pairs, representing "time series forecasting tasks," from the original time series data, which are used as input tokens for the Transformer, as shown 1 (b). This setup enables the Transformer to adaptively learn the most effective predictor for the target tasks based on the ground truth forecasting examples as context. Additionally, the ICTSP effectively resolves aforementioned longstanding issues in previous TSF Trasnformers, significantly enhancing the performance in few-shot and zero-shot learning scenarios in multivariate TSF settings.

The main contributions of this paper are summarized as follows:

a) We innovatively use forecasting tasks — instead of traditional timestep values or single series — as input tokens to construct the ICTSP structure. By utilizing ground truth (lookback, future) pairs as context examples, we fundamentally and efficiently leverage ICL abilities for TSF tasks. ICTSP outperforms previous methods across full-data, few-shot, and zero-shot scenarios, positioning it as a potential solution for building universal large TSF models.

b) From an ICL perspective, we explain that issues in existing TSF Transformers, like timestep mixing, permutation invariance, and channel structure restriction, are caused by inappropriate token formulation. We show how previous solutions have partially addressed these issues and how ICTSP effectively solves them without the drawbacks of previous approaches.

c) We show that the ICTSP structure encompasses several simpler models as special cases, allowing for a sequential adaptive reduction in complexity: i) predictors learned from context examples through ICL, ii) a series-wise Transformer without context examples (Liu et al., 2024a), and iii) univariate MLPs or linear predictors (Zeng et al., 2023). This connection ensures stable performance across different complexities of time series dataset and prevents the significant overfitting that has previously hindered Transformers from outperforming simpler models consistently in real-world datasets.

## 2 METHOD

### 2.1 PRELIMINARIES

**Time Series Forecasting**  Let $X \in \mathbb{R}^{C \times L}$ represent a multivariate time series, where $L$ is the total length and $C$ is the number of channels of input series. $X$ is split into historical input $X_I \in \mathbb{R}^{C \times L_I}$ and future data $X_P \in \mathbb{R}^{C \times L_P}$, with $L = L_I + L_P$. Here, $L_I$ and $L_P$ represent the lengths of the historical and forecasting periods, respectively. The value at the $t$-th timestep in the $j$-th channel is $X_j^{(t)}$, where $t \in \{1, \ldots, L\}$ and $j \in \{1, \ldots, C\}$. The objective is to develop the best predictor $f : \mathbb{R}^{C \times L_I} \to \mathbb{R}^{C \times L_P}$ that maps historical inputs to future outputs, yielding $\widehat{X}_P = f(X_I)$.

**Transformer Architecture**  In this study, we employ a Transformer architecture with pre-normalization settings, processing $D$-dimensional input tokens $\{\mathbf{z}_i\}_{i=1}^{N}$ from the input matrix $\mathbf{Z}^{D \times N} = [\mathbf{z}_1, \ldots, \mathbf{z}_N] \in \mathbb{R}^{D \times N}$. Each token $\mathbf{z}_i$ passes through $K$ layers of the Transformer. Each layer begins with layer normalization $\mathrm{LN}(\cdot)$, followed by self-attention $\mathrm{Attn}(\cdot)$ and a feed-forward network $\mathrm{FFN}(\cdot)$. The output $\mathbf{Z}_k$ for each Transformer layer, $\mathrm{TF}_k$, is computed as:

$$\mathbf{Z}_k = \mathrm{TF}_k(\mathbf{Z}_{k-1}) = \mathbf{Z}_{k-1} + \mathrm{Attn}_k\left(\mathrm{LN}(\mathbf{Z}_{k-1})\right) + \mathrm{LN}\left(\mathrm{FFN}_k\left(\mathbf{Z}_{k-1} + \mathrm{Attn}_k\left(\mathrm{LN}(\mathbf{Z}_{k-1})\right)\right)\right), \quad (1)$$

with $\mathbf{Z}_0 = \mathbf{Z}$ and the first addition of $\mathbf{Z}_{k-1}$ providing a residual shortcut directly from input to output. The Transformer's final output is $\mathrm{TF}(\mathbf{Z}_0) = \mathbf{Z}_K$.

**In-context Learning**  ICL involves each datapoint $(\mathbf{x}_i, \mathbf{y}_i) \in \mathbb{R}^a \times \mathbb{R}^b$ from a dataset $\{(\mathbf{x}_i, \mathbf{y}_i)\}_{i=1}^{N}$. Unlike traditional supervised learning that learns direct mappings from $\mathbf{x}_i$ to $\mathbf{y}_i$, ICL predicts $\mathbf{y}_i$ using both historical observations $\{(\mathbf{x}_j, \mathbf{y}_j)\}_{j<i}$ and the current input $\mathbf{x}_i$. The input format for the

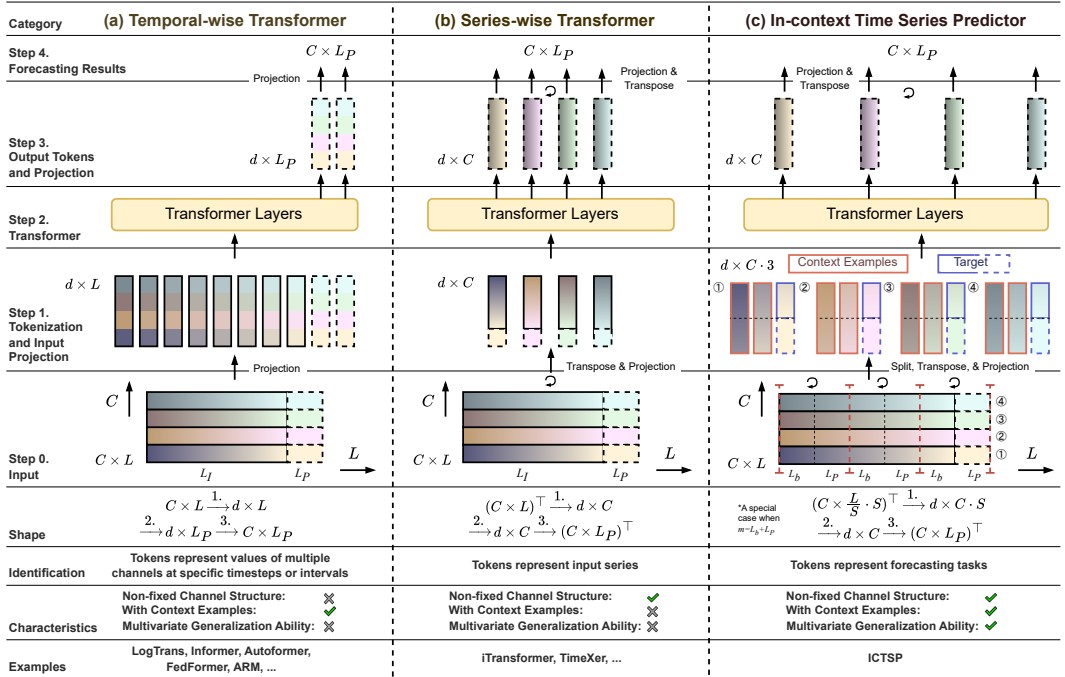

Figure 2: Architecture and characteristic comparison among the three main TSF Transformer structures. Please note that, for simple illustration, the ICTSP part present a special case, where the sampling steps equal to $L_b + L_P$, creating non-overlapping context forecasting examples.

$i$-th datapoint in the ICL settings is structured as:

$$\mathbf{Z}^{(a+b)\times i} = [\mathbf{z}_1, \ldots, \mathbf{z}_i] = \begin{bmatrix} \mathbf{p}_1 & \mathbf{p}_2 & \cdots & \mathbf{p}_{i-1} & \mathbf{p}_i \\ \mathbf{x}_1 & \mathbf{x}_2 & \cdots & \mathbf{x}_{i-1} & \mathbf{x}_i \\ \mathbf{y}_1 & \mathbf{y}_2 & \cdots & \mathbf{y}_{i-1} & \mathbf{o} \end{bmatrix}, \tag{2}$$

where $\mathbf{p}_.$ are positional embeddings and $\mathbf{o}$ is a placeholder embedding for the target, typically zero-filled. Note that in some ICL settings, $\mathbf{x}_.$ and $\mathbf{y}_.$ may be divided into separate tokens, which is proved interchangeable with our expression above (Bai et al., 2023; Guo et al., 2024). The positional embeddings $\mathbf{p}_.$ are added to the tokens and shown concatenated here just for illustration. Finally, the predicted output for the $i$-th data point, $\hat{\mathbf{y}}_i$, can be extracted from the corresponding token in the transformer's output $\hat{\mathbf{Z}} = \text{TF}(\mathbf{Z})$. For simplicity, we have omitted the input and output linear projections for matrices $\mathbf{Z}$ and $\hat{\mathbf{Z}}$ in this description. In practice, these projections map the tokens to a $d$-dimensional latent space. The computation is formulated as $\hat{\mathbf{Z}} = \mathbf{W}_{\text{out}}\text{TF}(\mathbf{W}_{\text{in}}\mathbf{Z} + \mathbf{b}_{\text{in}}) + \mathbf{b}_{\text{out}}$, where $\mathbf{W}_{\text{in}} \in \mathbb{R}^{d\times(a+b)}$ and $\mathbf{W}_{\text{out}} \in \mathbb{R}^{(a+b)\times d}$ are the linear layer weights, and $\mathbf{b}_{\text{in}}$ and $\mathbf{b}_{\text{out}}$ are the biases. The ICL settings allow the model to use $\{\mathbf{x}_1, \mathbf{y}_1, \cdots, \mathbf{x}_{i-1}, \mathbf{y}_{i-1}, \mathbf{x}_i\}$ to determine the best predictor $f_i^* : \mathbf{x}_i \to \mathbf{y}_i$.

## 2.2 FORMULATION OF TSF TRANSFORMERS

In this section, we present the existing Temporal-wise and Series-wise Transformers for TSF from an ICL perspective. Building on this, we introduce the In-context Time Series Predictor (ICTSP). In §2.3, we discuss the issues within existing structures and the advantages of ICTSP for TSF problems.

**Temporal-wise Transformer** This type of method, as shown in Figure 2 (a), typically constructs input tokens based on the values of multiple time series channels at timesteps or within intervals

along the temporal dimension. The format of these input tokens can be represented as:

$$\mathbf{Z}^{C \times L} = [\mathbf{z}_1, \ldots, \mathbf{z}_L] = \begin{bmatrix} \mathbf{P}_{1:L_I} & \mathbf{P}_{L_I+1:L_I+L_P} \\ X_I & \mathbf{O} \end{bmatrix} = \begin{bmatrix} \mathbf{p}_1 & \cdots & \mathbf{p}_{L_I} & \mathbf{p}_{L_I+1} & \cdots & \mathbf{p}_{L_I+L_P} \\ X_1^{(1)} & \cdots & X_1^{(L_I)} & o_1 & \cdots & o_1 \\ \vdots & \ddots & \vdots & \vdots & \ddots & \vdots \\ X_C^{(1)} & \cdots & X_C^{(L_I)} & o_C & \cdots & o_C \end{bmatrix}, \quad (3)$$

where $\mathbf{O}$ and $\{o_1, \cdots, o_C\}$ represent zero-padding over the forecasting horizon. Again, the added positional embeddings $\mathbf{p}.$ are shown concatenated here just for illustration. Each token represents the time series channel structure at a specific timestep. Therefore, from the ICL perspective, the Transformer doesn't learn the necessary prediction from $X_I$ to $X_P$ for TSF. Instead, it learns the mapping $f_t^* : \mathbf{p}_t \to [X_1^{(t)}, \cdots, X_C^{(t)}]^\top$, which tries to find the underlying dynamic of the channel structure determined by the position from the context and applies it to the outputs. If true dependencies between the input series are lacking, which frequently occurs in real-world data, its focus on channel structure over temporal dependencies leads to significant overfitting. In §2.3, we will explain how existing methods like channel independence can partially address overfitting while introducing new issues. In the last part of §3, we will use examples to show that Temporal-wise Transformers perform well with strong channel dependencies but overfit when these dependencies are weak. Previous models using this architecture include LogTrans(Li et al., 2019), Informer (Zhou et al., 2021), Autoformer (Wu et al., 2021), etc. Note that while some of these models use an encoder-decoder structure, their token construction based on multi-channel values along the temporal dimension shares the same issues we highlight.

**Series-wise Transformer**     Some recent methods (Liu et al., 2024a; Wang et al., 2024) transpose the input format above so each token represents an individual series, forming the Series-wise Transformer. This format, shown in Figure 2 (b), can be expressed as follows:

$$\mathbf{Z}^{L \times C} = [\mathbf{z}_1, \ldots, \mathbf{z}_C] = \begin{bmatrix} \mathbf{P}_{1:C} \\ X_I^\top \\ \mathbf{O} \end{bmatrix} = \begin{bmatrix} \mathbf{p}_1 & X_1^{(1)} & \ldots & X_1^{(L_I)} & o_{L_I+1} & \ldots & o_L \\ \vdots & \vdots & \ddots & \vdots & \vdots & \ddots & \vdots \\ \mathbf{p}_C & X_C^{(1)} & \ldots & X_C^{(L_I)} & o_{L_I+1} & \ldots & o_L \end{bmatrix}^\top . \quad (4)$$

However, iTransformer (Liu et al., 2024a) introduced this formulation believing it enhances modeling of inter-series dependencies compared to the Temporal-wise Transformer, which contrasts with our ICL analysis. Each token in the Series-wise Transformer represents an entire input series, capturing relationships between timesteps within that series. For series $j$, the Transformer learns a predictor $f_j^* : X_j^{(1:L_I)} \to X_j^{(L_I+1:L_I+L_P)}$, which also adjusts its own parameters to the inputs $\{X_i^{(1:L_I)}\}_{i \neq j}$ from other series. Essentially, it acts more like a univariate predictor capable of adapting its parameters based on the context series. It is capable of modeling temporally-aligned inter-series dependencies, but it is difficult to model shifted dependencies across timesteps.Similar to supervised learning, ICL treats each input within a token as independent features. This means that values across different timesteps are considered distinct features. The algorithm or model generated by ICL excels at building relationships within input features and can build relationship on the same feature (i.e. same timestep) across different data points (different series here) by performing linear combinations. However, it is less inclined to establish relationships between different features across different data points. In traditional supervised learning settings, different features from different data points are usually unrelated, so no relationship is built, but this contradicts the requirements of multivariate TSF settings when using this series-as-datapoint setting. We show in the last part of §3 that the Series-wise Transformer is better suited for real-world datasets with weak multivariate relationships than those with strong shifted inter-series dependencies, supporting our view from ICL. Also from this perspective, the Series-wise Transformer lacks ground truth context examples for the predictor, meaning it doesn't fully utilize the Transformer's ICL abilities. This limits its generalization and few-shot learning capabilities.

**In-context Time Series Predictor**     To fully exploit the Transformer's ICL capabilities, we treat forecasting tasks, instead of timestep values or input series, as tokens. For each input series $j$, we use a lookback length $L_b < L_I$ to construct context examples based on input data $X_j^{(1:L_I)}$. We perform stepwise sampling on input, obtaining $N = L_I - L_b - L_P$ ground truth training samples with their lookback and future parts span of length $L_b + L_P$. By combining each series' output target token

with the context examples to form each series' input $\mathbf{H}_j$, we concatenate these into the multivariate input token matrix $\mathbf{Z}$ for the ICTSP as follows:

$$\mathbf{Z}^{(L_b+L_P)\times(N+1)C} = [\mathbf{z}_1, \ldots, \mathbf{z}_{(N+1)C}] = [\mathbf{H}_1, \ldots, \mathbf{H}_C]$$

$$\mathbf{H}_j^{(L_b+L_P)\times(N+1)} = \begin{bmatrix} \mathbf{p}_{j,1} & X_j^{(1)} & \cdots & X_j^{(L_b)} & X_j^{(L_b+1)} & \cdots & X_j^{(L_b+L_P)} \\ \mathbf{p}_{j,2} & X_j^{(2)} & \cdots & X_j^{(L_b+1)} & X_j^{(L_b+2)} & \cdots & X_j^{(L_b+L_P+1)} \\ \vdots & \vdots & \ddots & \vdots & \vdots & \ddots & \vdots \\ \mathbf{p}_{j,N} & X_j^{(L_I-L_P-L_b)} & \cdots & X_j^{(L_I-L_P)} & X_j^{(L_I-L_P+1)} & \cdots & X_j^{(L_I)} \\ \mathbf{p}_{j,\text{target}} & X_j^{(L_I-L_b)} & \cdots & X_j^{(L_I)} & o_{L_I+1} & \cdots & o_{L_I+L_P} \end{bmatrix}^{\top}$$

(5)

where $\mathbf{p}_{j,\cdot}$ represents the positional embedding for each token on the $j$-th series illustrated as concatenation. Therefore, $\mathbf{Z}$ contains $(N+1)C$ tokens, with $NC$ context tokens. We can increase the step size in stepwise sampling to reduce computational costs, retaining one sample for every $m$ steps among $N$ samples per series. Figure 2 (c) shows a special case with no overlap between samples for simplicity, achievable when $m = L_b + L_P$. ICTSP flexibly forms ground truth forecasting examples to learn the predictor $f_j^* : X_j^{(t+1:t+L_b)} \to X_j^{(t+L_b+1:t+L_b+L_P)}$. Since ICTSP does not restrict the input channel structure (with $\mathbf{p}_{j,\cdot}$ serving only to distinguish) and can construct context examples during inference, it generalizes well across different multivariate datasets, exhibiting robust zero-shot learning capabilities.

**Adaptive Model Reduction of ICTSP** Existing complex Temporal-wise Transformers often struggle to outperform simple baselines, like linear predictors, in noisy datasets lacking clear temporal-wise and channel-wise dependencies. In contrast, ICTSP adapts to different dataset complexities and reduces to simpler models when necessary. Specifically: i) When temporal-wise dependencies are weak, ICTSP can adaptively select simpler functions for TSF, such as linear predictors and shallow MLPs, as achievable by ICL. ii) When series patterns change greatly over time where the impact of context examples is weak, ICTSP can reduce to a Series-wise Transformer focusing only on local lookback of target tokens without context tokens. iii) When local lookback from other series have minimal contribution, ICTSP degrades to a simple univariate MLP, in which case the inter-token information in attention term from Eq.equation 1 disappears, and token $j$ at layer $k$ becomes $\mathbf{z}_{j,k} = \mathbf{z}_{j,k-1} + \texttt{LN}(\texttt{FFN}_{j,k}(\mathbf{z}_{j,k-1}))$. If FFN also becomes ineffective, ICTSP processes target tokens only with input and output linear projections outside the Transformer. At this stage, the model reduces to a linear predictor, effective for highly noisy time series data. ICTSP's design provides appropriate shortcuts to enhance adaptability. We further explain in §A.4.1 how inappropriate shortcuts in the model reduction of the Temporal-wise Transformer limit its ability to learn dynamic patterns.

**Sampling and Token Retrieval** We can reduce computational costs of ICTSP by sampling one context token every $m$ steps from the total $N$ tokens for each series. For very large token amount, to avoid losing information with large sampling steps, we can use token retrieval (TR) to further reduce token size. We employ a simple method where each token $\mathbf{z}_i$ is reduced to a $\delta$-dimensional latent vector $\tilde{\mathbf{z}}_i$ via a linear layer. We calculate the cosine similarity between the $\tilde{\mathbf{z}}_i$ pair of each context token and target token, averaging the results for each context token. We rank the context tokens by similarity, select the top $q\%$ as context examples, and merge the remaining tokens into $r$ tokens by grouped weighted averaging, resulting in $\lfloor q(L_I - L_b - L_P)/m \rfloor C + r$ context tokens. TR reduces computational costs, preserves enough information, and helps the model infer which examples are more important with token positions. More details of the ICTSP structure can be found in §A.3.

## 2.3 Solving Key Issues of TSF Transformers

**Timestep Mixing and Overfitting** From the ICL perspective, the Temporal-wise Transformer's token formulation overemphasizes channel relationships within specific timesteps. This leads to outputs overfitted to the non-existent underlying channel structures of context examples in real-world datasets with weak inter-series relationships, as shown in Figure 3 (a). Previous studies attribute this overfitting to the mixing of channel values at timesteps in the attention mechanism, proposing various solutions. The following analysis shows these partially effective solutions are actually trying to enhance the focus on temporal dependencies in the token formulation.

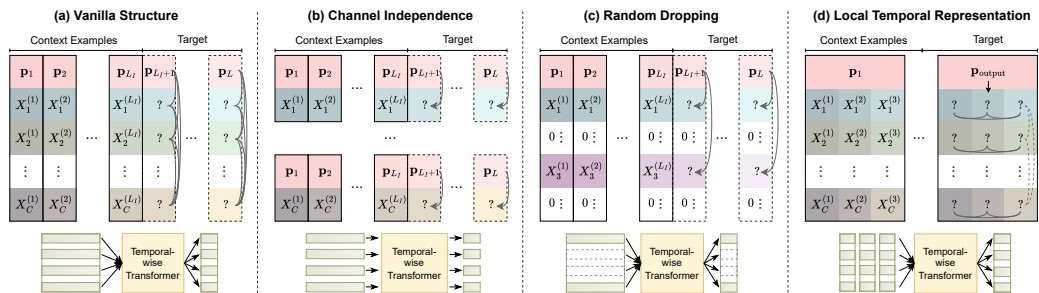

Figure 3: Previous solutions of Temporal-wise Transformers' overfitting issue. From the ICL perspective, they are actually introducing more learnable temporal dependencies within token formulation.

PatchTST (Nie et al., 2022) suggested that univariate Temporal-wise Transformers, which inputs each channel independently (Figure 3 (b)), can address overfitting. This approach breaks the series-wise relationship in the tokens, transforming the ICL predictor into $f_{j,t}^* : \mathbf{p}_t \to X_j^{(t)}$, which predicts values solely from positions. This shift enhances the model's focus on temporal dependencies while ignoring series-wise dependencies, making the formulation more suitable for real-world TSF tasks. Notably, the prompt structure used by recent LLMs for TSF, "... timestep: value ...", also benefits from this approach. Existing methods like ForecastPFN (Dooley et al., 2024) use similar structures.

ARM (Lu et al., 2024) introduced random dropping (Figure 3 (c)), which mitigates overfitting by randomly removing series from both input and output during training. This randomly breaks some relationships between channels within the tokens. With only positional information consistently preserved, the model focus more on the temporal generation process and retains only the most likely channel dependencies, reducing overfitting to non-existent channel relationships. However, the degree of channel dropout requires further tuning based on different dataset characteristics.

Methods such as LogTrans (Li et al., 2019) and ARM (Lu et al., 2024) use strategies similar to Figure 3 (d) to build local representations for tokens across time intervals instead of single timesteps. This introduces temporal relationships within the token. For TSF data with weak channel dependencies, tokens naturally focus on intra-series temporal relationships (solid lines in Figure 3 (d)) rather than the weak channel-wise relationships (dashed lines in Figure 3 (d)) as in the original token format. This approach alleviates overfitting but requires careful tuning of the local window selection. Recent methods like Liu et al. (2024b) tend to combine method (c) and (d) to further improve the generalization ability.

These previous solutions emphasize temporal dependencies more in tokens but do not address the fundamental formulation issue. The Series-wise Transformer solve the issue by treating input series as tokens, helping each token focus on temporal dependencies but lacks contextual examples and ICL capabilities. In contrast, the ICTSP treats forecasting tasks as tokens, providing context forecasting examples to help the model learn temporal predictors and potential series dependencies through interactions between the target token and the historical context of all the series.

**Permutation Invariance**  Studies like DLinear (Zeng et al., 2023) and iTransformer (Liu et al., 2024a) suggest that the poor performance of Temporal-wise Transformers is partly due to the inherent permutation invariance issue in Transformer, meaning swapping the positions of input tokens does not significantly affect the output. The Temporal-wise Transformer treats context tokens more as representations of the underlying channel structure. Thus, when a dataset lacks a temporally varying underlying channel structure, $\mathbf{p}_t$ in $f_t^* : \mathbf{p}_t \to [X_1^{(t)}, \cdots, X_C^{(t)}]^\top$ becomes ineffective, and the output tends to converge to the context's average. This is consistent with DLinear observations (Zeng et al., 2023) where shuffling of input token minimally impact the output. Conversely, in the ICTSP, each context token represents a forecasting training example, so swapping their positions does not harm the forecasting. This makes the ICTSP better suited to the Transformer's characteristics for TSF compared to previous structures.

**Channel Structure Restriction**  In the Temporal-wise Transformer, the number of input series and their relative positions within the tokens must be fixed. This limits the model's ability to adapt to datasets with varying series numbers. Channel independence (Nie et al., 2022) may simplify the

Temporal-wise Transformer to a univariate model, enhancing generalization but losing the ability to model inter-series dependencies. The Series-wise Transformer does not restrict the number of input series structurally but learns the characteristics of $C$ series during training through $\mathbf{p}_j$, making it hard to transfer embeddings to other datasets. Conversely, the ICTSP does not restrict the number of series and includes in-context training examples within the input, enabling easy transfer to input data of any size. Here, positional embeddings can be trained to only distinguish between samples without describing the characteristics of the series. The flexibility in input channel structure provides zero-shot multivariate forecasting ability, which previous multivariate models could not achieve.

## 3 EXPERIMENTS

We conduct comprehensive experiments under full-data, few-shot, and zero-shot settings using widely-used TSF datasets (details in §A.2), including ETTs (Zhou et al., 2021), Traffic, Electricity (ECL), and Weather. We use $K = 3$ TF layers with $d = 128$ and 8 heads. We set $L_I = 1440$, $L_b = 512$, and $L_P \in \{96, 192, 336, 720\}$, performing 4 experiments for each dataset. We use sampling step $m = 8$ and the token retrieval method with $q = 10\%$, $r = 30$ in main experiments. See §A.3.2 for more hyper-parameter and implementation details. We use the same experimental environment as (Zhou et al., 2023; Jin et al., 2024). Extensive efforts are made to ensure fair comparisons across all experiments (detailed in §A.3.3).

**Baselines** We compare our method with previous SOTA models in the categories of LLM for TSF (Time-LLM (Jin et al., 2024), GPT4TS (Zhou et al., 2023), LLMTime (Gruver et al., 2023)), Temporal-wise Transformers (PatchTST (Nie et al., 2022), FEDformer (Zhou et al., 2022), Autoformer (Wu et al., 2021), Informer (Zhou et al., 2021)), Series-wise Transformer (iTransformer (Liu et al., 2024a)), CNN (TimesNet (Wu et al., 2022)), and simple methods (DLinear (Zeng et al., 2023), Last-value Repeat). Baseline results are sourced from (Zhou et al., 2023; Jin et al., 2024) when applicable and rerun for the missing experiments. We report the average test set MSE for each dataset in the main paper and provide the full raw results in §A.1. The average rank and the number of times each model is ranked as the best are also summarized in the results.

**Full-data TSF Results** We train the ICTSP from scratch on TSF datasets, with the results shown in Table 1. ICTSP consistently shows superior performance compared to previous methods in all categories. The ICL capability, activated by the context examples, allows it to excel on larger datasets with stable patterns like ECL and Traffic. Its adaptive model reduction ability, originated from its token and structure design, ensures strong performance on small and noisy datasets like ETTs.

Table 1: Full-data TSF results. Averaged test set MSE on each dataset is reported. The best and second-best results are in bold and underlined, respectively. See Table 6 for the original results.

| Models | ICTSP | Time-LLM | GPT4TS | iTransformer | PatchTST | FEDformer | Autoformer | Informer | DLinear | Repeat |
|--------|-------|----------|--------|--------------|----------|-----------|------------|----------|---------|--------|
| ETTh1 | **0.404** | 0.408 | 0.428 | 0.454 | 0.413 | 0.440 | 0.496 | 1.040 | 0.423 | 1.321 |
| ETTh2 | **0.328** | 0.334 | 0.355 | 0.375 | 0.330 | 0.437 | 0.450 | 4.431 | 0.431 | 0.536 |
| ETTm1 | 0.342 | **0.329** | 0.352 | 0.374 | 0.351 | 0.448 | 0.588 | 0.961 | 0.357 | 1.269 |
| ETTm2 | **0.247** | 0.251 | 0.267 | 0.269 | 0.255 | 0.305 | 0.327 | 1.410 | 0.267 | 0.385 |
| Weather | **0.218** | 0.226 | 0.237 | 0.249 | 0.226 | 0.309 | 0.338 | 0.634 | 0.249 | 0.353 |
| ECL | **0.154** | 0.159 | 0.167 | 0.173 | 0.162 | 0.214 | 0.227 | 0.311 | 0.166 | 1.612 |
| Traffic | **0.386** | 0.388 | 0.414 | 0.420 | 0.391 | 0.610 | 0.628 | 0.764 | 0.434 | 2.770 |
| AvgRank | **1.36** | 1.93 | 4.29 | 5.64 | 2.79 | 6.86 | 7.89 | 9.43 | 5.00 | 9.50 |
| #Rank1 | **18** | 8 | 0 | 0 | 2 | 0 | 0 | 0 | 0 | 0 |

**Few-shot Learning** The few-shot learning ability of ICTSP is tested by training it on only the first 10% or 5% of the training data. ICTSP can flexibly build context samples based on the input data to perform in-context fitting without parameter updates, which compensates for the lack of training samples closer to the test set. Thus, it stably outperforms other models in both the 10% and 5% settings, as shown in Table 2. Note that we ensured ICTSP only uses datapoints perceivable by other baselines to build the context examples, ensuring a fair comparison, as detailed in A.3.3.

**Zero-shot Learning** We also test the zero-shot transfer learning ability of ICTSP by training on one ETT dataset and testing on another ETT dataset. ICTSP surpasses all previous methods by a large margin in the zero-shot settings, as shown in Table 3. The new patterns in unseen datasets can be prompted with the forecasting task examples constructed in the tokens of ICTSP, which perfectly fits the scenario of zero-shot transfer learning.

Table 2: Few-shot learning results on 10% and 5% training data. The best and second best results are in bold and underlined, respectively. See Table 7 and 8 for the original results.

| Models | ICTSP | Time-LLM | GPT4TS | PatchTST | FEDformer | Autoformer | Informer | TimesNet | DLinear | Repeat |
|---|---|---|---|---|---|---|---|---|---|---|
| ETTh1 (10%) | **0.525** | 0.555 | 0.590 | 0.633 | 0.639 | 0.702 | 1.199 | 0.869 | 0.691 | 1.321 |
| ETTh2 (10%) | **0.369** | 0.371 | 0.397 | 0.415 | 0.466 | 0.488 | 3.872 | 0.479 | 0.605 | 0.536 |
| ETTm1 (10%) | **0.397** | 0.404 | 0.464 | 0.501 | 0.722 | 0.802 | 1.192 | 0.677 | 0.411 | 1.269 |
| ETTm2 (10%) | **0.275** | 0.277 | 0.293 | 0.296 | 0.463 | 1.342 | 3.370 | 0.320 | 0.316 | 0.385 |
| Weather (10%) | 0.237 | **0.234** | 0.238 | 0.242 | 0.284 | 0.300 | 0.597 | 0.279 | 0.241 | 0.353 |
| ECL (10%) | **0.174** | 0.175 | 0.176 | 0.180 | 0.346 | 0.431 | 1.195 | 0.323 | 0.180 | 1.612 |
| Traffic (10%) | **0.428** | 0.429 | 0.440 | 0.430 | 0.663 | 0.749 | 1.534 | 0.951 | 0.447 | 2.770 |
| AvgRank (10%) | **1.57** | 1.89 | 3.21 | 4.07 | 6.39 | 7.46 | 9.43 | 6.68 | 4.93 | 9.21 |
| #Rank1 (10%) | **15** | 10 | 1 | 1 | 0 | 0 | 0 | 0 | 0 | 0 |
| ETTh1 (5%) | **0.540** | 0.627 | 0.682 | 0.695 | 0.659 | 0.722 | 1.225 | 0.926 | 0.750 | 1.314 |
| ETTh2 (5%) | **0.368** | 0.382 | 0.401 | 0.439 | 0.441 | 0.470 | 3.923 | 0.464 | 0.828 | 0.519 |
| ETTm1 (5%) | **0.397** | 0.425 | 0.472 | 0.527 | 0.731 | 0.796 | 1.163 | 0.717 | 0.401 | 1.269 |
| ETTm2 (5%) | 0.285 | **0.274** | 0.308 | 0.315 | 0.381 | 0.389 | 3.659 | 0.345 | 0.399 | 0.385 |
| Weather (5%) | **0.256** | 0.261 | 0.264 | 0.270 | 0.310 | 0.311 | 0.584 | 0.298 | 0.264 | 0.353 |
| ECL (5%) | **0.175** | 0.177 | 0.179 | 0.181 | 0.267 | 0.346 | 1.281 | 0.402 | 0.177 | 1.612 |
| Traffic (5%) | **0.418** | 0.423 | 0.434 | **0.418** | 0.677 | 0.833 | 1.591 | 0.867 | 0.451 | 2.757 |
| AvgRank (5%) | **1.56** | 2.36 | 3.48 | 3.96 | 6.04 | 7.08 | 9.44 | 6.80 | 4.84 | 9.24 |
| #Rank1 (5%) | **12** | 7 | 1 | 2 | 0 | 0 | 0 | 0 | 4 | 0 |

Table 3: Zero-shot transfer learning results. The best and second best results are in bold and underlined, respectively. See Table 9 for the original results.

| Models | ICTSP | Time-LLM | LLMTime | GPT4TS | PatchTST | Autoformer | TimesNet | DLinear | Repeat |
|---|---|---|---|---|---|---|---|---|---|
| ETTh1→ETTh2 | **0.337** | 0.352 | 0.992 | 0.407 | 0.381 | 0.582 | 0.421 | 0.494 | 0.536 |
| ETTh2→ETTh1 | **0.441** | 0.479 | 1.961 | 0.758 | 0.565 | 0.758 | 0.866 | 0.703 | 1.321 |
| ETTm1→ETTh2 | **0.356** | 0.381 | 0.992 | 0.433 | 0.439 | 0.470 | 0.454 | 0.464 | 0.536 |
| ETTm1→ETTm2 | **0.255** | 0.265 | 1.867 | 0.314 | 0.297 | 0.469 | 0.322 | 0.335 | 0.385 |
| ETTm2→ETTh2 | **0.352** | 0.354 | 0.992 | 0.435 | 0.409 | 0.423 | 0.435 | 0.456 | 0.536 |
| ETTm2→ETTm1 | **0.408** | 0.414 | 1.933 | 0.769 | 0.568 | 0.755 | 0.769 | 0.649 | 1.269 |
| AvgRank | **1.13** | 1.88 | 8.67 | 4.54 | 3.29 | 6.21 | 5.75 | 5.33 | 7.88 |
| #Rank1 | **21** | 3 | 0 | 0 | 0 | 0 | 0 | 0 | 0 |

**Ablation Studies** ICTSP has a concise model structure, with forecasting tasks as input tokens, enabling efficient transfer of ICL abilities to TSF. We conducted experiments on a) the inclusion of context examples, b) token retrieval (TR), and c) different sampling steps $m$, as shown in Table 4. Without context example tokens, ICTSP reduces to a Series-wise Transformer, lacking ground truth forecasting references, which significantly reduces performance. Token retrieval is used to lower computational costs while preserving model performance. As observed, using TR in the full model does not significantly reduce performance compared to the original model without TR. The sampling step $m$ controls the density of context samples that ICTSP can see. We aim to choose the largest $m$ that does not lose information to reduce computational cost. Selecting $m = 8$ does not lose much performance compared to using full samples with $m = 1$. while increasing $m$ to 256 degrades performance to the level close to the scenarios without context examples.

Table 4: Results of the ablation studies of the ICTSP structure with $L_P \in \{96, 192, 336, 720\}$.

| Models | Full ($m = 8$) | | w/o Context | | w/o TR | | $m = 1$ | | $m = 64$ | | $m = 256$ | |
|---|---|---|---|---|---|---|---|---|---|---|---|---|
| Metric | MSE | MAE | MSE | MAE | MSE | MAE | MSE | MAE | MSE | MAE | MSE | MAE |
| Weather (96) | **0.139** | **0.194** | 0.150 | 0.202 | 0.140 | 0.200 | 0.140 | 0.196 | 0.140 | **0.194** | 0.142 | 0.195 |
| Weather (192) | 0.186 | 0.236 | 0.195 | 0.241 | 0.188 | 0.240 | **0.185** | **0.233** | **0.185** | 0.234 | 0.187 | 0.239 |
| Weather (336) | **0.239** | **0.279** | 0.245 | 0.280 | 0.241 | 0.287 | 0.240 | 0.284 | **0.239** | 0.281 | 0.240 | 0.283 |
| Weather (720) | **0.306** | **0.320** | 0.309 | 0.324 | 0.307 | 0.328 | 0.307 | 0.321 | 0.310 | 0.325 | 0.312 | 0.327 |
| Weather (Avg) | **0.218** | **0.257** | 0.225 | 0.262 | 0.219 | 0.264 | **0.218** | 0.259 | 0.219 | 0.259 | 0.220 | 0.261 |
| ETTm2 (96) | **0.159** | **0.248** | 0.162 | 0.249 | **0.159** | **0.248** | 0.160 | 0.249 | 0.162 | 0.249 | 0.162 | 0.250 |
| ETTm2 (192) | 0.212 | 0.289 | 0.215 | 0.295 | 0.211 | **0.287** | **0.209** | 0.289 | 0.213 | **0.287** | 0.214 | 0.288 |
| ETTm2 (336) | 0.268 | 0.326 | 0.273 | 0.329 | **0.268** | 0.330 | 0.270 | 0.327 | 0.269 | 0.327 | 0.273 | 0.327 |
| ETTm2 (720) | 0.347 | 0.382 | 0.352 | 0.387 | 0.345 | 0.383 | **0.345** | **0.381** | 0.348 | 0.384 | 0.350 | 0.386 |
| ETTm2 (Avg) | 0.247 | **0.311** | 0.251 | 0.315 | **0.246** | 0.312 | **0.246** | 0.312 | 0.248 | 0.312 | 0.250 | 0.313 |

**Computational Costs** In Table 5, we compare the computational costs between different settings of $m$ and token retrieval (TR), as well as between ICTSP and other TSF Transformers. The total number of context tokens is $\lfloor q(L_I - L_b - L_P)/m \rfloor C + r$. Using $m = 8$ and token retrieval with $q = 10\%$ reduces the first part of tokens to $1.25\%$ of the original amount without losing much information, as shown above. These settings effectively reduce computational costs compared to the full context token scenario, making ICTSP more efficient than most of the previous TSF Transformers. Note that in our setting with a fixed input length $L_I$, longer forecasting horizon $L_P$ result in fewer context samples and may have less computational costs than shorter $L_P$. We do not include the computational

costs of LLMs for TSF here, as they usually have thousands of times more parameters, making a direct comparison unnecessary.

Table 5: Comparison of computational costs. The data format of ETTh2 is utilized to construct model inputs. The hyper-parameters for every model are set according to their default configurations.

| Models | **ICTSP** ($m = 8$, w/ TR) | | ICTSP ($m = 8$, w/o TR) | | ICTSP ($m = 1$, w/ TR) | | ICTSP ($m = 1$, w/o TR) | |
|---|---|---|---|---|---|---|---|---|
| Metric | FLOPs | Params | FLOPs | Params | FLOPs | Params | FLOPs | Params |
| $L_P = 96$ | 1.51G | 7.35M | 11.2G | 7.34M | 9.58G | 7.35M | 88.0G | 7.34M |
| $L_P = 192$ | 1.40G | 7.43M | 9.96G | 7.41M | 9.61G | 7.43M | 78.2G | 7.41M |
| $L_P = 336$ | 1.20G | 7.54M | 8.09G | 7.52M | 8.14G | 7.54M | 63.2G | 7.52M |
| $L_P = 720$ | 605M | 7.83M | 3.02G | 7.82M | 3.29G | 7.83M | 22.6G | 7.82M |
| Models | Autoformer | | Informer | | PatchTST | | iTransformer | |
| Metric | FLOPs | Params | FLOPs | Params | FLOPs | Params | FLOPs | Params |
| $L_P = 96$ | 10.9G | 10.5M | 9.41G | 11.3M | 12.6G | 10.7M | 148M | 6.72M |
| $L_P = 192$ | 11.6G | 10.5M | 10.1G | 11.3M | 12.7G | 15.2M | 149M | 6.77M |
| $L_P = 336$ | 12.7G | 10.5M | 11.2G | 11.3M | 12.8G | 21.8M | 151M | 6.85M |
| $L_P = 720$ | 15.5G | 10.5M | 14.0G | 11.3M | 13.2G | 39.5M | 155M | 7.04M |

**Analysis of TSF Transformer Characteristics** From the ICL perspective, the Temporal-wise Transformer is better for learning inter-series relationships, while the Series-wise Transformer better at intra-series relationships. Here, we use a synthesized "Multi" dataset from (Lu et al., 2024) with strong inter-series dependencies, generated by shifting an random walk process. Fig. 4 (a) and (b) show that the Temporal-wise Transformer performs well on this dataset, while the Series-wise Transformer fails to learn the shifting dependencies. ICTSP, with its context tokens featuring a shifting structure, excels in learning these time-interleaved dependencies. Conversely, the Temporal-wise Transformer struggles with real-world datasets with weak dependencies comparing to Series-wise Transformer, such as the ETTm2 datasets in Fig. 4 (c) and (d). ICTSP handles these datasets well, demonstrating significantly better performance. Note that we build these baselines in this analysis with the same structure shown in Fig. 2, using $L_I = 512$, $L_P = 192$, $K = 3$, $d = 128$ for the 3 models; and $L_b = 256$ for ICTSP. See §A.4.2 for more discussion about the inter-series learning abilities of the TSF Transformers.

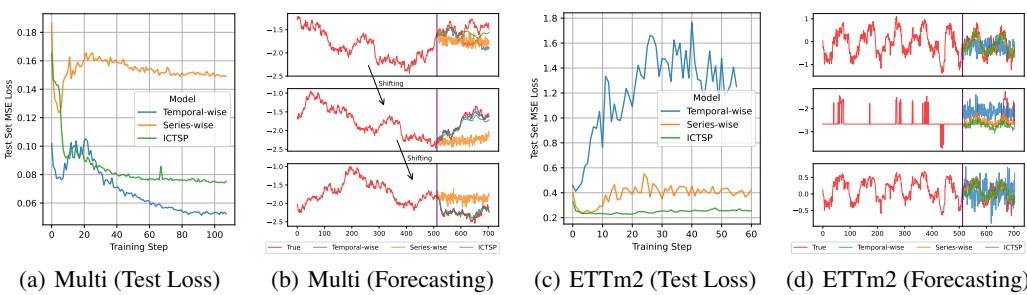

(a) Multi (Test Loss)     (b) Multi (Forecasting)     (c) ETTm2 (Test Loss)     (d) ETTm2 (Forecasting)

Figure 4: Comparison of the 3 architectures. The first 3 series of Multi and ETTm2 are visualized.

**Attention Visualization** We visualize the attention maps of the three TF layers in ICTSP using the same forecasting data points from the Multi and ETTm2 datasets as above. We disable token retrieval to observe the interaction within and between the tokens from each time series. In Fig. 5 (a), it can be observed that the shifted series 2 and 3 fetch information from the corresponding inter-series context tokens. In Fig. 5 (b), ICTSP focuses on the first half of the context tokens, aligning with Fig. 4 (d), where the future shape resembles the earlier part of the input rather than the later part.

## 4 RELATED WORKS

Extensive research on TSF has been conducted in both traditional statistics (Box et al., 1974; Holt, 2004; Sims, 1980) and deep learning (Hochreiter & Schmidhuber, 1997; Rangapuram et al., 2018; Salinas et al., 2020; Xing et al., 2024). Recent studies have seen a surge in Transformer-based TSF models (Wen et al., 2022; Li et al., 2019; Zhou et al., 2021; Wu et al., 2021; Zhou et al., 2022; Zhang & Yan, 2023). However, a study (Zeng et al., 2023) indicated that these complex models might not outperform linear predictors on real-world datasets, leading to subsequent research (Nie

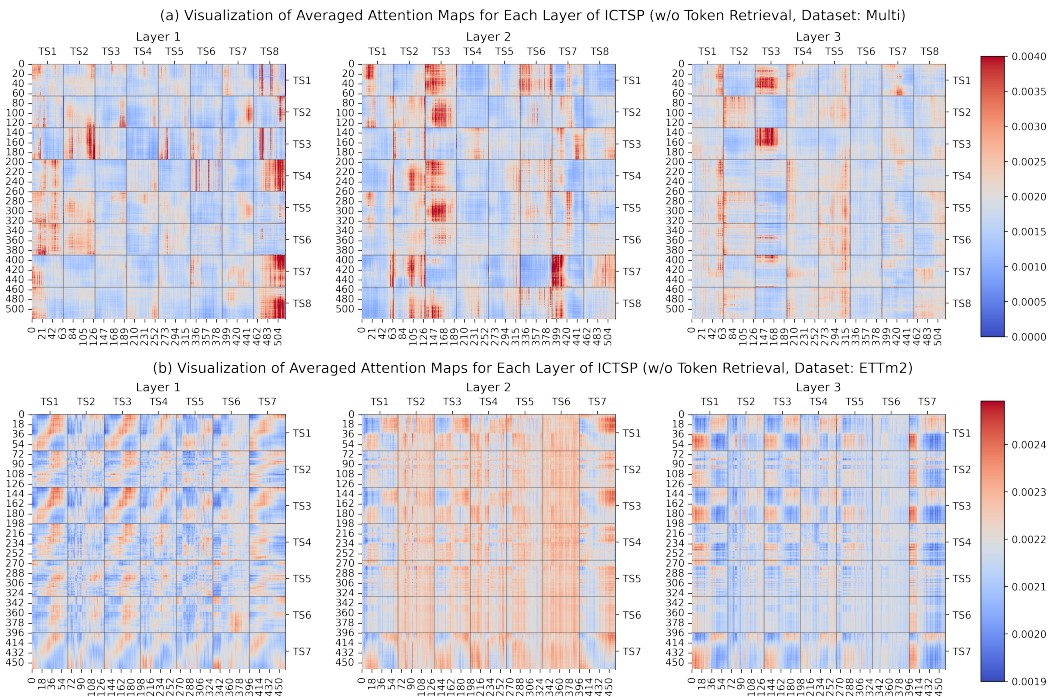

Figure 5: Visualization of averaged attention maps of the 3 TF layers of ICTSP on Multi and ETTm2.

et al., 2022; Liu et al., 2024a; Lu et al., 2024) proposing possible solutions. Additionally, due to the significant achievements of LLMs in various tasks (Brown et al., 2020), several LLM-based TSF models (Gruver et al., 2023; Zhou et al., 2023; Jin et al., 2024) have been introduced. These models also use Transformer architectures but rely on pre-trained parameters and fine-tuning to achieve superior few-shot and zero-shot capabilities (Zhang et al., 2024). While studies (Min et al., 2022; Wei et al., 2023; Xie et al., 2022; Zhang et al., 2023; Garg et al., 2022) have highlighted that these few-shot abilities in LLMs stem from ICL, there is a lack of research on how to efficiently implement ICL in TSF.

## 5 CONCLUSION AND LIMITATION

In this study, we propose the In-context Time Series Predictor (ICTSP) to leverage in-context learning capabilities for time series forecasting. By treating "forecasting tasks" as tokens, we build context forecasting examples into the input to activate the ICL abilities. ICTSP addresses issues in TSF Transformers such as timestep mixing and overfitting with its effective token formulation, achieving SOTA performance across full-data, few-shot, and zero-shot experiments. For limitation, due to the lack of large collective TSF datasets, we have not tested ICTSP's scaling ability with more layers on larger datasets. We also have not explored the possibility of training a single global model that can handle different lookback and future length, as allowed by the ICTSP structure.

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

# A APPENDIX

## A.1 ADDITIONAL EXPERIMENTAL RESULTS

In this section, we provide the detailed raw experimental results presented in the main paper. Table 6 presents the detailed results in the full-data setting. Tables 7 and 8 demonstrate the results of few-shot learning. Table 9 shows the results of zero-shot learning.

Table 6: Full-data results with forecasting horizons $L_P \in \{96, 192, 336, 720\}$. The test set MSE and MAE results are reported. The best and second best results are in bold and underlined, respectively.

| Models | ICTSP | | Time-LLM | | GPT4TS | | iTransformer | | PatchTST | | FedFormer | | Autoformer | | Informer | | TimesNet | | DLinear | | Repeat | |
|---|---|---|---|---|---|---|---|---|---|---|---|---|---|---|---|---|---|---|---|---|---|---|
| Metric | MSE | MAE | MSE | MAE | MSE | MAE | MSE | MAE | MSE | MAE | MSE | MAE | MSE | MAE | MSE | MAE | MSE | MAE | MSE | MAE | MSE | MAE |
| ETTh1 (96) | 0.366 | 0.393 | 0.362 | 0.392 | 0.376 | 0.397 | 0.396 | 0.428 | 0.370 | 0.399 | 0.376 | 0.419 | 0.449 | 0.459 | 0.865 | 0.713 | 0.384 | 0.402 | 0.375 | 0.399 | 1.295 | 0.713 |
| ETTh1 (192) | 0.399 | 0.412 | 0.398 | 0.418 | 0.416 | 0.418 | 0.431 | 0.451 | 0.413 | 0.421 | 0.420 | 0.448 | 0.500 | 0.482 | 1.008 | 0.792 | 0.436 | 0.429 | 0.405 | 0.416 | 1.325 | 0.733 |
| ETTh1 (336) | 0.426 | 0.427 | 0.430 | 0.427 | 0.442 | 0.433 | 0.459 | 0.470 | 0.422 | 0.436 | 0.459 | 0.465 | 0.521 | 0.496 | 1.107 | 0.809 | 0.491 | 0.469 | 0.439 | 0.443 | 1.323 | 0.744 |
| ETTh1 (720) | 0.424 | 0.446 | 0.442 | 0.457 | 0.477 | 0.456 | 0.528 | 0.523 | 0.447 | 0.466 | 0.506 | 0.507 | 0.514 | 0.512 | 1.181 | 0.865 | 0.521 | 0.500 | 0.472 | 0.490 | 1.339 | 0.756 |
| ETTh1 (Avg) | 0.404 | 0.420 | 0.408 | 0.424 | 0.428 | 0.426 | 0.454 | 0.468 | 0.413 | 0.431 | 0.440 | 0.460 | 0.496 | 0.487 | 1.040 | 0.795 | 0.458 | 0.450 | 0.423 | 0.437 | 1.321 | 0.737 |
| ETTh2 (96) | 0.265 | 0.335 | 0.268 | 0.328 | 0.285 | 0.342 | 0.299 | 0.358 | 0.274 | 0.336 | 0.358 | 0.397 | 0.346 | 0.388 | 3.755 | 1.525 | 0.340 | 0.374 | 0.289 | 0.353 | 0.432 | 0.422 |
| ETTh2 (192) | 0.326 | 0.374 | 0.329 | 0.375 | 0.354 | 0.389 | 0.365 | 0.399 | 0.339 | 0.379 | 0.429 | 0.439 | 0.456 | 0.452 | 5.602 | 1.931 | 0.402 | 0.414 | 0.383 | 0.418 | 0.534 | 0.473 |
| ETTh2 (336) | 0.349 | 0.396 | 0.368 | 0.409 | 0.373 | 0.407 | 0.407 | 0.429 | 0.329 | 0.380 | 0.496 | 0.487 | 0.482 | 0.486 | 4.721 | 1.835 | 0.452 | 0.452 | 0.448 | 0.465 | 0.591 | 0.508 |
| ETTh2 (720) | 0.370 | 0.397 | 0.372 | 0.420 | 0.406 | 0.441 | 0.427 | 0.454 | 0.379 | 0.422 | 0.463 | 0.474 | 0.515 | 0.511 | 3.647 | 1.625 | 0.462 | 0.468 | 0.605 | 0.551 | 0.588 | 0.517 |
| ETTh2 (Avg) | 0.328 | 0.376 | 0.334 | 0.383 | 0.355 | 0.395 | 0.375 | 0.410 | 0.330 | 0.379 | 0.437 | 0.449 | 0.450 | 0.459 | 4.431 | 1.729 | 0.414 | 0.427 | 0.431 | 0.447 | 0.536 | 0.480 |
| ETTm1 (96) | 0.280 | 0.336 | 0.272 | 0.334 | 0.292 | 0.346 | 0.325 | 0.376 | 0.290 | 0.342 | 0.379 | 0.419 | 0.505 | 0.475 | 0.672 | 0.571 | 0.338 | 0.375 | 0.299 | 0.343 | 1.214 | 0.665 |
| ETTm1 (192) | 0.318 | 0.361 | 0.310 | 0.358 | 0.332 | 0.372 | 0.356 | 0.391 | 0.332 | 0.369 | 0.426 | 0.441 | 0.553 | 0.496 | 0.795 | 0.669 | 0.374 | 0.387 | 0.335 | 0.365 | 1.261 | 0.690 |
| ETTm1 (336) | 0.355 | 0.383 | 0.352 | 0.384 | 0.366 | 0.394 | 0.382 | 0.405 | 0.366 | 0.392 | 0.445 | 0.459 | 0.621 | 0.537 | 1.212 | 0.871 | 0.410 | 0.411 | 0.369 | 0.386 | 1.283 | 0.707 |
| ETTm1 (720) | 0.416 | 0.415 | 0.383 | 0.411 | 0.417 | 0.421 | 0.432 | 0.434 | 0.416 | 0.420 | 0.543 | 0.490 | 0.671 | 0.561 | 1.166 | 0.823 | 0.478 | 0.450 | 0.425 | 0.421 | 1.319 | 0.729 |
| ETTm1 (Avg) | 0.342 | 0.374 | 0.329 | 0.372 | 0.352 | 0.383 | 0.374 | 0.402 | 0.351 | 0.381 | 0.448 | 0.452 | 0.588 | 0.517 | 0.961 | 0.734 | 0.400 | 0.406 | 0.357 | 0.379 | 1.269 | 0.698 |
| ETTm2 (96) | 0.159 | 0.248 | 0.161 | 0.253 | 0.173 | 0.262 | 0.187 | 0.281 | 0.165 | 0.255 | 0.203 | 0.287 | 0.255 | 0.339 | 0.365 | 0.453 | 0.187 | 0.267 | 0.167 | 0.269 | 0.266 | 0.328 |
| ETTm2 (192) | 0.212 | 0.289 | 0.219 | 0.293 | 0.229 | 0.301 | 0.238 | 0.316 | 0.220 | 0.292 | 0.269 | 0.328 | 0.281 | 0.340 | 0.533 | 0.563 | 0.249 | 0.309 | 0.224 | 0.303 | 0.340 | 0.371 |
| ETTm2 (336) | 0.268 | 0.326 | 0.271 | 0.329 | 0.286 | 0.341 | 0.285 | 0.344 | 0.274 | 0.329 | 0.325 | 0.366 | 0.339 | 0.372 | 1.363 | 0.887 | 0.321 | 0.351 | 0.281 | 0.342 | 0.412 | 0.410 |
| ETTm2 (720) | 0.347 | 0.382 | 0.352 | 0.379 | 0.378 | 0.401 | 0.365 | 0.393 | 0.362 | 0.385 | 0.421 | 0.415 | 0.433 | 0.432 | 3.379 | 1.338 | 0.408 | 0.403 | 0.397 | 0.421 | 0.521 | 0.465 |
| ETTm2 (Avg) | 0.247 | 0.311 | 0.251 | 0.314 | 0.267 | 0.326 | 0.269 | 0.334 | 0.255 | 0.315 | 0.305 | 0.349 | 0.327 | 0.371 | 1.410 | 0.810 | 0.291 | 0.333 | 0.267 | 0.334 | 0.385 | 0.394 |
| Weather (96) | 0.139 | 0.194 | 0.147 | 0.201 | 0.162 | 0.212 | 0.183 | 0.239 | 0.149 | 0.198 | 0.217 | 0.296 | 0.266 | 0.336 | 0.300 | 0.384 | 0.172 | 0.220 | 0.176 | 0.237 | 0.259 | 0.254 |
| Weather (192) | 0.186 | 0.236 | 0.189 | 0.234 | 0.204 | 0.248 | 0.225 | 0.273 | 0.194 | 0.241 | 0.276 | 0.336 | 0.307 | 0.367 | 0.598 | 0.544 | 0.219 | 0.261 | 0.220 | 0.282 | 0.309 | 0.292 |
| Weather (336) | 0.239 | 0.279 | 0.262 | 0.279 | 0.254 | 0.286 | 0.266 | 0.302 | 0.245 | 0.282 | 0.339 | 0.380 | 0.359 | 0.395 | 0.578 | 0.523 | 0.280 | 0.306 | 0.265 | 0.319 | 0.377 | 0.338 |
| Weather (720) | 0.306 | 0.320 | 0.304 | 0.316 | 0.326 | 0.337 | 0.322 | 0.340 | 0.314 | 0.334 | 0.403 | 0.428 | 0.419 | 0.428 | 1.059 | 0.741 | 0.365 | 0.359 | 0.333 | 0.362 | 0.465 | 0.394 |
| Weather (Avg) | 0.218 | 0.257 | 0.226 | 0.258 | 0.237 | 0.271 | 0.249 | 0.289 | 0.226 | 0.264 | 0.309 | 0.360 | 0.338 | 0.382 | 0.634 | 0.548 | 0.259 | 0.287 | 0.249 | 0.300 | 0.353 | 0.320 |
| ECL (96) | 0.127 | 0.221 | 0.131 | 0.224 | 0.139 | 0.238 | 0.145 | 0.249 | 0.129 | 0.222 | 0.193 | 0.308 | 0.201 | 0.317 | 0.274 | 0.368 | 0.168 | 0.272 | 0.140 | 0.237 | 1.588 | 0.946 |
| ECL (192) | 0.148 | 0.241 | 0.152 | 0.241 | 0.153 | 0.251 | 0.166 | 0.269 | 0.157 | 0.240 | 0.201 | 0.315 | 0.222 | 0.334 | 0.296 | 0.386 | 0.184 | 0.289 | 0.153 | 0.249 | 1.595 | 0.950 |
| ECL (336) | 0.158 | 0.257 | 0.160 | 0.248 | 0.169 | 0.266 | 0.176 | 0.270 | 0.163 | 0.259 | 0.214 | 0.329 | 0.231 | 0.338 | 0.300 | 0.394 | 0.198 | 0.300 | 0.169 | 0.267 | 1.617 | 0.961 |
| ECL (720) | 0.181 | 0.283 | 0.192 | 0.298 | 0.206 | 0.297 | 0.206 | 0.283 | 0.197 | 0.290 | 0.246 | 0.355 | 0.254 | 0.361 | 0.373 | 0.439 | 0.220 | 0.320 | 0.203 | 0.301 | 1.647 | 0.975 |
| ECL (Avg) | 0.154 | 0.251 | 0.159 | 0.253 | 0.167 | 0.263 | 0.173 | 0.268 | 0.162 | 0.253 | 0.214 | 0.327 | 0.227 | 0.338 | 0.311 | 0.397 | 0.193 | 0.295 | 0.166 | 0.264 | 1.612 | 0.958 |
| Traffic (96) | 0.359 | 0.251 | 0.362 | 0.248 | 0.388 | 0.282 | 0.381 | 0.266 | 0.360 | 0.249 | 0.587 | 0.366 | 0.613 | 0.388 | 0.719 | 0.391 | 0.593 | 0.321 | 0.410 | 0.282 | 2.723 | 1.079 |
| Traffic (192) | 0.372 | 0.256 | 0.374 | 0.247 | 0.407 | 0.290 | 0.410 | 0.278 | 0.379 | 0.256 | 0.604 | 0.373 | 0.616 | 0.382 | 0.696 | 0.379 | 0.617 | 0.336 | 0.423 | 0.287 | 2.756 | 1.087 |
| Traffic (336) | 0.382 | 0.273 | 0.385 | 0.271 | 0.412 | 0.294 | 0.431 | 0.281 | 0.392 | 0.264 | 0.621 | 0.383 | 0.622 | 0.337 | 0.777 | 0.420 | 0.629 | 0.336 | 0.436 | 0.296 | 2.791 | 1.095 |
| Traffic (720) | 0.432 | 0.291 | 0.430 | 0.288 | 0.450 | 0.312 | 0.458 | 0.299 | 0.432 | 0.286 | 0.626 | 0.382 | 0.660 | 0.408 | 0.864 | 0.472 | 0.640 | 0.350 | 0.466 | 0.315 | 2.811 | 1.097 |
| Traffic (Avg) | 0.386 | 0.268 | 0.388 | 0.264 | 0.414 | 0.295 | 0.420 | 0.281 | 0.391 | 0.264 | 0.610 | 0.376 | 0.628 | 0.379 | 0.764 | 0.416 | 0.620 | 0.336 | 0.434 | 0.295 | 2.770 | 1.090 |

## A.2 DATASETS

Our main TSF experiments are conducted based on commonly used time series forecasting datasets, detailed as follows:

**ETT Datasets**[2] **(Zhou et al., 2021):** This dataset includes load and oil temperature data from electricity transformers, recorded at 15-minute intervals from July 2016 to July 2018. It comprises four subsets: ETTm1, ETTm2, ETTh1, and ETTh2, representing two transformers (identified as 1 and 2) and two time resolutions (15 minutes and 1 hour). Each subset contains seven features related to oil and load of the transformers.

**Electricity Dataset**[3]**:** This dataset covers hourly electricity usage data of 321 consumers from 2012 to 2014 and is frequently used in energy consumption forecasting and analysis.

**Traffic Dataset**[4]**:** Sourced from freeway sensors in the San Francisco Bay area, this dataset provides hourly road occupancy data from 2015 to 2016, making it a key resource for traffic flow studies.

**Weather Dataset**[5]**:** This dataset captures 21 weather variables, such as temperature and humidity, recorded every 10 minutes throughout 2020, supporting detailed meteorological studies.

**Multi Dataset (Lu et al., 2024)** is generated from a master random walk series. The first series is the master series, while the second to fifth series are created by shifting the master series backward by 96, 192, 336, and 720 steps, respectively. The last three series are combinations of these first five series.

---

[2] https://github.com/zhouhaoyi/ETDataset
[3] https://archive.ics.uci.edu/ml/datasets/ElectricityLoadDiagrams20112014
[4] http://pems.dot.ca.gov/
[5] https://www.bgc-jena.mpg.de/wetter/

Table 7: Few-shot learning results on 10% training data with forecasting horizons $L_P \in \{96, 192, 336, 720\}$. The best and second best results are in bold and underlined, respectively.

| Models | ICTSP | | Time-LLM | | GPT4TS | | PatchTST | | FedFormer | | Autoformer | | Informer | | TimesNet | | DLinear | | Repeat | |
|---|---|---|---|---|---|---|---|---|---|---|---|---|---|---|---|---|---|---|---|---|
| Metric | MSE | MAE | MSE | MAE | MSE | MAE | MSE | MAE | MSE | MAE | MSE | MAE | MSE | MAE | MSE | MAE | MSE | MAE | MSE | MAE |
| ETTh1 (96) | 0.411 | 0.428 | 0.448 | 0.460 | 0.458 | 0.456 | 0.516 | 0.485 | 0.512 | 0.499 | 0.613 | 0.552 | 1.179 | 0.792 | 0.861 | 0.628 | 0.492 | 0.495 | 1.295 | 0.713 |
| ETTh1 (192) | 0.465 | 0.462 | 0.484 | 0.483 | 0.570 | 0.516 | 0.598 | 0.524 | 0.624 | 0.555 | 0.722 | 0.598 | 1.199 | 0.806 | 0.797 | 0.593 | 0.565 | 0.538 | 1.325 | 0.733 |
| ETTh1 (336) | 0.512 | 0.498 | 0.589 | 0.540 | 0.608 | 0.535 | 0.657 | 0.550 | 0.691 | 0.574 | 0.750 | 0.619 | 1.202 | 0.811 | 0.941 | 0.648 | 0.721 | 0.622 | 1.323 | 0.744 |
| ETTh1 (720) | 0.710 | 0.612 | 0.700 | 0.604 | 0.725 | 0.591 | 0.762 | 0.610 | 0.721 | 0.614 | 0.721 | 0.616 | 1.217 | 0.825 | 0.877 | 0.641 | 0.986 | 0.743 | 1.339 | 0.756 |
| ETTh1 (Avg) | 0.525 | 0.500 | 0.555 | 0.522 | 0.590 | 0.525 | 0.633 | 0.542 | 0.639 | 0.561 | 0.702 | 0.596 | 1.199 | 0.809 | 0.869 | 0.628 | 0.691 | 0.600 | 1.321 | 0.737 |
| ETTh2 (96) | 0.280 | 0.335 | 0.275 | 0.326 | 0.331 | 0.374 | 0.353 | 0.389 | 0.382 | 0.416 | 0.413 | 0.451 | 3.837 | 1.508 | 0.378 | 0.409 | 0.357 | 0.411 | 0.432 | 0.422 |
| ETTh2 (192) | 0.366 | 0.385 | 0.374 | 0.373 | 0.402 | 0.411 | 0.403 | 0.414 | 0.478 | 0.474 | 0.474 | 0.477 | 3.856 | 1.513 | 0.490 | 0.467 | 0.569 | 0.519 | 0.534 | 0.473 |
| ETTh2 (336) | 0.401 | 0.433 | 0.406 | 0.429 | 0.406 | 0.433 | 0.426 | 0.441 | 0.504 | 0.501 | 0.547 | 0.543 | 3.952 | 1.526 | 0.537 | 0.494 | 0.671 | 0.572 | 0.588 | 0.508 |
| ETTh2 (720) | 0.430 | 0.451 | 0.427 | 0.449 | 0.449 | 0.464 | 0.477 | 0.480 | 0.499 | 0.509 | 0.516 | 0.523 | 3.842 | 1.503 | 0.510 | 0.491 | 0.824 | 0.648 | 0.588 | 0.517 |
| ETTh2 (Avg) | 0.369 | 0.401 | 0.371 | 0.394 | 0.397 | 0.421 | 0.415 | 0.431 | 0.466 | 0.475 | 0.488 | 0.499 | 3.872 | 1.513 | 0.479 | 0.465 | 0.605 | 0.538 | 0.536 | 0.480 |
| ETTm1 (96) | 0.341 | 0.376 | 0.346 | 0.388 | 0.390 | 0.404 | 0.410 | 0.419 | 0.578 | 0.518 | 0.774 | 0.614 | 1.162 | 0.785 | 0.583 | 0.501 | 0.352 | 0.392 | 1.214 | 0.665 |
| ETTm1 (192) | 0.367 | 0.390 | 0.373 | 0.416 | 0.429 | 0.423 | 0.437 | 0.434 | 0.617 | 0.546 | 0.754 | 0.592 | 1.172 | 0.793 | 0.630 | 0.528 | 0.382 | 0.412 | 1.261 | 0.690 |
| ETTm1 (336) | 0.405 | 0.415 | 0.413 | 0.426 | 0.469 | 0.439 | 0.476 | 0.454 | 0.998 | 0.775 | 0.869 | 0.677 | 1.227 | 0.908 | 0.725 | 0.568 | 0.419 | 0.434 | 1.283 | 0.707 |
| ETTm1 (720) | 0.473 | 0.458 | 0.485 | 0.476 | 0.569 | 0.498 | 0.681 | 0.556 | 0.693 | 0.579 | 0.810 | 0.630 | 1.207 | 0.797 | 0.769 | 0.549 | 0.490 | 0.477 | 1.319 | 0.729 |
| ETTm1 (Avg) | 0.397 | 0.410 | 0.404 | 0.427 | 0.464 | 0.441 | 0.501 | 0.466 | 0.722 | 0.605 | 0.802 | 0.628 | 1.192 | 0.821 | 0.677 | 0.537 | 0.411 | 0.429 | 1.269 | 0.698 |
| ETTm2 (96) | 0.176 | 0.258 | 0.177 | 0.261 | 0.188 | 0.269 | 0.191 | 0.274 | 0.291 | 0.399 | 0.352 | 0.454 | 3.203 | 1.407 | 0.212 | 0.285 | 0.213 | 0.303 | 0.266 | 0.328 |
| ETTm2 (192) | 0.239 | 0.307 | 0.241 | 0.314 | 0.251 | 0.309 | 0.252 | 0.317 | 0.307 | 0.379 | 0.694 | 0.691 | 3.112 | 1.387 | 0.270 | 0.323 | 0.278 | 0.345 | 0.340 | 0.371 |
| ETTm2 (336) | 0.288 | 0.336 | 0.274 | 0.327 | 0.307 | 0.346 | 0.306 | 0.353 | 0.543 | 0.559 | 2.408 | 1.407 | 3.255 | 1.421 | 0.323 | 0.353 | 0.338 | 0.385 | 0.412 | 0.410 |
| ETTm2 (720) | 0.395 | 0.391 | 0.417 | 0.390 | 0.426 | 0.417 | 0.433 | 0.427 | 0.712 | 0.614 | 1.913 | 1.166 | 3.909 | 1.543 | 0.474 | 0.449 | 0.436 | 0.440 | 0.521 | 0.465 |
| ETTm2 (Avg) | 0.275 | 0.323 | 0.277 | 0.323 | 0.293 | 0.335 | 0.296 | 0.343 | 0.463 | 0.488 | 1.342 | 0.930 | 3.370 | 1.440 | 0.320 | 0.353 | 0.316 | 0.368 | 0.385 | 0.394 |
| Weather (96) | 0.164 | 0.214 | 0.161 | 0.210 | 0.163 | 0.215 | 0.165 | 0.215 | 0.188 | 0.253 | 0.221 | 0.297 | 0.374 | 0.401 | 0.184 | 0.230 | 0.171 | 0.224 | 0.259 | 0.254 |
| Weather (192) | 0.209 | 0.252 | 0.204 | 0.248 | 0.210 | 0.254 | 0.210 | 0.257 | 0.250 | 0.304 | 0.270 | 0.322 | 0.552 | 0.478 | 0.245 | 0.283 | 0.215 | 0.263 | 0.309 | 0.292 |
| Weather (336) | 0.259 | 0.294 | 0.261 | 0.302 | 0.256 | 0.292 | 0.259 | 0.297 | 0.312 | 0.346 | 0.320 | 0.351 | 0.724 | 0.541 | 0.305 | 0.321 | 0.258 | 0.299 | 0.377 | 0.338 |
| Weather (720) | 0.315 | 0.333 | 0.309 | 0.332 | 0.321 | 0.339 | 0.332 | 0.346 | 0.387 | 0.393 | 0.390 | 0.396 | 0.739 | 0.558 | 0.381 | 0.371 | 0.320 | 0.346 | 0.465 | 0.394 |
| Weather (Avg) | 0.237 | 0.273 | 0.234 | 0.273 | 0.238 | 0.275 | 0.242 | 0.279 | 0.284 | 0.324 | 0.300 | 0.342 | 0.597 | 0.495 | 0.279 | 0.301 | 0.241 | 0.283 | 0.353 | 0.320 |
| ECL (96) | 0.138 | 0.236 | 0.139 | 0.241 | 0.139 | 0.237 | 0.140 | 0.238 | 0.231 | 0.323 | 0.261 | 0.348 | 1.259 | 0.919 | 0.299 | 0.373 | 0.150 | 0.253 | 1.588 | 0.946 |
| ECL (192) | 0.153 | 0.252 | 0.151 | 0.248 | 0.156 | 0.252 | 0.160 | 0.255 | 0.261 | 0.356 | 0.338 | 0.406 | 1.160 | 0.873 | 0.305 | 0.379 | 0.164 | 0.264 | 1.595 | 0.950 |
| ECL (336) | 0.174 | 0.272 | 0.169 | 0.270 | 0.175 | 0.270 | 0.180 | 0.276 | 0.360 | 0.445 | 0.410 | 0.474 | 1.157 | 0.872 | 0.319 | 0.391 | 0.181 | 0.282 | 1.617 | 0.961 |
| ECL (720) | 0.229 | 0.318 | 0.240 | 0.322 | 0.233 | 0.317 | 0.241 | 0.323 | 0.530 | 0.585 | 0.715 | 0.685 | 1.203 | 0.898 | 0.369 | 0.426 | 0.223 | 0.321 | 1.647 | 0.975 |
| ECL (Avg) | 0.174 | 0.270 | 0.175 | 0.270 | 0.176 | 0.269 | 0.180 | 0.273 | 0.346 | 0.427 | 0.431 | 0.478 | 1.195 | 0.891 | 0.323 | 0.392 | 0.180 | 0.280 | 1.612 | 0.958 |
| Traffic (96) | 0.416 | 0.298 | 0.418 | 0.291 | 0.414 | 0.297 | 0.403 | 0.289 | 0.639 | 0.400 | 0.672 | 0.405 | 1.557 | 0.821 | 0.719 | 0.416 | 0.419 | 0.298 | 2.723 | 1.079 |
| Traffic (192) | 0.413 | 0.306 | 0.414 | 0.296 | 0.426 | 0.301 | 0.415 | 0.296 | 0.637 | 0.416 | 0.727 | 0.424 | 1.454 | 0.765 | 0.748 | 0.428 | 0.434 | 0.305 | 2.756 | 1.087 |
| Traffic (336) | 0.422 | 0.307 | 0.421 | 0.311 | 0.434 | 0.303 | 0.426 | 0.304 | 0.749 | 0.454 | 0.655 | 0.427 | 1.521 | 0.812 | 0.853 | 0.471 | 0.449 | 0.313 | 2.791 | 1.095 |
| Traffic (720) | 0.460 | 0.330 | 0.462 | 0.327 | 0.487 | 0.337 | 0.474 | 0.331 | 0.722 | 0.456 | 0.847 | 0.499 | 1.605 | 0.846 | 1.485 | 0.825 | 0.484 | 0.336 | 2.811 | 1.097 |
| Traffic (Avg) | 0.428 | 0.310 | 0.429 | 0.306 | 0.440 | 0.310 | 0.430 | 0.305 | 0.663 | 0.425 | 0.749 | 0.446 | 1.534 | 0.811 | 0.951 | 0.535 | 0.447 | 0.313 | 2.770 | 1.090 |

Table 8: Few-shot learning results on 5% training data with forecasting horizons $L_P \in \{96, 192, 336, 720\}$. The best and second best results are in bold and underlined, respectively. "-" means the lacking of data to build the training set in the 5% scenario.

| Models | ICTSP | | Time-LLM | | GPT4TS | | PatchTST | | FedFormer | | Autoformer | | Informer | | TimesNet | | DLinear | | Repeat | |
|---|---|---|---|---|---|---|---|---|---|---|---|---|---|---|---|---|---|---|---|---|
| Metric | MSE | MAE | MSE | MAE | MSE | MAE | MSE | MAE | MSE | MAE | MSE | MAE | MSE | MAE | MSE | MAE | MSE | MAE | MSE | MAE |
| ETTh1 (96) | 0.499 | 0.468 | 0.483 | 0.464 | 0.543 | 0.506 | 0.557 | 0.519 | 0.593 | 0.529 | 0.681 | 0.570 | 1.225 | 0.812 | 0.547 | 0.503 | 0.892 | 0.625 | 1.295 | 0.713 |
| ETTh1 (192) | 0.550 | 0.497 | 0.629 | 0.540 | 0.748 | 0.580 | 0.711 | 0.570 | 0.652 | 0.563 | 0.725 | 0.602 | 1.249 | 0.828 | 0.720 | 0.593 | 0.940 | 0.665 | 1.325 | 0.733 |
| ETTh1 (336) | 0.572 | 0.500 | 0.768 | 0.626 | 0.754 | 0.595 | 0.816 | 0.619 | 0.731 | 0.594 | 0.761 | 0.624 | 1.202 | 0.811 | 0.984 | 0.727 | 0.945 | 0.653 | 1.323 | 0.744 |
| ETTh1 (720) | - | - | - | - | - | - | - | - | - | - | - | - | - | - | - | - | - | - | - | - |
| ETTh1 (Avg) | 0.540 | 0.488 | 0.627 | 0.543 | 0.682 | 0.560 | 0.695 | 0.569 | 0.659 | 0.562 | 0.722 | 0.599 | 1.225 | 0.817 | 0.750 | 0.611 | 0.926 | 0.648 | 1.314 | 0.730 |
| ETTh2 (96) | 0.304 | 0.360 | 0.336 | 0.397 | 0.376 | 0.421 | 0.401 | 0.421 | 0.390 | 0.424 | 0.428 | 0.468 | 3.837 | 1.508 | 0.442 | 0.456 | 0.409 | 0.420 | 0.432 | 0.422 |
| ETTh2 (192) | 0.381 | 0.408 | 0.406 | 0.425 | 0.418 | 0.441 | 0.452 | 0.455 | 0.457 | 0.465 | 0.496 | 0.504 | 3.975 | 1.933 | 0.617 | 0.542 | 0.483 | 0.464 | 0.534 | 0.473 |
| ETTh2 (336) | 0.419 | 0.430 | 0.405 | 0.432 | 0.408 | 0.439 | 0.464 | 0.469 | 0.477 | 0.483 | 0.486 | 0.496 | 3.956 | 1.520 | 1.424 | 0.849 | 0.499 | 0.479 | 0.591 | 0.508 |
| ETTh2 (720) | - | - | - | - | - | - | - | - | - | - | - | - | - | - | - | - | - | - | - | - |
| ETTh2 (Avg) | 0.368 | 0.399 | 0.382 | 0.418 | 0.401 | 0.434 | 0.439 | 0.448 | 0.441 | 0.457 | 0.470 | 0.489 | 3.923 | 1.654 | 0.828 | 0.616 | 0.464 | 0.454 | 0.519 | 0.468 |
| ETTm1 (96) | 0.331 | 0.372 | 0.316 | 0.377 | 0.386 | 0.405 | 0.399 | 0.414 | 0.628 | 0.544 | 0.726 | 0.578 | 1.130 | 0.775 | 0.332 | 0.374 | 0.606 | 0.518 | 1.214 | 0.665 |
| ETTm1 (192) | 0.363 | 0.376 | 0.450 | 0.464 | 0.440 | 0.438 | 0.441 | 0.436 | 0.666 | 0.566 | 0.750 | 0.591 | 1.150 | 0.788 | 0.358 | 0.390 | 0.681 | 0.539 | 1.261 | 0.690 |
| ETTm1 (336) | 0.413 | 0.418 | 0.450 | 0.424 | 0.485 | 0.459 | 0.499 | 0.467 | 0.807 | 0.628 | 0.851 | 0.659 | 1.198 | 0.809 | 0.402 | 0.416 | 0.786 | 0.597 | 1.283 | 0.707 |
| ETTm1 (720) | 0.481 | 0.472 | 0.483 | 0.471 | 0.577 | 0.499 | 0.767 | 0.587 | 0.822 | 0.633 | 0.857 | 0.655 | 1.175 | 0.794 | 0.511 | 0.489 | 0.796 | 0.593 | 1.319 | 0.729 |
| ETTm1 (Avg) | 0.397 | 0.410 | 0.425 | 0.434 | 0.472 | 0.450 | 0.527 | 0.476 | 0.731 | 0.593 | 0.796 | 0.621 | 1.163 | 0.792 | 0.401 | 0.417 | 0.717 | 0.562 | 1.269 | 0.698 |
| ETTm2 (96) | 0.181 | 0.265 | 0.174 | 0.261 | 0.199 | 0.280 | 0.206 | 0.288 | 0.229 | 0.320 | 0.232 | 0.322 | 3.599 | 1.478 | 0.236 | 0.326 | 0.220 | 0.299 | 0.266 | 0.328 |
| ETTm2 (192) | 0.244 | 0.310 | 0.215 | 0.287 | 0.256 | 0.316 | 0.264 | 0.324 | 0.394 | 0.361 | 0.291 | 0.357 | 3.578 | 1.475 | 0.306 | 0.373 | 0.311 | 0.361 | 0.340 | 0.371 |
| ETTm2 (336) | 0.301 | 0.378 | 0.273 | 0.330 | 0.318 | 0.353 | 0.334 | 0.367 | 0.378 | 0.427 | 0.478 | 0.517 | 3.561 | 1.473 | 0.380 | 0.423 | 0.338 | 0.366 | 0.412 | 0.410 |
| ETTm2 (720) | 0.412 | 0.410 | 0.433 | 0.412 | 0.460 | 0.436 | 0.454 | 0.432 | 0.523 | 0.510 | 0.553 | 0.538 | 3.896 | 1.533 | 0.674 | 0.583 | 0.509 | 0.465 | 0.521 | 0.465 |
| ETTm2 (Avg) | 0.285 | 0.341 | 0.274 | 0.323 | 0.308 | 0.346 | 0.315 | 0.353 | 0.381 | 0.405 | 0.389 | 0.434 | 3.659 | 1.490 | 0.399 | 0.426 | 0.345 | 0.373 | 0.385 | 0.394 |
| Weather (96) | 0.170 | 0.225 | 0.172 | 0.263 | 0.175 | 0.230 | 0.171 | 0.224 | 0.229 | 0.309 | 0.227 | 0.299 | 0.497 | 0.497 | 0.184 | 0.242 | 0.207 | 0.253 | 0.259 | 0.254 |
| Weather (192) | 0.219 | 0.267 | 0.224 | 0.271 | 0.227 | 0.276 | 0.230 | 0.317 | 0.265 | 0.317 | 0.278 | 0.333 | 0.620 | 0.545 | 0.228 | 0.283 | 0.272 | 0.307 | 0.309 | 0.292 |
| Weather (336) | 0.278 | 0.313 | 0.282 | 0.321 | 0.286 | 0.322 | 0.294 | 0.326 | 0.353 | 0.392 | 0.351 | 0.393 | 0.649 | 0.547 | 0.279 | 0.322 | 0.313 | 0.328 | 0.377 | 0.338 |
| Weather (720) | 0.358 | 0.371 | 0.366 | 0.381 | 0.366 | 0.379 | 0.384 | 0.387 | 0.391 | 0.394 | 0.387 | 0.389 | 0.570 | 0.522 | 0.364 | 0.388 | 0.400 | 0.385 | 0.465 | 0.394 |
| Weather (Avg) | 0.256 | 0.294 | 0.261 | 0.309 | 0.264 | 0.302 | 0.270 | 0.304 | 0.310 | 0.353 | 0.311 | 0.354 | 0.584 | 0.528 | 0.264 | 0.309 | 0.298 | 0.318 | 0.353 | 0.320 |
| ECL (96) | 0.145 | 0.240 | 0.147 | 0.242 | 0.143 | 0.241 | 0.145 | 0.244 | 0.235 | 0.322 | 0.297 | 0.367 | 1.265 | 0.919 | 0.150 | 0.251 | 0.315 | 0.389 | 1.588 | 0.946 |
| ECL (192) | 0.158 | 0.246 | 0.158 | 0.241 | 0.159 | 0.255 | 0.163 | 0.260 | 0.247 | 0.341 | 0.308 | 0.375 | 1.298 | 0.939 | 0.163 | 0.263 | 0.318 | 0.396 | 1.595 | 0.950 |
| ECL (336) | 0.177 | 0.273 | 0.178 | 0.277 | 0.179 | 0.274 | 0.183 | 0.281 | 0.267 | 0.356 | 0.354 | 0.411 | 1.302 | 0.942 | 0.175 | 0.278 | 0.340 | 0.415 | 1.617 | 0.961 |
| ECL (720) | 0.221 | 0.315 | 0.224 | 0.312 | 0.233 | 0.323 | 0.233 | 0.323 | 0.318 | 0.394 | 0.426 | 0.466 | 1.259 | 0.919 | 0.219 | 0.311 | 0.635 | 0.613 | 1.647 | 0.975 |
| ECL (Avg) | 0.175 | 0.269 | 0.177 | 0.268 | 0.179 | 0.273 | 0.181 | 0.277 | 0.267 | 0.353 | 0.346 | 0.408 | 1.281 | 0.930 | 0.177 | 0.276 | 0.402 | 0.453 | 1.612 | 0.958 |
| Traffic (96) | 0.406 | 0.293 | 0.414 | 0.291 | 0.419 | 0.298 | 0.404 | 0.286 | 0.670 | 0.421 | 0.795 | 0.481 | 1.557 | 0.821 | 0.427 | 0.304 | 0.854 | 0.492 | 2.723 | 1.079 |
| Traffic (192) | 0.415 | 0.295 | 0.419 | 0.291 | 0.434 | 0.305 | 0.412 | 0.294 | 0.653 | 0.405 | 0.837 | 0.503 | 1.596 | 0.834 | 0.447 | 0.315 | 0.894 | 0.517 | 2.756 | 1.087 |
| Traffic (336) | 0.432 | 0.311 | 0.437 | 0.314 | 0.449 | 0.313 | 0.439 | 0.310 | 0.707 | 0.445 | 0.867 | 0.523 | 1.621 | 0.841 | 0.478 | 0.333 | 0.853 | 0.471 | 2.791 | 1.095 |
| Traffic (720) | - | - | - | - | - | - | - | - | - | - | - | - | - | - | - | - | - | - | - | - |
| Traffic (Avg) | 0.418 | 0.300 | 0.423 | 0.299 | 0.434 | 0.305 | 0.418 | 0.297 | 0.677 | 0.424 | 0.833 | 0.502 | 1.591 | 0.832 | 0.451 | 0.317 | 0.867 | 0.493 | 2.757 | 1.087 |

Table 9: Zero-shot tranfer learning results between ETT datasets with forecasting horizons $L_P \in \{96, 192, 336, 720\}$. The best and second best results are in bold and underlined, respectively.

| Models | ICTSP | | Time-LLM | | LLMTime | | GPT4TS | | PatchTST | | Autoformer | | TimesNet | | DLinear | | Repeat | |
|---|---|---|---|---|---|---|---|---|---|---|---|---|---|---|---|---|---|---|
| Metric | MSE | MAE | MSE | MAE | MSE | MAE | MSE | MAE | MSE | MAE | MSE | MAE | MSE | MAE | MSE | MAE | MSE | MAE |
| h1→h2 (96) | 0.272 | 0.335 | 0.279 | 0.337 | 0.510 | 0.576 | 0.335 | 0.374 | 0.304 | 0.350 | 0.469 | 0.486 | 0.358 | 0.387 | 0.347 | 0.400 | 0.432 | 0.422 |
| h1→h2 (192) | 0.330 | 0.373 | 0.351 | 0.374 | 0.523 | 0.586 | 0.412 | 0.417 | 0.386 | 0.400 | 0.634 | 0.567 | 0.427 | 0.429 | 0.447 | 0.460 | 0.534 | 0.473 |
| h1→h2 (336) | 0.359 | 0.402 | 0.388 | 0.415 | 0.640 | 0.637 | 0.441 | 0.444 | 0.414 | 0.428 | 0.655 | 0.588 | 0.449 | 0.451 | 0.515 | 0.505 | 0.591 | 0.508 |
| h1→h2 (720) | 0.387 | 0.431 | 0.391 | 0.420 | 2.296 | 1.034 | 0.438 | 0.452 | 0.419 | 0.443 | 0.570 | 0.549 | 0.448 | 0.458 | 0.665 | 0.589 | 0.588 | 0.517 |
| h1→h2 (Avg) | 0.337 | 0.385 | 0.352 | 0.387 | 0.992 | 0.708 | 0.407 | 0.422 | 0.381 | 0.405 | 0.582 | 0.548 | 0.421 | 0.431 | 0.494 | 0.489 | 0.536 | 0.480 |
| h2→h1 (96) | 0.411 | 0.426 | 0.450 | 0.452 | 1.130 | 0.777 | 0.732 | 0.577 | 0.485 | 0.465 | 0.693 | 0.569 | 0.848 | 0.601 | 0.689 | 0.555 | 1.295 | 0.713 |
| h2→h1 (192) | 0.435 | 0.447 | 0.465 | 0.461 | 1.242 | 0.820 | 0.758 | 0.559 | 0.565 | 0.509 | 0.760 | 0.601 | 0.860 | 0.610 | 0.707 | 0.568 | 1.325 | 0.733 |
| h2→h1 (336) | 0.448 | 0.461 | 0.501 | 0.482 | 1.328 | 0.864 | 0.759 | 0.578 | 0.581 | 0.515 | 0.781 | 0.619 | 0.867 | 0.626 | 0.710 | 0.577 | 1.323 | 0.744 |
| h2→h1 (720) | 0.469 | 0.481 | 0.501 | 0.502 | 4.145 | 1.461 | 0.781 | 0.597 | 0.628 | 0.561 | 0.796 | 0.644 | 0.887 | 0.648 | 0.704 | 0.596 | 1.339 | 0.756 |
| h2→h1 (Avg) | 0.441 | 0.454 | 0.479 | 0.474 | 1.961 | 0.981 | 0.758 | 0.578 | 0.565 | 0.513 | 0.758 | 0.608 | 0.866 | 0.621 | 0.703 | 0.574 | 1.321 | 0.737 |
| m1→h2 (96) | 0.299 | 0.359 | 0.321 | 0.369 | 0.510 | 0.576 | 0.353 | 0.392 | 0.354 | 0.385 | 0.435 | 0.470 | 0.377 | 0.407 | 0.365 | 0.415 | 0.432 | 0.422 |
| m1→h2 (192) | 0.361 | 0.394 | 0.389 | 0.410 | 0.523 | 0.586 | 0.443 | 0.437 | 0.447 | 0.434 | 0.495 | 0.489 | 0.471 | 0.453 | 0.454 | 0.462 | 0.534 | 0.473 |
| m1→h2 (336) | 0.376 | 0.412 | 0.408 | 0.433 | 0.640 | 0.637 | 0.469 | 0.461 | 0.481 | 0.463 | 0.470 | 0.472 | 0.472 | 0.484 | 0.496 | 0.494 | 0.591 | 0.508 |
| m1→h2 (720) | 0.388 | 0.431 | 0.406 | 0.436 | 2.296 | 1.034 | 0.466 | 0.468 | 0.474 | 0.471 | 0.480 | 0.485 | 0.495 | 0.482 | 0.541 | 0.529 | 0.588 | 0.517 |
| m1→h2 (Avg) | 0.356 | 0.399 | 0.381 | 0.412 | 0.992 | 0.708 | 0.433 | 0.440 | 0.439 | 0.438 | 0.470 | 0.479 | 0.454 | 0.457 | 0.464 | 0.475 | 0.536 | 0.480 |
| m1→m2 (96) | 0.168 | 0.255 | 0.169 | 0.257 | 0.646 | 0.563 | 0.217 | 0.294 | 0.195 | 0.271 | 0.385 | 0.457 | 0.222 | 0.295 | 0.221 | 0.314 | 0.266 | 0.328 |
| m1→m2 (192) | 0.223 | 0.293 | 0.227 | 0.318 | 0.934 | 0.654 | 0.277 | 0.327 | 0.258 | 0.311 | 0.433 | 0.469 | 0.288 | 0.337 | 0.286 | 0.359 | 0.340 | 0.371 |
| m1→m2 (336) | 0.273 | 0.329 | 0.290 | 0.338 | 1.157 | 0.728 | 0.331 | 0.360 | 0.317 | 0.348 | 0.476 | 0.477 | 0.341 | 0.367 | 0.357 | 0.406 | 0.412 | 0.410 |
| m1→m2 (720) | 0.355 | 0.384 | 0.375 | 0.367 | 4.730 | 1.531 | 0.429 | 0.413 | 0.416 | 0.404 | 0.582 | 0.535 | 0.436 | 0.418 | 0.476 | 0.476 | 0.521 | 0.465 |
| m1→m2 (Avg) | 0.255 | 0.315 | 0.265 | 0.320 | 1.867 | 0.869 | 0.314 | 0.349 | 0.297 | 0.334 | 0.469 | 0.485 | 0.322 | 0.354 | 0.335 | 0.389 | 0.385 | 0.394 |
| m2→h2 (96) | 0.288 | 0.351 | 0.298 | 0.356 | 0.510 | 0.576 | 0.360 | 0.401 | 0.327 | 0.367 | 0.353 | 0.393 | 0.360 | 0.401 | 0.333 | 0.391 | 0.432 | 0.422 |
| m2→h2 (192) | 0.352 | 0.395 | 0.359 | 0.397 | 0.523 | 0.586 | 0.434 | 0.437 | 0.411 | 0.418 | 0.432 | 0.437 | 0.434 | 0.437 | 0.441 | 0.456 | 0.534 | 0.473 |
| m2→h2 (336) | 0.371 | 0.415 | 0.367 | 0.412 | 0.640 | 0.637 | 0.460 | 0.459 | 0.439 | 0.447 | 0.452 | 0.459 | 0.460 | 0.459 | 0.505 | 0.503 | 0.591 | 0.508 |
| m2→h2 (720) | 0.398 | 0.445 | 0.393 | 0.434 | 2.296 | 1.034 | 0.485 | 0.477 | 0.459 | 0.470 | 0.453 | 0.467 | 0.485 | 0.477 | 0.543 | 0.534 | 0.588 | 0.517 |
| m2→h2 (Avg) | 0.352 | 0.402 | 0.354 | 0.400 | 0.992 | 0.708 | 0.435 | 0.444 | 0.409 | 0.426 | 0.423 | 0.439 | 0.435 | 0.444 | 0.456 | 0.471 | 0.536 | 0.480 |
| m2→m1 (96) | 0.371 | 0.395 | 0.359 | 0.397 | 1.179 | 0.781 | 0.747 | 0.558 | 0.491 | 0.437 | 0.735 | 0.576 | 0.747 | 0.558 | 0.570 | 0.490 | 1.214 | 0.665 |
| m2→m1 (192) | 0.387 | 0.411 | 0.390 | 0.420 | 1.327 | 0.846 | 0.781 | 0.560 | 0.530 | 0.470 | 0.753 | 0.586 | 0.781 | 0.560 | 0.590 | 0.506 | 1.261 | 0.690 |
| m2→m1 (336) | 0.416 | 0.427 | 0.421 | 0.445 | 1.478 | 0.902 | 0.778 | 0.578 | 0.565 | 0.497 | 0.750 | 0.593 | 0.778 | 0.578 | 0.706 | 0.567 | 1.283 | 0.707 |
| m2→m1 (720) | 0.456 | 0.444 | 0.487 | 0.488 | 3.749 | 1.408 | 0.769 | 0.573 | 0.686 | 0.565 | 0.782 | 0.609 | 0.769 | 0.573 | 0.731 | 0.584 | 1.319 | 0.729 |
| m2→m1 (Avg) | 0.408 | 0.419 | 0.414 | 0.438 | 1.933 | 0.984 | 0.769 | 0.567 | 0.568 | 0.492 | 0.755 | 0.591 | 0.769 | 0.567 | 0.649 | 0.537 | 1.269 | 0.698 |

## A.3 IMPLEMENTATION DETAILS

### A.3.1 MORE DETAILS OF MODEL IMPLEMENTATION

**Token Construction** Figure 6 provides another illustration of how to construct the input tokens in ICTSP. In the implementation, we first concatenate the output embedding to the input series. Then we cut the last $L_b + L_P$ part as the target tokens. We perform stepwise sampling from the end part to the initial part of $X_I$. If the remaining initial part of $X_I$ cannot constitute a token with the required length of $L_b + L_P$, we drop these initial steps in $X_I$.

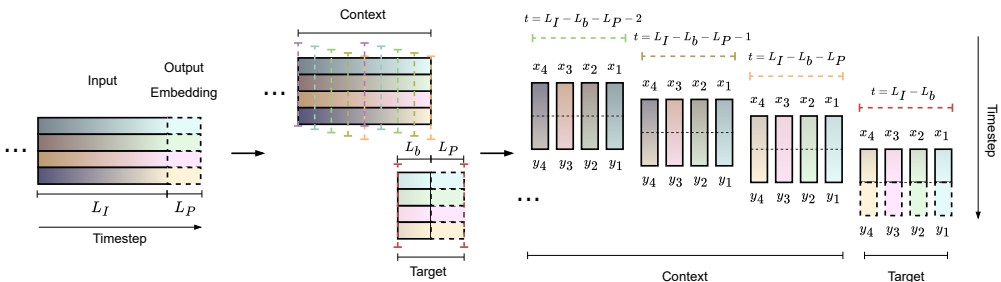

Figure 6: Another illustration of the process of sampling the context examples from the input time series in ICTSP.

**Trainable Embeddings** As illustrated in Equation 5, we use two types of trainable positional embeddings in the token construction. One embedding represents the series source of a token, and the other represents the position of the token within the entire input. All the positional embeddings have the same $d$ dimension and are added to the latent tokens.

**Randomized Training** During training, we can introduce some random shifting $r \in \{0, \cdots, m-1\}$ to the sampling start point to expose the model to varied context examples. To prevent the positional embedding from learning information with specific time series channel, we randomly shuffle the order of series in the input to make the series show up in different position each training step. This

can slightly enhance the performance in the few-shot and zero-shot tasks and for small datasets. Since ICTSP does not restrict the number of series in each input, we can randomly pick subsets of input series to build the tokens during training. This random selection slightly improves performance in few-shot learning with small datasets that have weak dependencies, while not significantly affecting performance on larger datasets with stable dependencies.

**Token Retrieval** We can reduce the computational costs of ICTSP by sampling one context token every $m$ steps from the total $N$ tokens for each series. For very large token amounts, to avoid losing information with large sampling steps, we use token retrieval (TR) to further reduce token size. Below, we provide a detailed description of the TR process.

Let $\mathbf{z}_i$ be the $i$-th token in the context. Each token $\mathbf{z}_i$ is reduced to a $\delta$-dimensional latent vector $\tilde{\mathbf{z}}_i$ via a linear layer:

$$\tilde{\mathbf{z}}_i = \mathbf{W}\mathbf{z}_i + \mathbf{b} \tag{6}$$

where $\mathbf{W} \in \mathbb{R}^{\delta \times d}$ and $\mathbf{b} \in \mathbb{R}^\delta$ are the weights and biases of the linear layer. We calculate the cosine similarity between the $\tilde{\mathbf{z}}_i$ of each context token and each target token $\tilde{\mathbf{z}}_c$:

$$\texttt{cosine\_similarity}(\tilde{\mathbf{z}}_i, \tilde{\mathbf{z}}_c) = \frac{\tilde{\mathbf{z}}_i \cdot \tilde{\mathbf{z}}_c}{\|\tilde{\mathbf{z}}_i\|\|\tilde{\mathbf{z}}_c\|} \tag{7}$$

Next, we average the cosine similarity results for each context token:

$$\texttt{average\_similarity}(\mathbf{z}_i) = \frac{1}{C}\sum_{c=1}^{C} \texttt{cosine\_similarity}(\tilde{\mathbf{z}}_i, \tilde{\mathbf{z}}_c) \tag{8}$$

where $C$ is the number of target tokens. We then rank the context tokens by their average similarity and select the top $q\%$ as context examples.

The remaining tokens are merged into $r$ tokens by grouped weighted averaging. Let $k = \lfloor qN \rfloor$ be the number of selected tokens, and let $p = N - k$ be the number of remaining tokens. The $r$ merged tokens are obtained as follows:

$$\mathbf{z}_i^{\text{merged}} = \sum_{j \in \mathcal{G}_i} \alpha_j \mathbf{z}_j \tag{9}$$

where $\mathcal{G}_i$ is the set of tokens in group $i$, determined by the ranked order, and $\alpha_j$ are the weights obtained from a softmax over the cosine similarities:

$$\alpha_j = \frac{e^{\texttt{cosine\_similarity}(\tilde{\mathbf{z}}_j, \tilde{\mathbf{z}}_{j'})}}{\sum_{k \in \mathcal{G}_i} e^{\texttt{cosine\_similarity}(\tilde{\mathbf{z}}_k, \tilde{\mathbf{z}}_{j'})}} \tag{10}$$

The final set of context tokens includes the selected $k$ tokens and the $r$ merged tokens:

$$\mathbf{Z}^{\text{final}} = \{\mathbf{z}_1, \dots, \mathbf{z}_k, \mathbf{z}_1^{\text{merged}}, \dots, \mathbf{z}_r^{\text{merged}}\} \tag{11}$$

This results in $\lfloor q(L_I - L_b - L_P)/m \rfloor C + r$ context tokens.

**Token Rationalization and Linear Layer Warm-up** ICTSP excels at handling datasets with varied characteristics and performing transfer learning across different datasets. The robustness of ICTSP, when facing small and noisy datasets, is further enhanced with following training strategies. For each context or target token $\mathbf{z}_i$, we can subtract the last value in its lookback from all values to align forecasting tasks across all tokens. This reduces the impact of mean shifting within or across series, serving as a naive repeat method baseline. Additionally, as mentioned earlier, the direct shortcut in Equation 1 helps ICTSP achieve a simple linear baseline in weak dependency scenarios. To prevent the attention and FFN terms from learning noise patterns too fast, we can initially train only the shortcut: for the first $W$ steps, remove the transformer layers and train only the input and output projections on the target tokens. After this warm-up, reintroduce the transformer blocks with context examples, gradually increasing the model's complexity to adapt to small and noisy data. Figure 7 shows three cases: i) not using linear layer warm-up ($W = 0$), ii) using linear layer warm-up ($W = 5000$), and iii) only using the linear layer ($W = \infty$). We trained the ICTSP with these settings on ECL ($L_P = 192$) with reduced input ($L_I = 512, L_b = 256$). The model with initialized weights for the input and output projection shortcut outperforms the models without warm-up and with only the linear layer as a baseline comparison. The linear layer warm-up stage structures the embedding space into a forecasting task, guiding the model to build tokens that focus on temporal forecasting power. The effectiveness of this warm-up stage is due to the token design and the shortcuts in the model structure.

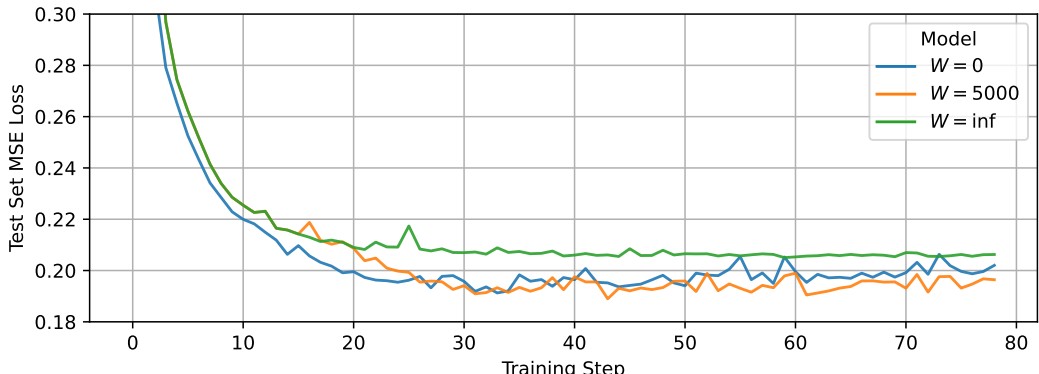

Figure 7: The effects of using linear layer warm-up to initialize the embedding space as forecasting tasks with the input and output projection shortcut. Using a warm-up step $W = 5000$ helps the model structure the embedding space as forecasting tasks before fitting the entire transformer structure, gaining performance margin and preventing overfitting. These three model are trained on ECL ($L_P = 192$) with reduced ICTSP settings ($L_I = 512, L_b = 256$).

### A.3.2 HYPERPARAMETER SETTINGS AND TRAINING DETAILS

**Model Training Setup** The ICTSP model is trained using the Adam optimizer and MSE loss in Pytorch, with a learning rate of 0.0005 each dataset. We test the model every 200 training steps with a early-stopping patience being $30 \times 200$ steps. The first 1000 steps are for learning rate warm-up, followed by a linear decay of learning rate. We set the random seed as 2024. Our models are trained on single Nvidia RTX 4090 GPU with a batch size equals to 32 for most of the datasets. We decrease the batch size to 16 and 8 for the larger dataset ECL and Traffic, respectively. In the full-data experiment setting, we split each dataset with 70% training set, 10% validation, set and 20% test set. We fit a standardization scaler on the training set and apply it to the whole dataset. This setting is applied in consistent with previous studies like (Wu et al., 2021; 2022; Nie et al., 2022; Jin et al., 2024).

**Hyper-parameters of ICTSP** For the hyper-parameters of ICTSP, we use $L_I = 1440$ and $L_b = 512$ for all the experiments. We use the pre-normalization Transformer with $K = 3$ layers, $d = 128$ latent dimension, and 8 heads. The hidden dimension of FFN is set to 4 times of latent dimension $d$ and the dropout rate of Transformer layers is set to 0.5. We use $m = 8$ as the sampling step and $q = 10\%, r = 30$ for the token retrieval module.

**Randomness of Training** We test the randomness of training the ICTSP by alternating the random seed among {2022, 2023, 2024, 2025, 2026} and conducting multiple training runs on the same dataset settings. The error bars of the results with the five random seeds on the Weather and ECL datasets are shown in Figure 8, alongside the results of the two best previous baselines, Time-LLM (Jin et al., 2024) and PatchTST (Nie et al., 2022). The results show that ICTSP can stably surpass previous methods, even with the worst results among the five random seeds, indicating the robustness of ICTSP in performance improvement.

### A.3.3 DISCUSSION OF FAIR COMPARISON

Our experimental environment aligns with the codes used by previous studies, including (Wu et al., 2021; Zeng et al., 2023; Wu et al., 2022; Nie et al., 2022; Zhou et al., 2023; Jin et al., 2024). Our few-shot and zero-shot learning setups follow (Zhou et al., 2023; Jin et al., 2024).

In the full-data experiment, for the input length $L_I$ setting, previous studies like (Nie et al., 2022; Zhou et al., 2023) suggest that $L_I$ should be considered a hyper-parameter that needs tuning for each dataset. Therefore, we chose to directly source their results where applicable, to avoid underestimating their performance when rerunning with a fixed $L_I$.

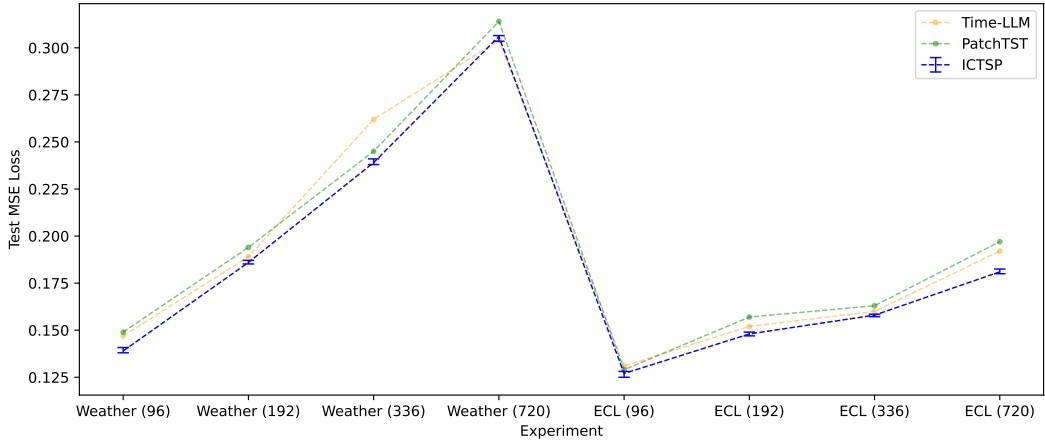

Figure 8: Randomness in Training ICTSP. The error bars for training ICTSP on the Weather and ECL datasets with random seeds {2022, 2023, 2024, 2025, 2026} are illustrated, compared with the previous best two baselines, Time-LLM and PatchTST. The results show that ICTSP can stably surpass previous baselines even in the worst training cases, demonstrating the robustness of the ICTSP structure.

In the few-shot experiment, careful consideration of $L_I$ is very crucial. Since (Zhou et al., 2023; Jin et al., 2024) use the first 10% or 5% of the training set for few-shot training data, this represents a scenario where the model's forecasting time is far from the training data collection. Thus, we should avoid leaking too much recent data to the model, especially for ICTSP. Because ICTSP can use historical input as in-context training samples, increasing $L_I$ too much might leak data that should be invisible to the model, defeating the purpose of the few-shot experiment.

Given the difference between $L_I = 512$ used by (Zhou et al., 2023; Jin et al., 2024) and $L_I = 1440, L_b = 512$ for ICTSP, we chose to mask the part of the test set input data not visible to the $L_I = 512$ setting in (Zhou et al., 2023; Jin et al., 2024) with zero-filling for ICTSP. This ensures that no unseen data for (Zhou et al., 2023; Jin et al., 2024) is used by ICTSP. As a result, there are no context examples for ICTSP (which reduces it to a Series-wise Transformer) at the very beginning of the test set input data, and the number of context examples gradually increases in the subsequent test set input data points. Thus, the few-shot learning results in Tables 2, 7, and 8 can be regarded as results from a weakened version of ICTSP, yet still achieving SOTA performance.

## A.4  ADDITIONAL ANALYSIS OF TSF TRANSFORMERS

### A.4.1  DIRECT SHORTCUT IN TEMPORAL-WISE TRANSFORMERS

Revisiting the expression of Temporal-wise Transformer in Equation 3, it is clear that if context tokens become ineffective, the remaining MLPs or linear predictors are essentially guessing channel values and structures based solely on target positional embeddings. Past studies often found such Transformers to produce output resembling a flat line of input means rather than learning dynamic patterns, which could be explained by a failure of context and ineffective learning from positional embeddings.

### A.4.2  MORE DISCUSSION OF INTER-SERIES MODELING ABILITY

From the ICL perspective, the Series-wise Transformer primarily learns temporal dependencies rather than inter-series dependencies. This is evident in the "Multi" dataset with simple shifting inter-series dependencies, as shown in Figure 4 (b). This may indicate a lack of representation ability in the series token construction. Transformers can easily transfer information within the same token vector channel between different tokens via multi-head attention. When building the token example pool for ICTSP, a shifting process inherently exposes inter-series time-interleaved dependencies directly in the token representations, making it easy to capture these dependencies.

However, if these dependencies can be directly modeled into a single token dimension rather than between tokens, the Series-wise Transformer with enlarged tokens may still acquire inter-series learning ability alongside its inherent temporal learning ability. In a scenario with a very large latent dimension for the input token, where the token vector encompasses every shifting possibility of the input series, multi-head attention can find the alignment of possible inter-series time-interleaved dependencies between tokens by exploring this high dimension and fixing these alignments into the learned token embedding. This analysis suggests that increasing the token dimension of the Series-wise Transformer may enhance its inter-series modeling capability on specific datasets. Nonetheless, since these inter-series dependencies are learned and fixed in the token representation space, this capability cannot be transferred to other datasets in few-shot or zero-shot scenarios, unlike ICTSP.

## A.5 COMPARISON SUMMARY OF DIFFERENT TRANSFORMER-BASED TSF STRUCTURES

Table 10: Summary of the characteristics of Transformer-based TSF structures

| Methods | Temporal-wise Transformer | Temporal-wise Transformer (with Channel Independence) | Temporal-wise Transformer (with Patching) | Series-wise Transformer | ICTSP |
|---|---|---|---|---|---|
| **Tokenization** | Timestep embedding | Scalar embedding | Patch embedding | Series embedding | Embedding of forecasting tasks |
| **ICL (x, y) Pair** | (Timestep index, Multi-channel values) | (Timestep index, Single-channel value) | (Timestep index, Multi-channel values in a temporal interval) | (Input series, Output series) | (Input series, Output series) |
| **Token Representation** for | Single-step inter-series dependencies | Scalar value at a single timestep | Temporal-wise and series-wise relationships in a local interval | Temporal-wise relationships within a single series input | Temporal-wise relationships in a forecasting task |
| **Attention to Learn** | Temporal-wise dynamics of series dependencies | Temporal-wise dependencies | Temporal-wise relationships between local intervals | Aligned inter-series relationships | Forecasting task predictors, considering both aligned and unaligned inter-series relationships in shifted context examples |
| **Suited For** | Strong and consistent inter-series dependencies | Strong and fast-changing temporal dependencies | Temporal and series relationships can be automatically balanced by this structure | Real-world datasets with strong temporal-wise and aligned series-wise relationships (e.g., traffic) | Datasets with fast-changing temporal relationships, aligned or unaligned series relationships; transfering to new datasets with different channel structures |
| **Not Suited For** | Weak inter-series dependencies in real-world datasets | Strong inter-series dependencies | Few-shot/zero-shot cases with different series relationship characteristics | Unaligned series-wise relationships, such as shifting effects between series | / |
| **ICL Context Examples Describing** | Underlying inter-series effects vary with the time indices in the context lookback (likely nonexistent or weak) | Temporal relationships within the context for one series | Temporal-wise relationships between local intervals | No context examples | Historical ground truth forecasting tasks as references |
| **Channel Structure** | Fixed | Fixed | Fixed | Flexible but requires additional training to adapt to datasets with new channel structure | Flexible; additional training is not necessary since context examples exist |
| **Problem** | Only series relationships in tokenization; risks overfitting to series-wise effects from the input context | Unable to model series-wise relationships | Fixed channel structure in tokenization prevents generalization to datasets with different structures | Restricted to specific dataset channel structures; unable to learn unaligned series-wise effects | / |

## A.6 ADDITIONAL ABLATION STUDY: ICTSP WITH LIMITED (MASKED) LOOKBACK INFORMATION

To fully align with the $L_I = 512$ used in the baseline models, we relax the design of ICTSP, which originally required all context tokens to be ground truth for forecasting tasks. We allow some forecasting tasks to contain incomplete information. In this case, we still use $L_I = 1440$, but mask all information from the first $1440 - 512 = 928$ timesteps in the input, ensuring that only the last 512-step true information remains in the input. For the masked portion of the first 928 steps, we fill it with the mean value of each series token from the last 512 steps.

To maintain the presence of some true forecasting task tokens for shorter $L_P$, we shorten $L_b$ to 336. All other hyperparameter settings remain unchanged. We denote this structure as ICTSP (Limited). We repeated the experiments on the ETTm2 and Weather datasets, including full data, few-shot 10% (FS10%), few-shot 5% (FS05%), and zero-shot (ZS) experiments on the ETT datasets. The results are shown in the Table A.6.

Table 11: Performance comparison of ICTSP (Limited) with six baselines on ETTm2 and Weather datasets across full data, few-shot (FS10% and FS5%), and zero-shot (ZS) settings.

| Methods | ICTSP | | ICTSP (Limited) | | Time-LLM | | GPT4TS | | PatchTST | | TimesNet | | DLinear | |
|---|---|---|---|---|---|---|---|---|---|---|---|---|---|---|
| Metric | MSE | MAE | MSE | MAE | MSE | MAE | MSE | MAE | MSE | MAE | MSE | MAE | MSE | MAE |
| Full ETTm2 (96) | 0.159 | 0.248 | 0.160 | 0.241 | 0.161 | 0.253 | 0.173 | 0.262 | 0.165 | 0.255 | 0.187 | 0.267 | 0.167 | 0.269 |
| Full ETTm2 (192) | 0.212 | 0.289 | 0.216 | 0.290 | 0.219 | 0.293 | 0.229 | 0.301 | 0.220 | 0.292 | 0.249 | 0.309 | 0.224 | 0.303 |
| Full ETTm2 (336) | 0.268 | 0.326 | 0.268 | 0.326 | 0.271 | 0.329 | 0.286 | 0.341 | 0.274 | 0.329 | 0.321 | 0.351 | 0.281 | 0.342 |
| Full ETTm2 (720) | 0.347 | 0.382 | 0.354 | 0.383 | 0.352 | 0.379 | 0.378 | 0.401 | 0.362 | 0.385 | 0.408 | 0.403 | 0.397 | 0.421 |
| Full ETTm2 (Avg) | 0.247 | 0.311 | 0.250 | 0.310 | 0.251 | 0.314 | 0.267 | 0.326 | 0.255 | 0.315 | 0.291 | 0.333 | 0.267 | 0.334 |
| Full Weather (96) | 0.139 | 0.194 | 0.142 | 0.198 | 0.147 | 0.201 | 0.162 | 0.212 | 0.149 | 0.198 | 0.172 | 0.220 | 0.176 | 0.237 |
| Full Weather (192) | 0.186 | 0.236 | 0.187 | 0.237 | 0.189 | 0.234 | 0.204 | 0.248 | 0.194 | 0.241 | 0.219 | 0.261 | 0.220 | 0.282 |
| Full Weather (336) | 0.239 | 0.279 | 0.242 | 0.281 | 0.262 | 0.279 | 0.254 | 0.286 | 0.245 | 0.282 | 0.280 | 0.306 | 0.265 | 0.319 |
| Full Weather (720) | 0.306 | 0.320 | 0.308 | 0.322 | 0.304 | 0.316 | 0.326 | 0.337 | 0.314 | 0.334 | 0.365 | 0.359 | 0.333 | 0.362 |
| Full Weather (Avg) | 0.218 | 0.257 | 0.220 | 0.260 | 0.226 | 0.258 | 0.237 | 0.271 | 0.226 | 0.264 | 0.259 | 0.287 | 0.249 | 0.300 |
| FS10% ETTm2 (96) | 0.176 | 0.258 | 0.176 | 0.263 | 0.177 | 0.261 | 0.188 | 0.269 | 0.191 | 0.274 | 0.212 | 0.285 | 0.213 | 0.303 |
| FS10% ETTm2 (192) | 0.239 | 0.307 | 0.238 | 0.308 | 0.241 | 0.314 | 0.251 | 0.309 | 0.252 | 0.317 | 0.270 | 0.323 | 0.278 | 0.345 |
| FS10% ETTm2 (336) | 0.288 | 0.336 | 0.286 | 0.334 | 0.274 | 0.327 | 0.307 | 0.346 | 0.306 | 0.353 | 0.323 | 0.353 | 0.338 | 0.385 |
| FS10% ETTm2 (720) | 0.395 | 0.391 | 0.394 | 0.403 | 0.417 | 0.390 | 0.426 | 0.417 | 0.433 | 0.427 | 0.474 | 0.449 | 0.436 | 0.440 |
| FS10% ETTm2 (Avg) | 0.275 | 0.323 | 0.274 | 0.327 | 0.277 | 0.323 | 0.293 | 0.335 | 0.296 | 0.343 | 0.320 | 0.353 | 0.316 | 0.368 |
| FS10% Weather (96) | 0.164 | 0.214 | 0.165 | 0.212 | 0.161 | 0.210 | 0.163 | 0.215 | 0.165 | 0.215 | 0.184 | 0.230 | 0.171 | 0.224 |
| FS10% Weather (192) | 0.209 | 0.252 | 0.208 | 0.253 | 0.204 | 0.248 | 0.210 | 0.254 | 0.210 | 0.257 | 0.245 | 0.283 | 0.215 | 0.263 |
| FS10% Weather (336) | 0.259 | 0.294 | 0.261 | 0.295 | 0.261 | 0.302 | 0.256 | 0.292 | 0.259 | 0.297 | 0.305 | 0.321 | 0.258 | 0.299 |
| FS10% Weather (720) | 0.315 | 0.333 | 0.319 | 0.336 | 0.309 | 0.332 | 0.321 | 0.339 | 0.332 | 0.346 | 0.381 | 0.371 | 0.320 | 0.346 |
| FS10% Weather (Avg) | 0.237 | 0.273 | 0.238 | 0.274 | 0.234 | 0.273 | 0.238 | 0.275 | 0.242 | 0.279 | 0.279 | 0.301 | 0.241 | 0.283 |
| FS05% ETTm2 (96) | 0.181 | 0.265 | 0.179 | 0.264 | 0.174 | 0.261 | 0.199 | 0.280 | 0.206 | 0.288 | 0.236 | 0.326 | 0.220 | 0.299 |
| FS05% ETTm2 (192) | 0.244 | 0.310 | 0.241 | 0.312 | 0.215 | 0.287 | 0.256 | 0.316 | 0.264 | 0.324 | 0.306 | 0.373 | 0.311 | 0.361 |
| FS05% ETTm2 (336) | 0.301 | 0.378 | 0.299 | 0.341 | 0.273 | 0.330 | 0.318 | 0.353 | 0.334 | 0.367 | 0.380 | 0.423 | 0.338 | 0.366 |
| FS05% ETTm2 (720) | 0.412 | 0.410 | 0.405 | 0.408 | 0.433 | 0.412 | 0.460 | 0.436 | 0.454 | 0.432 | 0.674 | 0.583 | 0.509 | 0.465 |
| FS05% ETTm2 (Avg) | 0.285 | 0.341 | 0.281 | 0.331 | 0.274 | 0.323 | 0.308 | 0.346 | 0.315 | 0.353 | 0.399 | 0.426 | 0.345 | 0.373 |
| FS05% Weather (96) | 0.170 | 0.225 | 0.170 | 0.223 | 0.172 | 0.263 | 0.175 | 0.230 | 0.171 | 0.224 | 0.184 | 0.242 | 0.207 | 0.253 |
| FS05% Weather (192) | 0.219 | 0.267 | 0.223 | 0.269 | 0.224 | 0.271 | 0.227 | 0.276 | 0.230 | 0.277 | 0.228 | 0.283 | 0.272 | 0.307 |
| FS05% Weather (336) | 0.278 | 0.313 | 0.277 | 0.310 | 0.282 | 0.321 | 0.286 | 0.322 | 0.294 | 0.326 | 0.279 | 0.322 | 0.313 | 0.328 |
| FS05% Weather (720) | 0.358 | 0.371 | 0.355 | 0.368 | 0.366 | 0.381 | 0.366 | 0.379 | 0.384 | 0.387 | 0.364 | 0.388 | 0.400 | 0.385 |
| FS05% Weather (Avg) | 0.256 | 0.294 | 0.256 | 0.293 | 0.261 | 0.309 | 0.264 | 0.302 | 0.270 | 0.304 | 0.264 | 0.309 | 0.298 | 0.318 |
| m1→m2 (96) | 0.168 | 0.255 | 0.169 | 0.261 | 0.169 | 0.257 | 0.217 | 0.294 | 0.195 | 0.271 | 0.222 | 0.295 | 0.221 | 0.314 |
| m1→m2 (192) | 0.223 | 0.293 | 0.223 | 0.295 | 0.227 | 0.318 | 0.277 | 0.327 | 0.258 | 0.311 | 0.288 | 0.337 | 0.286 | 0.359 |
| m1→m2 (336) | 0.273 | 0.329 | 0.276 | 0.335 | 0.290 | 0.338 | 0.331 | 0.360 | 0.317 | 0.348 | 0.341 | 0.367 | 0.357 | 0.406 |
| m1→m2 (720) | 0.355 | 0.384 | 0.362 | 0.381 | 0.375 | 0.367 | 0.429 | 0.413 | 0.416 | 0.404 | 0.436 | 0.418 | 0.476 | 0.476 |
| m1→m2 (Avg) | 0.255 | 0.315 | 0.258 | 0.318 | 0.265 | 0.320 | 0.314 | 0.349 | 0.297 | 0.334 | 0.322 | 0.354 | 0.335 | 0.389 |
| h1→h2 (96) | 0.272 | 0.335 | 0.274 | 0.336 | 0.279 | 0.337 | 0.335 | 0.374 | 0.304 | 0.350 | 0.358 | 0.387 | 0.347 | 0.400 |
| h1→h2 (192) | 0.330 | 0.373 | 0.332 | 0.375 | 0.351 | 0.374 | 0.412 | 0.417 | 0.386 | 0.400 | 0.427 | 0.429 | 0.447 | 0.460 |
| h1→h2 (336) | 0.359 | 0.402 | 0.361 | 0.406 | 0.388 | 0.415 | 0.441 | 0.444 | 0.414 | 0.428 | 0.449 | 0.451 | 0.515 | 0.505 |
| h1→h2 (720) | 0.387 | 0.431 | 0.388 | 0.437 | 0.391 | 0.420 | 0.438 | 0.452 | 0.419 | 0.443 | 0.448 | 0.458 | 0.665 | 0.589 |
| h1→h2 (Avg) | 0.337 | 0.385 | 0.339 | 0.389 | 0.352 | 0.387 | 0.407 | 0.422 | 0.381 | 0.405 | 0.421 | 0.431 | 0.494 | 0.489 |

From the average ranking, it can be observed that even with masking applied to information prior to the 512-step window, the limited ICTSP still significantly outperforms all baselines. In the full data and zero-shot experiments, limited ICTSP performs slightly weaker than the original ICTSP but remains superior to all previous baselines. However, in the few-shot training experiments, limited ICTSP surpasses the original ICTSP to some extent. It is important to note that this improvement is due to an additional constraint in the original ICTSP, which, despite using $L_I = 1440$, imposes zero-filling restrictions outside of $L_b = 512$ to ensure fair comparison with baselines using $L_I = 512$ (see Section A.3.3 Discussion of fair comparison for details on few-shot experiments).

In the full data and zero-shot experiments, all data can be used with any $L_I$ without causing data leakage. However, in the few-shot scenario, due to the skipping nature of data sampling, a longer $L_I$ accesses more lookback data (in val/test set), creating an unfair advantage over models with shorter $L_I$. To eliminate this effect, in the original method, the input to the original ICTSP for few-shot experiments was subjected to zero-masking, ensuring that it could access at most 512 timesteps of data before the start of the validation and test sets, avoiding any unfair data usage. This means that in the first few datapoints, the original ICTSP could only see 512 timesteps of valid data, gradually expanding to $L_I = 1440$ as valid data became available.

However, since the model applies a last-value demeaning operation to the forecasting task tokens, the initial zero-filling leads to meaningless negative last values being filled in those positions. This effectively suppresses the model's performance. In ICTSP (Limited), we address this issue by using mean-value filling, ensuring that the filled values in the forecasting task tokens are more reasonable. This adjustment allows ICTSP (Limited) to outperform the original ICTSP in few-shot experiments.

To clarify this, we replaced the zero-filling used for fair comparison in original ICTSP with mean-filling and present the results below. The results show that ICTSP with mean-filling for fair comparison significantly outperforms all models, including Time-LLM, in all experiments. This demonstrates that ICTSP's few-shot capability is inherently stronger than the zero-filling results presented in the original paper when proper mean-filling strategies are used to handle unknown values in forecasting tasks. This result further enhances the significance of ICTSP. The results can be found in Table A.6.

Table 12: Results of using mean-filling strategy for the 512-step data retaining of fair comparison in the few-shot experiments.

| Methods | ICTSP (Zero-filling) | | ICTSP (Mean-filling) | | ICTSP (Limited) | | Time-LLM | | GPT4TS | | PatchTST | | TimesNet | | DLinear | |
|---|---|---|---|---|---|---|---|---|---|---|---|---|---|---|---|---|
| Metric | MSE | MAE | MSE | MAE | MSE | MAE | MSE | MAE | MSE | MAE | MSE | MAE | MSE | MAE | MSE | MAE |
| FS10% ETTm2 (96) | 0.176 | 0.258 | 0.173 | 0.259 | 0.176 | 0.263 | 0.177 | 0.261 | 0.188 | 0.269 | 0.191 | 0.274 | 0.212 | 0.285 | 0.213 | 0.303 |
| FS10% ETTm2 (192) | 0.239 | 0.307 | 0.226 | 0.298 | 0.238 | 0.308 | 0.241 | 0.314 | 0.251 | 0.309 | 0.252 | 0.317 | 0.270 | 0.323 | 0.278 | 0.345 |
| FS10% ETTm2 (336) | 0.288 | 0.336 | 0.276 | 0.329 | 0.286 | 0.334 | 0.274 | 0.327 | 0.307 | 0.346 | 0.306 | 0.353 | 0.323 | 0.353 | 0.338 | 0.385 |
| FS10% ETTm2 (720) | 0.395 | 0.391 | 0.381 | 0.388 | 0.394 | 0.403 | 0.417 | 0.390 | 0.426 | 0.417 | 0.433 | 0.427 | 0.474 | 0.449 | 0.436 | 0.440 |
| FS10% ETTm2 (Avg) | 0.275 | 0.323 | 0.264 | 0.319 | 0.274 | 0.327 | 0.277 | 0.323 | 0.293 | 0.335 | 0.296 | 0.343 | 0.320 | 0.353 | 0.316 | 0.368 |
| FS10% Weather (96) | 0.164 | 0.214 | 0.164 | 0.214 | 0.165 | 0.212 | 0.161 | 0.210 | 0.163 | 0.215 | 0.165 | 0.215 | 0.184 | 0.230 | 0.171 | 0.224 |
| FS10% Weather (192) | 0.209 | 0.252 | 0.204 | 0.250 | 0.208 | 0.253 | 0.204 | 0.248 | 0.210 | 0.254 | 0.210 | 0.257 | 0.245 | 0.283 | 0.215 | 0.263 |
| FS10% Weather (336) | 0.259 | 0.294 | 0.259 | 0.293 | 0.261 | 0.295 | 0.261 | 0.302 | 0.256 | 0.292 | 0.259 | 0.297 | 0.305 | 0.321 | 0.258 | 0.299 |
| FS10% Weather (720) | 0.315 | 0.333 | 0.310 | 0.332 | 0.319 | 0.336 | 0.309 | 0.332 | 0.321 | 0.339 | 0.332 | 0.346 | 0.381 | 0.371 | 0.320 | 0.346 |
| FS10% Weather (Avg) | 0.237 | 0.273 | 0.234 | 0.272 | 0.238 | 0.274 | 0.234 | 0.273 | 0.238 | 0.275 | 0.242 | 0.279 | 0.279 | 0.301 | 0.241 | 0.283 |
| FS05% ETTm2 (96) | 0.181 | 0.265 | 0.177 | 0.263 | 0.179 | 0.264 | 0.174 | 0.261 | 0.199 | 0.280 | 0.206 | 0.288 | 0.236 | 0.326 | 0.220 | 0.299 |
| FS05% ETTm2 (192) | 0.244 | 0.310 | 0.234 | 0.306 | 0.241 | 0.312 | 0.215 | 0.287 | 0.256 | 0.316 | 0.264 | 0.324 | 0.306 | 0.373 | 0.311 | 0.361 |
| FS05% ETTm2 (336) | 0.301 | 0.378 | 0.287 | 0.341 | 0.299 | 0.341 | 0.273 | 0.330 | 0.318 | 0.353 | 0.334 | 0.367 | 0.380 | 0.423 | 0.338 | 0.366 |
| FS05% ETTm2 (720) | 0.412 | 0.410 | 0.391 | 0.406 | 0.405 | 0.408 | 0.433 | 0.412 | 0.460 | 0.436 | 0.454 | 0.432 | 0.674 | 0.583 | 0.509 | 0.465 |
| FS05% ETTm2 (Avg) | 0.285 | 0.341 | 0.272 | 0.329 | 0.281 | 0.331 | 0.274 | 0.323 | 0.308 | 0.346 | 0.315 | 0.353 | 0.399 | 0.426 | 0.345 | 0.373 |
| FS05% Weather (96) | 0.170 | 0.225 | 0.167 | 0.215 | 0.170 | 0.223 | 0.172 | 0.263 | 0.175 | 0.230 | 0.171 | 0.224 | 0.184 | 0.242 | 0.207 | 0.253 |
| FS05% Weather (192) | 0.219 | 0.267 | 0.217 | 0.265 | 0.223 | 0.269 | 0.224 | 0.271 | 0.227 | 0.276 | 0.230 | 0.277 | 0.228 | 0.283 | 0.272 | 0.307 |
| FS05% Weather (336) | 0.278 | 0.313 | 0.268 | 0.299 | 0.277 | 0.310 | 0.282 | 0.321 | 0.286 | 0.322 | 0.294 | 0.326 | 0.279 | 0.322 | 0.313 | 0.328 |
| FS05% Weather (720) | 0.358 | 0.371 | 0.341 | 0.348 | 0.355 | 0.368 | 0.366 | 0.381 | 0.366 | 0.379 | 0.384 | 0.387 | 0.364 | 0.388 | 0.400 | 0.385 |
| FS05% Weather (Avg) | 0.256 | 0.294 | 0.248 | 0.282 | 0.256 | 0.293 | 0.261 | 0.309 | 0.264 | 0.302 | 0.270 | 0.304 | 0.264 | 0.309 | 0.298 | 0.318 |

## A.7 ADDITIONAL ABLATION STUDY: ADAPTIVE MODEL REDUCTION AND RANDOM SELECTION OF CONTEXT EXAMPLES

Regarding the Adaptive Model Reduction of ICTSP, because we use a pre-LayerNorm Transformer as described in Eq. 1, it inherently maintains a direct residual shortcut from the input token to the output token.

a) With small-scale initialization, if the Transformer layer does not introduce any residual effects, the model can be approximated as having only a linear projection from input to output. This is similar to a single linear layer NLinear model (as we apply last-value demeaning for tokens). b) If all attention layers in ICTSP are bypassed during training, the model can be viewed as a pure univariate MLP predictor, which many recent TSF structures consider the core temporal predictor [5-7]. Note that neither of above scenarios utilizes context tokens. c) If the information in context examples provides no benefit for predicting the target token, the context information is entirely bypassed, reducing the model to a Series-wise Transformer (like iTransformer). In these cases, the target token can be directly interpreted as the series embedding since the future portion of the target token is not trained in-context. d) When context forecasting task tokens are helpful in determining the in-context predictor for target tokens, ICTSP adjusts its prediction for the target token based on the context forecasting tasks. Here, context examples can be viewed as training samples in a supervised learning setup. Even when only a subset of context examples is used, ICTSP can still perform in-context predictor updates based on the provided few samples, achieving better performance than models that

completely disregard context examples. However, its performance will not match that of the full ICTSP, which uses more in-context training samples.

Based on the above discussion, we designed the following progressive experiments to compare the performance differences resulting from using only a subset of the model structure. The experiments were conducted on the Weather and ETTm2 datasets, as shown in Table A.7.

A. We remove all Transformer blocks from ICTSP. In this case, the model only includes the input and output linear layers for the target token, effectively reducing it to an NLinear model.

B. We retain only the MLP feedforward layers within the Transformer blocks of ICTSP. The model can then be considered a univariate MLP predictor.

C. We retain the Transformer blocks in ICTSP but drop all context tokens. The model then reduces to a Series-wise Transformer (iTransformer).

D. We retain the full ICTSP structure but randomly keep only 25% of the context tokens in the context examples. The model sees randomly incomplete in-context forecasting task samples.

E. We increase the random retention rate of context tokens to 50%.

F. We further increase the random retention rate of context tokens to 75%.

G. We use all context tokens, representing the full ICTSP model.

Table 13: Ablation study results of the adaptive model reduction and random selection of context examples

| Methods | A. | | B. | | C. | | D. | | E. | | F. | | G. | |
|---|---|---|---|---|---|---|---|---|---|---|---|---|---|---|
| Metric | MSE | MAE | MSE | MAE | MSE | MAE | MSE | MAE | MSE | MAE | MSE | MAE | MSE | MAE |
| Weather (96) | 0.168 | 0.216 | 0.148 | 0.200 | 0.145 | 0.205 | 0.146 | 0.203 | 0.140 | 0.199 | 0.140 | 0.198 | 0.139 | 0.194 |
| Weather (192) | 0.213 | 0.259 | 0.191 | 0.247 | 0.193 | 0.253 | 0.190 | 0.248 | 0.188 | 0.245 | 0.185 | 0.238 | 0.186 | 0.236 |
| Weather (336) | 0.259 | 0.294 | 0.245 | 0.295 | 0.249 | 0.302 | 0.244 | 0.295 | 0.241 | 0.286 | 0.240 | 0.283 | 0.239 | 0.279 |
| Weather (720) | 0.325 | 0.344 | 0.319 | 0.349 | 0.314 | 0.343 | 0.311 | 0.341 | 0.312 | 0.339 | 0.309 | 0.335 | 0.306 | 0.320 |
| Weather (Avg) | 0.241 | 0.278 | 0.226 | 0.273 | 0.225 | 0.276 | 0.223 | 0.272 | 0.220 | 0.267 | 0.219 | 0.264 | 0.218 | 0.257 |
| ETTm2 (96) | 0.163 | 0.253 | 0.163 | 0.255 | 0.165 | 0.258 | 0.162 | 0.255 | 0.161 | 0.253 | 0.161 | 0.251 | 0.159 | 0.248 |
| ETTm2 (192) | 0.219 | 0.290 | 0.219 | 0.298 | 0.220 | 0.301 | 0.216 | 0.297 | 0.214 | 0.295 | 0.215 | 0.294 | 0.212 | 0.289 |
| ETTm2 (336) | 0.274 | 0.328 | 0.272 | 0.331 | 0.271 | 0.331 | 0.267 | 0.329 | 0.268 | 0.330 | 0.268 | 0.328 | 0.268 | 0.326 |
| ETTm2 (720) | 0.361 | 0.385 | 0.354 | 0.387 | 0.361 | 0.386 | 0.358 | 0.386 | 0.355 | 0.388 | 0.349 | 0.385 | 0.347 | 0.382 |
| ETTm2 (Avg) | 0.254 | 0.314 | 0.252 | 0.318 | 0.254 | 0.319 | 0.251 | 0.317 | 0.250 | 0.317 | 0.248 | 0.315 | 0.247 | 0.311 |

It can be observed that when the model transitions from A. NLinear to B. Univariate MLP, there is a significant performance improvement. This is because the MLP introduces more weights, enabling it to more effectively handle the temporal relationships within tokenization.

When the model transitions from B. Univariate MLP to C. Series-wise Transformer, the performance improvement is less significant. This indicates that on weakly correlated real-world datasets, the primary improvement of the Series-wise Transformer compared to a Temporal-wise Transformer comes from the use of series tokenization for better handling of temporal relationship, rather than using the attention layers to capture aligned series-wise relationships. The MLP structure in B. alone is sufficient for this architecture to achieve performance comparable to the Series-wise Transformer.

As the model expands from C. to D., E., F., and G., the number of visible context examples gradually increases. It can be seen that using only 25%-50% of randomly selected context tokens is sufficient for ICTSP to significantly outperform the Series-wise Transformer without any context examples. This demonstrates that the attention layers in the model effectively perform the task of in-context predictor determination. Moreover, increasing the proportion of context tokens to 75% yields further performance improvements, with the model's performance closely approaching that of the full ICTSP utilizing all context tokens.

## A.8 EXPLAINING THE EFFECTIVE CAPTURING OF SERIES-WISE RELATIONSHIP OF ICTSP THROUGH A FEATURE SPACE PERSPECTIVE

We propose a feature space perspective to explain why ICTSP effectively models both temporal-wise and series-wise relationships.

- From the ICL perspective, the attention layers in Transformers construct algorithms (e.g., linear regression, shallow MLPs) between the input $(x, y)$ pairs. These layers can be seen as dynamically solving a supervised learning problem based on the input datapoint tokens.

- Temporal-wise Transformers can be viewed as solving a regression problem where the input is temporal indices $(x)$, and the output is multi-channel values $(y)$.

- Series-wise Transformers and ICTSP, on the other hand, can be seen as solving a regression problem where the input is a time series segment $(x)$, and the output is another time series segment $(y)$.

- In the default assumptions of regression problems, it is relatively straightforward to model relationships between input features within the same datapoint and relationships between the same input features across different datapoints (e.g., through linear combinations). However, it is much harder to model relationships between different input features across different datapoints.

- For Series-wise Transformers and ICTSP, we want the input time series segment $(x)$ to have shifting lag effects between its features (similar to the classic Box–Jenkins method). However, without explicitly indicating this, the default assumptions of regression problems make it easier for ICL to model:

    - Relationships between features within the same datapoint (temporal dependencies).
    - Relationships between the same features across different datapoints (temporal-aligned inter-series dependencies).

    However, it is much harder for ICL to recognize that relationships may exist between different features across different datapoints (unaligned inter-series dependencies). This is a key limitation of Series-wise Transformers when modeling cross-channel dependencies.

- ICTSP addresses this limitation by explicitly providing historical forecasting tasks as context example tokens. These historical forecasting tasks inherently have a shifted structure, which directly prompts the model to consider the necessity of unaligned inter-series relationships.

- As a result, ICTSP achieves:

    - The ability to model temporal relationships through its tokenization approach.
    - The ability to model aligned inter-series dependencies through its output token formulation.
    - The ability to model unaligned inter-series dependencies through the provision of shifted context forecasting examples.

The underlying cause of these challenges lies in the shifting time-dependent structure unique to time series data. In other supervised learning problems, we typically do not consider shifting certain features to analyze their relationships with others. Therefore, in ICL problems where each datapoint is treated as a token, ICTSP outperforms Series-wise Transformers by effectively teaching the model to learn these shifting dependencies.

## A.9 ADDITIONAL RESULTS OF ABLATION STUDY: ICTSP WITH LIMITED (MASKED) LOOKBACK INFORMATION

## A.10 ADDITIONAL RESULTS OF ABLATION STUDY: ADAPTIVE MODEL REDUCTION AND RANDOM SELECTION OF CONTEXT EXAMPLES

Building on the experiments A. - G. discussed earlier, we add the following three experiments, H. - J.. Here, we consider 4096 historical timesteps preceding each forecasting timestep. We sample our context examples from the first 25% of this historical time interval. In other words, the starting points of our context forecasting examples may range from step 1 to step 1024. Since the forecasting task length $L_b + L_P$ ranges from $512 + 96 = 608$ to $512 + 720 = 1232$, the ending points of the context forecasting examples range from step 609 to step 2256.

Because the target token requires a lookback window of 512 timesteps, any context portion entirely unrelated to the target token lies before step 3584. In this setting:

For $L_P = 720$, the region from step 2256 to step 3584, a span of 1328 steps, is completely skipped and does not contribute to forecasting.

Table 14: Full results of performance comparison of ICTSP (Limited) with six baselines on full data setting.

| Methods | ICTSP | | ICTSP (Limited) | | Time-LLM | | GPT4TS | | PatchTST | | FedFormer | | TimesNet | | DLinear | |
|---|---|---|---|---|---|---|---|---|---|---|---|---|---|---|---|---|
| Metric | MSE | MAE | MSE | MAE | MSE | MAE | MSE | MAE | MSE | MAE | MSE | MAE | MSE | MAE | MSE | MAE |
| ETTh1 (96) | 0.366 | 0.393 | 0.366 | 0.393 | 0.362 | 0.392 | 0.376 | 0.397 | 0.370 | 0.399 | 0.376 | 0.419 | 0.384 | 0.402 | 0.375 | 0.399 |
| ETTh1 (192) | 0.399 | 0.412 | 0.400 | 0.414 | 0.398 | 0.418 | 0.416 | 0.418 | 0.413 | 0.421 | 0.420 | 0.448 | 0.436 | 0.429 | 0.405 | 0.416 |
| ETTh1 (336) | 0.426 | 0.427 | 0.428 | 0.429 | 0.430 | 0.427 | 0.442 | 0.433 | 0.422 | 0.436 | 0.459 | 0.465 | 0.491 | 0.469 | 0.439 | 0.443 |
| ETTh1 (720) | 0.424 | 0.446 | 0.428 | 0.451 | 0.442 | 0.457 | 0.477 | 0.456 | 0.447 | 0.466 | 0.506 | 0.507 | 0.521 | 0.500 | 0.472 | 0.490 |
| ETTh1 (Avg) | 0.404 | 0.420 | 0.406 | 0.422 | 0.408 | 0.424 | 0.428 | 0.426 | 0.413 | 0.431 | 0.440 | 0.460 | 0.458 | 0.450 | 0.423 | 0.437 |
| ETTh2 (96) | 0.265 | 0.335 | 0.265 | 0.335 | 0.268 | 0.328 | 0.285 | 0.342 | 0.274 | 0.336 | 0.358 | 0.397 | 0.340 | 0.374 | 0.289 | 0.353 |
| ETTh2 (192) | 0.326 | 0.374 | 0.327 | 0.375 | 0.329 | 0.375 | 0.354 | 0.389 | 0.339 | 0.379 | 0.429 | 0.439 | 0.402 | 0.414 | 0.383 | 0.418 |
| ETTh2 (336) | 0.349 | 0.396 | 0.352 | 0.399 | 0.368 | 0.409 | 0.373 | 0.407 | 0.329 | 0.380 | 0.496 | 0.487 | 0.452 | 0.452 | 0.448 | 0.465 |
| ETTh2 (720) | 0.370 | 0.397 | 0.371 | 0.402 | 0.372 | 0.420 | 0.406 | 0.441 | 0.379 | 0.422 | 0.463 | 0.474 | 0.462 | 0.468 | 0.605 | 0.551 |
| ETTh2 (Avg) | 0.328 | 0.376 | 0.329 | 0.378 | 0.334 | 0.383 | 0.355 | 0.395 | 0.330 | 0.379 | 0.437 | 0.449 | 0.414 | 0.427 | 0.431 | 0.447 |
| ETTm1 (96) | 0.280 | 0.336 | 0.282 | 0.340 | 0.272 | 0.334 | 0.292 | 0.346 | 0.290 | 0.342 | 0.379 | 0.419 | 0.338 | 0.375 | 0.299 | 0.343 |
| ETTm1 (192) | 0.318 | 0.361 | 0.324 | 0.365 | 0.310 | 0.358 | 0.332 | 0.372 | 0.332 | 0.369 | 0.426 | 0.441 | 0.374 | 0.387 | 0.335 | 0.365 |
| ETTm1 (336) | 0.355 | 0.383 | 0.355 | 0.385 | 0.352 | 0.384 | 0.366 | 0.394 | 0.366 | 0.392 | 0.445 | 0.459 | 0.410 | 0.411 | 0.369 | 0.386 |
| ETTm1 (720) | 0.416 | 0.415 | 0.415 | 0.418 | 0.383 | 0.411 | 0.417 | 0.421 | 0.416 | 0.420 | 0.543 | 0.490 | 0.478 | 0.450 | 0.425 | 0.421 |
| ETTm1 (Avg) | 0.342 | 0.374 | 0.344 | 0.377 | 0.329 | 0.372 | 0.352 | 0.383 | 0.351 | 0.381 | 0.448 | 0.452 | 0.400 | 0.406 | 0.357 | 0.379 |
| ETTm2 (96) | 0.159 | 0.248 | 0.160 | 0.241 | 0.161 | 0.253 | 0.173 | 0.262 | 0.165 | 0.255 | 0.203 | 0.287 | 0.187 | 0.267 | 0.167 | 0.269 |
| ETTm2 (192) | 0.212 | 0.289 | 0.216 | 0.290 | 0.219 | 0.293 | 0.229 | 0.301 | 0.220 | 0.292 | 0.269 | 0.328 | 0.249 | 0.309 | 0.224 | 0.303 |
| ETTm2 (336) | 0.268 | 0.326 | 0.268 | 0.326 | 0.271 | 0.329 | 0.286 | 0.341 | 0.274 | 0.329 | 0.325 | 0.366 | 0.321 | 0.351 | 0.281 | 0.342 |
| ETTm2 (720) | 0.347 | 0.382 | 0.354 | 0.383 | 0.352 | 0.379 | 0.378 | 0.401 | 0.362 | 0.385 | 0.421 | 0.415 | 0.408 | 0.403 | 0.397 | 0.421 |
| ETTm2 (Avg) | 0.247 | 0.311 | 0.250 | 0.310 | 0.251 | 0.314 | 0.267 | 0.326 | 0.255 | 0.315 | 0.305 | 0.349 | 0.291 | 0.333 | 0.267 | 0.334 |
| Weather (96) | 0.139 | 0.194 | 0.142 | 0.198 | 0.147 | 0.201 | 0.162 | 0.212 | 0.149 | 0.198 | 0.217 | 0.296 | 0.172 | 0.220 | 0.176 | 0.237 |
| Weather (192) | 0.186 | 0.236 | 0.187 | 0.237 | 0.189 | 0.234 | 0.204 | 0.248 | 0.194 | 0.241 | 0.276 | 0.336 | 0.219 | 0.261 | 0.220 | 0.282 |
| Weather (336) | 0.239 | 0.279 | 0.242 | 0.281 | 0.262 | 0.279 | 0.254 | 0.282 | 0.245 | 0.282 | 0.339 | 0.380 | 0.280 | 0.306 | 0.265 | 0.319 |
| Weather (720) | 0.306 | 0.320 | 0.308 | 0.322 | 0.304 | 0.316 | 0.326 | 0.337 | 0.314 | 0.334 | 0.403 | 0.428 | 0.365 | 0.359 | 0.333 | 0.362 |
| Weather (Avg) | 0.218 | 0.257 | 0.220 | 0.260 | 0.226 | 0.258 | 0.237 | 0.271 | 0.226 | 0.264 | 0.309 | 0.360 | 0.259 | 0.287 | 0.249 | 0.300 |
| ECL (96) | 0.127 | 0.221 | 0.129 | 0.221 | 0.131 | 0.224 | 0.139 | 0.238 | 0.129 | 0.222 | 0.193 | 0.308 | 0.168 | 0.272 | 0.140 | 0.237 |
| ECL (192) | 0.148 | 0.241 | 0.151 | 0.243 | 0.152 | 0.241 | 0.153 | 0.251 | 0.157 | 0.240 | 0.201 | 0.315 | 0.184 | 0.289 | 0.153 | 0.249 |
| ECL (336) | 0.158 | 0.257 | 0.158 | 0.259 | 0.160 | 0.248 | 0.169 | 0.266 | 0.163 | 0.259 | 0.214 | 0.329 | 0.198 | 0.300 | 0.169 | 0.267 |
| ECL (720) | 0.181 | 0.283 | 0.187 | 0.286 | 0.192 | 0.298 | 0.206 | 0.297 | 0.197 | 0.290 | 0.246 | 0.355 | 0.220 | 0.320 | 0.203 | 0.301 |
| ECL (Avg) | 0.154 | 0.251 | 0.156 | 0.252 | 0.159 | 0.253 | 0.167 | 0.263 | 0.162 | 0.253 | 0.214 | 0.327 | 0.193 | 0.295 | 0.166 | 0.264 |
| Traffic (96) | 0.359 | 0.251 | 0.368 | 0.252 | 0.362 | 0.248 | 0.388 | 0.282 | 0.360 | 0.249 | 0.587 | 0.366 | 0.593 | 0.321 | 0.410 | 0.282 |
| Traffic (192) | 0.372 | 0.256 | 0.371 | 0.251 | 0.374 | 0.247 | 0.407 | 0.290 | 0.379 | 0.256 | 0.604 | 0.373 | 0.617 | 0.336 | 0.423 | 0.287 |
| Traffic (336) | 0.382 | 0.273 | 0.383 | 0.275 | 0.385 | 0.271 | 0.412 | 0.294 | 0.392 | 0.264 | 0.621 | 0.383 | 0.629 | 0.336 | 0.436 | 0.296 |
| Traffic (720) | 0.432 | 0.291 | 0.432 | 0.295 | 0.430 | 0.288 | 0.450 | 0.312 | 0.432 | 0.286 | 0.626 | 0.382 | 0.640 | 0.350 | 0.466 | 0.315 |
| Traffic (Avg) | 0.386 | 0.268 | 0.389 | 0.268 | 0.388 | 0.264 | 0.414 | 0.295 | 0.391 | 0.264 | 0.610 | 0.376 | 0.620 | 0.336 | 0.434 | 0.295 |

For $L_P = 96$, the region from step 609 to step 3584, a span of 2975 steps, is completely skipped and does not contribute to forecasting.

This pattern goes accordingly for other values of $L_P$.

Since ICL can be viewed as dynamic supervised learning performed during inference, the above setup can be considered a few-shot context example ICL scenario. It resembles the few-shot training setup described in the original paper, where datapoints from a certain time period are skipped.

We conduct the experiments as follows:

H. From the range of 1 to 1024 for context example starting points, we randomly select 10% of the steps as starting points to construct forecasting task examples. This is referred to as using 10% Distant Context Examples.

I. From the range of 1 to 1024 for context example starting points, we randomly select 25% of the steps as starting points to construct forecasting task examples. This is referred to as using 25% Distant Context Examples.

J. From the range of 1 to 1024 for context example starting points, we randomly select 75% of the steps as starting points to construct forecasting task examples. This is referred to as using 75% Distant Context Examples.

It can be observed that even when introducing relatively distant context examples, they still provide better performance compared to the Series-wise Transformer, which does not use any context tokens. Additionally, when only a small portion of context examples is introduced, the performance improvement compared to having no context is relatively modest. However, as the number of context examples increases, their contribution to performance also grows. This aligns with previous observations in ICL literature regarding the impact of the number of context examples used.

Table 15: Ablation study results of the adaptive model reduction and the effects of distant context example.

| Methods | A. | B. | C. | H. | I. | J. | G. |
|---|---|---|---|---|---|---|---|
| Weather (96) | 0.168 | 0.148 | 0.145 | 0.146 | 0.145 | 0.142 | 0.139 |
| Weather (192) | 0.213 | 0.191 | 0.193 | 0.192 | 0.191 | 0.189 | 0.186 |
| Weather (336) | 0.259 | 0.245 | 0.249 | 0.246 | 0.243 | 0.242 | 0.239 |
| Weather (720) | 0.325 | 0.319 | 0.314 | 0.314 | 0.311 | 0.310 | 0.306 |
| Weather (Avg) | 0.241 | 0.226 | 0.225 | 0.225 | 0.223 | 0.221 | 0.218 |
| ETTm2 (96) | 0.163 | 0.163 | 0.165 | 0.164 | 0.164 | 0.161 | 0.159 |
| ETTm2 (192) | 0.219 | 0.219 | 0.220 | 0.218 | 0.219 | 0.215 | 0.212 |
| ETTm2 (336) | 0.274 | 0.272 | 0.271 | 0.270 | 0.270 | 0.268 | 0.268 |
| ETTm2 (720) | 0.361 | 0.354 | 0.361 | 0.355 | 0.352 | 0.351 | 0.347 |
| ETTm2 (Avg) | 0.254 | 0.252 | 0.254 | 0.252 | 0.251 | 0.249 | 0.247 |

## A.11 ABLATION STUDY: UNDERSTANDING THE SPECIFIC CONTRIBUTIONS OF CONTEXT EXAMPLES

We designed the following experiments to demonstrate how introducing forecasting tasks as prompt tokens specifically contributes to performance. These experiments aim to validate that ICTSP relies on context examples to work via ICL. Particularly in the zero-shot setting, where no related training samples' information is in model parameters, ICTSP's superior performance is primarily driven by its ability to update the forecasting predictor using context examples. Different types of context forecasting examples provide specific improvements to the corresponding forecasting task targets.

We use ETTm2 as the target dataset and train ICTSP under three different training set configurations: ETTm2, ETTm1, and a combined set of (ETTh1, ETTh2, ETTm1). For these configurations, we conduct the corresponding full data training and zero-shot transfer learning. During evaluation, we independently generate context examples from series $X_1, \ldots, X_7$ in ETTm2 and feed them to the context part of ICTSP for test set MSE evaluation. We use $L_P = 96$, with all other settings matching those in the main experiments.

We construct the following variants:

1. w/o Context: No context examples are used from any series.

2. Context ($X_j$): Only the context examples corresponding to series $X_j$ are used.

3. Context (Full): The full ICTSP using all context examples.

For each variant, we report the MSE ($X_j$) corresponding to each series $j$ in the test set forecasting results. Below are the results for the three experimental settings.

## A.11.1 ETTM2 -> ETTM2 (FULL DATA)

From the results below, we can observe that using the context examples from a specific series $X_j$ improves the model's performance on the corresponding MSE ($X_j$). However, we cannot fully attribute this improvement to the benefits of ICL. In the full data setting, the model's parameters already contain information about the relationships between series within the dataset. As a result, the model may still leverage the information from context examples through these fixed relationships encoded in the weights. We can see that introducing $X_j$ context not only improves the performance on $X_j$ but also significantly enhances the performance of other series that are highly correlated with $X_j$. Therefore, we cannot really distinguish whether the improvement arises from fixed series relationships in the parameters or from ICL.

Table 16: Ablation study results of understanding the specific contributions of context examples. The models are trained on ETTm2 and tested on ETTm2.

| Method | w/o Context | Context ($X_1$) | Context ($X_2$) | Context ($X_3$) | Context ($X_4$) | Context ($X_5$) | Context ($X_6$) | Context ($X_7$) | Context (Full) | Time-LLM | GPT4TS | PatchTST | TimesNet | DLinear |
|---|---|---|---|---|---|---|---|---|---|---|---|---|---|---|
| MSE ($X_1$) | 0.281 | 0.268 | 0.286 | 0.268 | 0.277 | 0.278 | 0.294 | 0.283 | 0.267 | - | - | - | - | - |
| MSE ($X_2$) | 0.198 | 0.194 | 0.192 | 0.195 | 0.194 | 0.209 | 0.195 | 0.196 | 0.192 | - | - | - | - | - |
| MSE ($X_3$) | 0.095 | 0.095 | 0.099 | 0.095 | 0.096 | 0.107 | 0.098 | 0.095 | 0.092 | - | - | - | - | - |
| MSE ($X_4$) | 0.394 | 0.389 | 0.400 | 0.385 | 0.383 | 0.407 | 0.388 | 0.383 | 0.378 | - | - | - | - | - |
| MSE ($X_5$) | 0.111 | 0.109 | 0.109 | 0.116 | 0.106 | 0.105 | 0.111 | 0.124 | 0.107 | - | - | - | - | - |
| MSE ($X_6$) | 0.008 | 0.009 | 0.009 | 0.010 | 0.007 | 0.011 | 0.007 | 0.012 | 0.008 | - | - | - | - | - |
| MSE ($X_7$) | 0.078 | 0.072 | 0.080 | 0.070 | 0.079 | 0.093 | 0.083 | 0.068 | 0.074 | - | - | - | - | - |
| MSE (All) | 0.166 | 0.162 | 0.168 | 0.163 | 0.163 | 0.173 | 0.168 | 0.166 | 0.160 | 0.161 | 0.173 | 0.165 | 0.187 | 0.167 |

## A.11.2   ETTM1 -> ETTM2 (ZERO-SHOT)

We conducted zero-shot experiments where ICTSP was trained on ETTm1 and evaluated on ETTm2. In this scenario, the model parameters contain no fixed effects from the target dataset. Without using context examples, the model relies solely on its fixed weights learned from the source dataset to perform temporal forecasting inference for each target token, resulting in a significantly worse MSE of 0.208 compared to other results.

We can observe the performance imporvements when introducing context tokens from $X_j$ corresponding to series $j$. Since the model parameters lack prior knowledge of the channel structure of the target dataset, the improvement is most pronounced for the corresponding $X_j$ and diminish for other series. This task-specific improvement originates from ICL, where the model uses a task's question-answer prompt to enhance performance on the specific task.

Through these experiments, we effectively disentangled the influence of learned channel structure from the effectiveness of ICL. This demonstrates the true reason behind ICTSP's strong performance on zero-shot tasks.

Table 17: Ablation study results of understanding the specific contributions of context examples. The models are trained on ETTm1 and tested on ETTm2.

| Method | w/o Context | Context ($X_1$) | Context ($X_2$) | Context ($X_3$) | Context ($X_4$) | Context ($X_5$) | Context ($X_6$) | Context ($X_7$) | Context (Full) | Time-LLM | GPT4TS | PatchTST | TimesNet | DLinear |
|---|---|---|---|---|---|---|---|---|---|---|---|---|---|---|
| MSE ($X_1$) | 0.320 | 0.279 | 0.287 | 0.284 | 0.282 | 0.287 | 0.280 | 0.290 | 0.284 | - | - | - | - | - |
| MSE ($X_2$) | 0.226 | 0.204 | 0.203 | 0.207 | 0.204 | 0.204 | 0.205 | 0.213 | 0.198 | - | - | - | - | - |
| MSE ($X_3$) | 0.141 | 0.100 | 0.105 | 0.100 | 0.102 | 0.103 | 0.101 | 0.111 | 0.098 | - | - | - | - | - |
| MSE ($X_4$) | 0.440 | 0.436 | 0.435 | 0.446 | 0.429 | 0.437 | 0.432 | 0.432 | 0.391 | - | - | - | - | - |
| MSE ($X_5$) | 0.144 | 0.109 | 0.112 | 0.111 | 0.108 | 0.108 | 0.109 | 0.129 | 0.112 | - | - | - | - | - |
| MSE ($X_6$) | 0.048 | 0.008 | 0.009 | 0.008 | 0.009 | 0.008 | 0.007 | 0.025 | 0.008 | - | - | - | - | - |
| MSE ($X_7$) | 0.140 | 0.076 | 0.081 | 0.085 | 0.080 | 0.079 | 0.079 | 0.074 | 0.083 | - | - | - | - | - |
| MSE (All) | 0.208 | 0.173 | 0.176 | 0.177 | 0.174 | 0.175 | 0.173 | 0.182 | 0.168 | 0.169 | 0.217 | 0.195 | 0.222 | 0.221 |

## A.11.3   ETTH1,ETTH2,ETTM1 -> ETTM2 (ZERO-SHOT)

We further trained ICTSP on a combined dataset (ETTh1, ETTh2, ETTm1) containing more datapoints. Compared to the previous experiment, we observed two key improvements:

1. The model's ability to use different context prompt tokens to achieve corresponding performance improvements was further enhanced, resulting in better overall performance than in the previous experiment.

2. The task-specific performance improvement effect became more pronounced for the series corresponding to the provided context.

These results highlight the dominant role of ICTSP's ICL ability in achieving strong performance on zero-shot tasks.

Table 18: Ablation study results of understanding the specific contributions of context examples. The models are trained on ETTm1 and tested on ETTm2.

| Method | w/o Context | Context ($X_1$) | Context ($X_2$) | Context ($X_3$) | Context ($X_4$) | Context ($X_5$) | Context ($X_6$) | Context ($X_7$) | Context (Full) |
|---|---|---|---|---|---|---|---|---|---|
| MSE ($X_1$) | 0.317 | 0.279 | 0.286 | 0.284 | 0.287 | 0.290 | 0.287 | 0.285 | 0.278 |
| MSE ($X_2$) | 0.225 | 0.207 | 0.196 | 0.206 | 0.202 | 0.204 | 0.198 | 0.198 | 0.200 |
| MSE ($X_3$) | 0.116 | 0.101 | 0.107 | 0.099 | 0.105 | 0.100 | 0.101 | 0.102 | 0.098 |
| MSE ($X_4$) | 0.442 | 0.432 | 0.413 | 0.439 | 0.410 | 0.431 | 0.418 | 0.414 | 0.392 |
| MSE ($X_5$) | 0.126 | 0.110 | 0.112 | 0.109 | 0.110 | 0.107 | 0.112 | 0.108 | 0.107 |
| MSE ($X_6$) | 0.013 | 0.008 | 0.013 | 0.008 | 0.010 | 0.008 | 0.007 | 0.008 | 0.009 |
| MSE ($X_7$) | 0.085 | 0.083 | 0.088 | 0.079 | 0.083 | 0.079 | 0.084 | 0.073 | 0.082 |
| MSE (All) | 0.189 | 0.174 | 0.174 | 0.175 | 0.172 | 0.174 | 0.173 | 0.170 | 0.167 |

