# OpenReview forum: "In-context Time Series Predictor"
_ICLR.cc/2025/Conference — ICLR 2025 Poster_

### Official Review · Reviewer_uZY9 · 2024-10-30

**Soundness:** 3
**Presentation:** 3
**Contribution:** 4
**Rating:** 8
**Confidence:** 3

**Summary:**

The authors propose a new in-context learning framework for time series forecasting by using "time series forecasting tasks" as input tokens. They show that this new framework outperforms the more traditional Temporal-wise Transformers and Series-wise Transformers across full-data, few-shot, and zero-shot settings.

**Strengths:**

In-context learning for time series forecasting is an important problem with wide applications. As far as I can tell, using "time series forecasting tasks" as input tokens is a new and interesting idea, and the authors demonstrate that it indeed has utility using systematic benchmarks. The approach has a "category theory" flavor to it, and the differences from earlier frameworks are well explained.

**Weaknesses:**

In terms of presentation, I think there is room to make it more accessible to readers not directly working in the same field. In particular, more detailed explanation and illustration of the "using time series forecasting tasks as input tokens" idea and how it is implemented could be helpful.

**Questions:**

N/A

---

> ### Author Response · Authors · 2024-11-20
>
> Thank you very much for recognizing the contribution of our paper. We are delighted to receive your acknowledgment. We are pleased that you found our work to have a "category theory" flavor, as this aligns with what we aimed to convey in the paper. Using the ICL perspective, we analyzed previous Transformer-based time series forecasting structures, identified their issues, categorized them into Temporal-wise and Series-wise Transformers, and explained how past strategies, such as channel independence and patching operations, enhance Temporal-wise Transformers.
>
> We are happy to provide a summary of the core discussions in the paper as follows:
>
> | **Methods** | **Tokenization** | **ICL (x, y) Pair** | **Token Representation for** | **Attention to Learn** | **Suited For** | **Not Suited For** | **ICL Context Examples Describing** | **Channel Structure** | **Problem** |
> |:-:|:-:|:-:|:-:|:-:|:-:|:-:|:-:|:-:|:-:|
> | **Temporal-wise Transformer** | Timestep embedding | (Timestep index, Multi-channel values) | Single-step inter-series dependencies | Temporal-wise dynamics of series dependencies | Strong and consistent inter-series dependencies | Weak inter-series dependencies in real-world datasets | Underlying inter-series effects vary with the time indices in the context lookback (likely nonexistent or weak) | Fixed | Only series relationships in tokenization; risks overfitting to series-wise effects from the input context |
> | **Temporal-wise Transformer (with Channel Independence)** | Scalar embedding | (Timestep index, Single-channel value) | Scalar value at a single timestep | Temporal-wise dependencies | Strong and fast-changing temporal dependencies | Strong inter-series dependencies | Temporal relationships within the context for one series | Fixed | Unable to model series-wise relationships |
> | **Temporal-wise Transformer (with Patching)** | Patch embedding | (Timestep index, Multi-channel values in a temporal interval) | Temporal-wise and series-wise relationships in a local interval | Temporal-wise relationships between local intervals | Temporal and series relationships can be automatically balanced by this structure | Few-shot/zero-shot cases with different series relationship characteristics | Temporal-wise relationships between local intervals | Fixed | Fixed channel structure in tokenization prevents generalization to datasets with different structures |
> | **Series-wise Transformer** | Series embedding | (Input series, Output series) | Temporal-wise relationships within a single series input | Aligned inter-series relationships | Real-world datasets with strong temporal-wise and aligned series-wise relationships (e.g., traffic) | Unaligned series-wise relationships, such as shifting effects between series | No context examples | Flexible but requires additional training to adapt to datasets with new channel structure | Restricted to specific dataset channel structures; unable to learn unaligned series-wise effects |
> | **ICTSP** | Embedding of forecasting tasks | (Input series, Output series) | Temporal-wise relationships in a forecasting task | Forecasting task predictors, considering both aligned and unaligned inter-series relationships in shifted context examples | Datasets with fast-changing temporal relationships, aligned or unaligned series relationships; transfering to new datasets with different channel structures | / | Historical ground truth forecasting tasks as references | Flexible; additional training is not necessary since context examples exist  | / |
>
> In the revised paper, we have added more discussions on the effectiveness of forecasting task tokenization's capturing of different relationships, and included additional ablation studies to demonstrate the superior and consistent performance of ICTSP compared to prior models across full data, few-shot, and zero-shot tasks.
>
> Thank you again for your recognition and valuable suggestions. If you have any further questions, we would be more than happy to address them.

---

### Official Review · Reviewer_H6bM · 2024-11-01

**Soundness:** 3
**Presentation:** 4
**Contribution:** 2
**Rating:** 6
**Confidence:** 2

**Summary:**

This paper present In-context Time Series Predictor (ICTSP), a new framework of doing time series forecasting. It formulates the time series prediction task into the form of in-context learning, such that in the input data is split into a few forecasting examples to guide the model complete the target prediction with the adjacent input. Experimental results demonstrate the method’s effectiveness.

**Strengths:**

1. **Novel Framework**: The paper proposes an innovative approach to time series forecasting by adopting in-context learning, which could inspire further research and development in this field.
2. **Comprehensive Experiments**: The authors present extensive experimental results, including ablation studies, which provide a robust foundation for evaluating ICTSP’s effectiveness.
3. **Clarity of Presentation**: The paper is well-written and clearly presented, facilitating a good understanding of the approach and its contributions.

**Weaknesses:**

1. **Comparative Fairness**: The baselines use an input length of 512, while ICTSP utilizes an input length of 1440. This discrepancy could impact the fairness of the comparisons.
2. **In-context Example Selection**: ICTSP frames forecasting as an in-context learning task, but it uses adjacent historical data as context examples. This approach raises questions about whether the model truly learns from in-context examples or simply encodes the input tokens in a different manner.
3. **Model Reduction Analysis**: Although the paper highlights the adaptive model reduction feature of ICTSP, there is limited visualization or analysis to support this claim.

**Questions:**

1. Could you provide results with ICTSP’s input length ($L_I$) set to 512 for a more equitable comparison with other baselines in Table 1 and 6?
2. Could you conduct an ablation study on in-context examples, possibly using randomly selected examples (without look-ahead bias) instead of adjacent history to clarify the influence of this design choice?
3. Could you provide more detailed analysis or visualizations related to the Adaptive Model Reduction to support its impact on performance?
4. How does ICTSP handle cross-channel dependencies, especially in cases with specific inter-channel relationships in the data (like traffic)?

---

> ### Author Response · Authors · 2024-11-20
>
> ```
>     1. Could you provide results with ICTSP’s input length set to 512 for a more equitable comparison with other baselines in Table 1 and 6?
> ```
>
> Thank you for this meaningful question. First, the choice of input length \(L_I\) is widely discussed in the literature, with different models often requiring different optimal lengths depending on the dataset. For example:
>
> - **GPT4TS** [1] (Section H.10) suggests using the optimal \(L_I\) reported in original papers rather than a single \(L_I\) for all baselines, as longer lookbacks often cause overfitting in some models, resulting in unfair comparisons.
> - Studies like **TiDE** [2] (Section 5.1) selects the best \(L_I\) from multiple tested lengths to ensure fair comparisons.
> - **DLinear** [3] (Figure 6), **PatchTST** [4] (Table 9), and **iTransformer** [5] (Figure 6) show that for many real-world datasets, shorter \(L_I\) often performs better as longer lookbacks can lead to overfitting.
>
> Thus, effectively utilizing longer \(L_I\) while avoiding overfitting remains a common challenge.
>
> ICTSP addresses this by treating segments of length \(L_b + L_P\) as forecasting task tokens and creating \(N = L_I - (L_b + L_P)\) task tokens to incorporate lookback information. With \(L_b = 512\), each token represents a forecasting task used by models like PatchTST or GPT4TS. ICTSP can be viewed as leveraging past forecasting task datapoints to optimize the predictor for the current forecasting task, enhancing generalization performance.
>
> While we believe this rationale supports our approach, **we are happy to conduct an ablation study using only 512-step input information**. However, this configuration disrupts ICTSP's **forecasting tasks as context** structure and is **not recommended for practical use**. ICTSP requires \(L_I > L_b + L_P\) to split the input into multiple ground truth forecasting tasks. Given \(L_b = 512\), when \(L_P = 720\), \(L_I > 512 + 720 = 1232\) is necessary to construct at least one ground truth forecasting task context token.
>
> ### Ablation Study: ICTSP with limited lookback information
>
> To fully align with the baseline models using \(L_I = 512\), we **relax ICTSP's design**, which originally required all context tokens to be ground truth forecasting tasks. Instead, we **allow some forecasting tasks to include incomplete information**. Specifically, we retain \(L_I = 1440\) but **mask the first \(1440 - 512 = 928\) timesteps in the input**, ensuring **only the last 512 steps contain true information**. The masked portion is **filled with the mean value** of each series token **from the last 512 steps**. To maintain the presence of some true forecasting task tokens for shorter \(L_P\), we shorten \(L_b\) to 336. All other hyperparameters remain unchanged. This adjusted structure is denoted as **ICTSP (Limited)**.
>
> We repeated experiments on the ETTm2 and Weather datasets, covering full data, few-shot 10\% (FS10\%), few-shot 5\% (FS05\%), and zero-shot (ZS) settings for the ETT datasets. Full results are available in Table 11 of the revised paper. Below is the summary of average MSE and average rank compared to six strong baselines.
>
> | Methods | ICTSP | ICTSP (Limited) | Time-LLM | GPT4TS | PatchTST | TimesNet | DLinear |
> |---|:-:|:-:|:-:|:-:|:-:|:-:|:-:|
> | Full ETTm2 (Avg) | **0.247** | 0.250  | 0.251 | 0.267 | 0.255 | 0.291 | 0.267 |
> | Full Weather (Avg) | **0.218** | 0.220 | 0.226 | 0.237 | 0.226 | 0.259 | 0.249 |
> | FS10% ETTm2 (Avg) | 0.275 | **0.274** | 0.277 | 0.293 | 0.296 | 0.320 | 0.316 |
> | FS10% Weather (Avg) | 0.237 | 0.238 | **0.234** | 0.238 | 0.242 | 0.279 | 0.241 |
> | FS05% ETTm2 (Avg) | 0.285 | 0.281 | **0.274** | 0.308 | 0.315 | 0.399 | 0.345 |
> | FS05% Weather (Avg)  | **0.256** | **0.256** | 0.261 | 0.264 | 0.270 | 0.264 | 0.298 |
> | ZS m1→m2 (Avg) | **0.255** | 0.258 | 0.265 | 0.314 | 0.297 | 0.322 | 0.335 |
> | ZS h1→h2 (Avg) | **0.337** | 0.339 | 0.352 | 0.407 | 0.381 | 0.421 | 0.494 |
> | AvgRank | **1.66** | 1.97 | 2.66 | 4.56 | 4.34 | 6.31 | 6.19 |
>
> From the average rankings, we observe that even with masking applied to information outside the 512-step window, **ICTSP (Limited) still significantly outperforms all baselines**. In full data and zero-shot experiments, ICTSP (Limited) performs slightly weaker than the original ICTSP but remains superior to all baselines. However, in few-shot training experiments, ICTSP (Limited) surpasses the original ICTSP. It is important to note that this improvement is **due to an additional constraint in the original ICTSP**, which, despite using $L_I=1440$, imposes **zero-filling** restrictions outside of $L_b=512$ to ensure fair comparison with baselines using $L_I=512$ (see Section A.3.3 Discussion of fair comparison for details on **few-shot experiments**).

---

> ### Author Response · Authors · 2024-11-20
>
> In the full data and zero-shot experiments, all data can be used with any $L_I$ without causing data leakage. However, in few-shot experiments, due to skipping in data sampling, a longer \(L_I\) accesses more lookback data during validation/testing, giving an unfair advantage. To address this, the original ICTSP applied **zero-masking** to restrict access to only 512 valid timesteps at the start of validation/testing, gradually expanding to \(L_I = 1440\) as more valid data became available.
>
> Nonetheless, zero-filling in forecasting task tokens caused issues because of the **last-value demeaning operation** used on the forecasting task tokens, resulting in meaningless negative last-value placeholders and reduced model performance. **ICTSP (Limited)** resolves this by using **mean-value filling**, ensuring more reasonable values in forecasting task tokens. This adjustment allows ICTSP (Limited) to outperform the original ICTSP in few-shot experiments.
>
> To clarify this, we **replaced the zero-filling used for few-shot fair comparison in the original ICTSP with mean-filling** and present the results below. ICTSP with mean-filling **significantly outperforms** all models, including Time-LLM, across all experiments. This highlights that ICTSP’s few-shot capability is inherently **stronger** than original results presented in the paper when proper mean-filling strategies are applied. Full results can be found in Table 12 of the revised paper.
>
> | Methods | ICTSP (Zero-filling) | ICTSP (Mean-filling) | ICTSP (Limited) | Time-LLM | GPT4TS | PatchTST | TimesNet | DLinear |
> |---|:-:|:-:|:-:|:-:|:-:|:-:|:-:|:-:|
> | FS10% ETTm2 (Avg) | 0.275 | **0.264** | 0.274 | 0.277 | 0.293 | 0.296 | 0.320 | 0.316 |
> | FS10% Weather (Avg) | 0.237 | **0.234** | 0.238 | **0.234** | 0.238 | 0.242 | 0.279 | 0.241 |
> | FS05% ETTm2 (Avg) | 0.285 | **0.272** | 0.281 | 0.274 | 0.308 | 0.315 | 0.399 | 0.345 |
> | FS05% Weather (Avg)  | 0.256 | **0.248** | 0.256 | 0.261 | 0.264 | 0.270 | 0.264 | 0.298 |
>
> ```
>     2. Could you conduct an ablation study on in-context examples, possibly using randomly selected examples (without look-ahead bias) instead of adjacent history to clarify the influence of this design choice?
>
>     3. Could you provide more detailed analysis or visualizations related to the Adaptive Model Reduction to support its impact on performance?
> ```
>
> Thank you for your valuable questions. We will address both issues together in our response. Regarding the **Adaptive Model Reduction** of ICTSP, because we use a pre-LayerNorm Transformer as described in Eq. 1, it inherently maintains a direct residual shortcut from the input token to the output token.
>
> a) With small-scale weight initialization, if Transformer layers introduce no residual effects (bypassed), the model simplifies to a linear projection, akin to a **NLinear model**(as we apply last-value demeaning for tokens).
> b) If attention layers are bypassed during training, the model acts as a pure **univariate MLP predictor**, which many recent TSF models [5-7] use as their core temporal predictor. Note that neither of above scenarios utilizes context tokens.
> c) When context examples provide no useful information for the predictor of the target token, ICTSP reduces to a **Series-wise Transformer (like iTransformer)**. Here, the target token can be directly interpreted as the series embedding since the future portion of the target token is not trained in-context.
> d) When **context forecasting task tokens** aid in determining the target token's predictor, ICTSP adjusts predictions based on the them. Context examples act as training samples in a supervised learning setup. Even with limited subset of context examples, ICTSP can still update its predictor in-context, outperforming models that ignore context entirely. However, its performance remains below that of the full ICTSP, which uses more context samples.
>
> ### Ablation Study: Adaptive Model Reduction and Random Selection of Context Examples
>
> Based on the above discussion, we designed the following progressive experiments to compare the performance differences resulting from using only a subset of the model structure. The experiments were conducted on the Weather and ETTm2 datasets, as shown below. The original results can be found in Table 13 of the revised paper.

---

> ### Author Response · Authors · 2024-11-20
>
> - **Model Variants**
>
> A. We **remove all Transformer blocks** from ICTSP. In this case, the model only includes the input and output linear layers for the target token, effectively **reducing it to an NLinear model**.
>
> B. We **retain only the MLP feedforward layers** within the Transformer blocks of ICTSP. The model can then be considered a **univariate MLP predictor**.
>
> C. We **retain the Transformer blocks** in ICTSP but **drop all context tokens**. The model then reduces to a **Series-wise Transformer (iTransformer)**.
>
> D. We **retain the full ICTSP structure** but **randomly keep only 25% of the context tokens** in the context examples. The model sees randomly incomplete in-context forecasting task samples.
>
> E. We increase the random retention rate of context tokens to **50%**.
>
> F. We further increase the random retention rate of context tokens to **75%**.
>
> G. We **use all context tokens**, representing the **full ICTSP model**.
>
> | Methods | A. w/o Transformer (Reduced to NLinear) | B. w/o Attention (Reduced to Univariate MLP) | C. w/o Context Tokens (Reduced to Series-wise Transformer) | D. w/ 25% Random Selected Context Tokens | E. w/ 50% Random Selected Context Tokens | F. w/ 75% Random Selected Context Tokens | G. Full ICTSP |
> |---|:-:|:-:|:-:|:-:|:-:|:-:|:-:|
> | Weather (96) | 0.168 | 0.148 | 0.145 | 0.146 | 0.140 | 0.140 | 0.139 |
> | Weather (192) | 0.213 | 0.191 | 0.193 | 0.190 | 0.188 | 0.185 | 0.186 |
> | Weather (336) | 0.259 | 0.245 | 0.249 | 0.244 | 0.241 | 0.240 | 0.239 |
> | Weather (720) | 0.325 | 0.319 | 0.314 | 0.311 | 0.312 | 0.309 | 0.306 |
> | **Weather (Avg)** | **0.241** | **0.226** | **0.225** | **0.223** | **0.220** | **0.219** | **0.218** |
> | ETTm2 (96) | 0.163 | 0.163 | 0.165 | 0.162 | 0.161 | 0.161 | 0.159 |
> | ETTm2 (192) | 0.219 | 0.219 | 0.220 | 0.216 | 0.214 | 0.215 | 0.212 |
> | ETTm2 (336) | 0.274 | 0.272 | 0.271 | 0.268 | 0.268 | 0.268 | 0.268 |
> | ETTm2 (720) | 0.361 | 0.354 | 0.361 | 0.358 | 0.355 | 0.349 | 0.347 |
> | **ETTm2 (Avg)** | **0.254** | **0.252** | **0.254** | **0.251** | **0.250** | **0.248** | **0.247** |
>
> The transition from **A. NLinear** to **B. Univariate MLP** shows a **significant performance improvement**, as the MLP introduces more weights, enabling better handling of temporal relationships within tokenization.
>
> The improvement from **B. Univariate MLP** to **C. Series-wise Transformer** is **less significant**. This indicates that on weakly correlated real-world datasets, the primary improvement of the Series-wise Transformer compared to a Temporal-wise Transformer comes from the use of series tokenization for better handling of temporal relationship, rather than using the attention layers to capture aligned series-wise relationships. The MLP structure in **B.** alone is sufficient for this architecture to achieve performance comparable to the Series-wise Transformer.
>
> As the model expands from **C.** to **D., E., F., and G.**, the number of visible context examples increases. Using **only 25%-50% of randomly selected context tokens allows ICTSP to significantly outperform the Series-wise Transformer** without any context examples. This demonstrates that the attention layers in the model effectively perform the task of in-context predictor determination. **Increasing the proportion of context tokens to 75% provides further improvements**, with performance **nearly matching the full ICTSP** using all context tokens.
>
> ```
>     4. How does ICTSP handle cross-channel dependencies, especially in cases with specific inter-channel relationships in the data (like traffic)?
> ```
>
> Thank you for raising this important question, which is key to understanding the significance of our paper. In the main paper, we discussed that **Temporal-wise Transformers** take the values of different channels at a given timestep as input tokens. This approach results in token vectors that primarily encode potentially weak inter-series relationships from real-world datasets, while leaving the more critical **temporal-wise relationships** in TSF tasks to the attention layers with dynamic weights to construct algorithms. This leads to significant overfitting to inter-series effects in the lookback window.
>
> Recent models with **patch embeddings** mitigate this by introducing local temporal relationships inside tokenization, reducing overfitting. Also, models like **PatchTST** enforce channel independence, cutting off inter-series relationships entirely to focus solely on temporal relationships.
>
> **Series-wise Transformers**, like **iTransformer**, push this further by embedding entire series as tokens, emphasizing temporal relationships even more in tokenization while leaving series-wise relationships to attention layers. However, these models mainly capture **temporally aligned effects** but not unaligned effects, making them less effective for datasets with stable **shifted inter-series dependencies**.

---

> ### Author Response · Authors · 2024-11-20
>
> **ICTSP** addresses this by incorporating historical forecasting tasks as context tokens, which naturally prompts the model to account for shifted inter-series dependencies. These task tokens inherently exhibit shifted relationships with one another, guiding the model to explore these dependencies.
>
> **Figure 4** illustrates this clearly. On a synthetic dataset (`Multi`) with weak temporal relationships but strong shifted inter-series dependencies, **Temporal-wise Transformer** and **ICTSP** significantly outperform **Series-wise Transformer**. On the real-world dataset (`ETTm2`), where temporal relationships are stronger and inter-series dependencies weaker, **ICTSP** and **Series-wise Transformer** outperform the overfitted Temporal-wise Transformer.
>
> To further aid your understanding, we propose a **feature space perspective** to explain why ICTSP effectively models both temporal-wise and series-wise relationships.
>
> - **Feature Space Perspective**
>     - From the ICL perspective, the attention layers in Transformers construct algorithms (e.g., linear regression, shallow MLPs) between the input (x, y) pairs. These layers can be seen as dynamically solving a supervised learning problem based on the input datapoint tokens.
>     - **Temporal-wise Transformers** can be viewed as solving a regression problem where the input is temporal indices (x), and the output is multi-channel values (y).
>     - **Series-wise Transformers** and **ICTSP**, on the other hand, can be seen as solving a regression problem where the input is a time series segment (x), and the output is another time series segment (y).
>     - In the default assumptions of regression problems, it is relatively straightforward to model relationships between input features within the same datapoint and relationships between the same input features across different datapoints (e.g., through linear combinations). However, it is much harder to model relationships between **different input features across different datapoints**.
>     - For **Series-wise Transformers** and **ICTSP**, we want the input time series segment (x) to have **shifting lag effects** between its features (similar to the classic Box–Jenkins method). However, without explicitly indicating this, the default assumptions of regression problems make it easier for ICL to model:
>         - Relationships between features within the same datapoint (temporal dependencies).
>         - Relationships between the same features across different datapoints (temporal-aligned inter-series dependencies).
>     - However, it is much harder for ICL to recognize that relationships may exist between **different features across different datapoints** (unaligned inter-series dependencies). This is a key limitation of Series-wise Transformers when modeling cross-channel dependencies.
>     - **ICTSP** addresses this limitation by explicitly providing historical forecasting tasks as **context example tokens**. These historical forecasting tasks inherently have a **shifted structure**, which directly prompts the model to consider the necessity of **unaligned inter-series relationships**.
>     - As a result, ICTSP achieves:
>         - The ability to model **temporal relationships** through its tokenization approach.
>         - The ability to model **aligned inter-series dependencies** through its output token formulation.
>         - The ability to model **unaligned inter-series dependencies** through the provision of shifted context forecasting examples.
>
> The underlying cause of these challenges lies in the **shifting time-dependent structure** unique to time series data. In other supervised learning problems, we typically do not consider shifting certain features to analyze their relationships with others. Therefore, in ICL problems where each datapoint is treated as a token, ICTSP outperforms Series-wise Transformers by effectively teaching the model to learn these shifting dependencies.
>
> We greatly appreciate your insightful feedback and are open to any further questions or suggestions. We look forward to engaging in more detailed discussions and thank you for your consideration.

---

> ### Author Response · Authors · 2024-11-20
>
> ### References
>
> [1] Zhou, Tian, et al. "One fits all: Power general time series analysis by pretrained lm." Advances in neural information processing systems 36 (2023): 43322-43355.
>
> [2] Das, Abhimanyu, et al. "Long-term Forecasting with TiDE: Time-series Dense Encoder." Transactions on Machine Learning Research.
>
> [3] Zeng, Ailing, et al. "Are transformers effective for time series forecasting?." Proceedings of the AAAI conference on artificial intelligence. Vol. 37. No. 9. 2023.
>
> [4] Nie, Yuqi, et al. "A Time Series is Worth 64 Words: Long-term Forecasting with Transformers." The Eleventh International Conference on Learning Representations.
>
> [5] Liu, Yong, et al. "iTransformer: Inverted Transformers Are Effective for Time Series Forecasting." The Twelfth International Conference on Learning Representations.
>
> [6] Ekambaram, Vijay, et al. "Tsmixer: Lightweight mlp-mixer model for multivariate time series forecasting." Proceedings of the 29th ACM SIGKDD Conference on Knowledge Discovery and Data Mining. 2023.
>
> [7] Wang, Shiyu, et al. "Timemixer: Decomposable multiscale mixing for time series forecasting." arXiv preprint arXiv:2405.14616 (2024).

---

> ### Comment · Reviewer_H6bM · 2024-11-22
>
> I sincerely appreciate the authors for their detailed and thoughtful response, as well as for conducting additional experiments. I still have some follow-up questions:
>
> 1.Regarding your response to Q1, the results are only on ETTm2 and Weather rather than the whole dataset. Seeing the experiment results for all the data would be more convincing. It does not need to include FS 10% / 5 % or  ZS. Just additional results like Table 1 in paper would be fine.
>
> 2.Regarding your response to Q2, Q3, unfortunately I may not be entirely clear of what I am proposing about the “randomly selected examples”. My primary doubt lies in whether the purported “in-context learning” mechanism genuinely contributes unique forecasting capabilities. Specifically, I am interested in understanding whether the model’s performance gains stem from effectively having a longer input length, or from reformulating the original forecasting problem into the proposed ICTSP framework. (See weakness 2)
>
> The experiment described in your response, involving x% randomly selected context tokens, does not fully address this question. It still could have a longer input window. Instead, I propose testing with random examples that are not from adjacent history, perhaps drawn from much older (less relevant) history or even other datasets. In this scenario, the benefit of a longer input window would be eliminated. If “in-context learning” is truly effective, the model should demonstrate improved performance even with these random examples, indicating that it can derive meaningful insights from the provided context.
>
> That said, the above is just a suggested experiment to address my concerns. If you can propose alternative experiments that more directly address this question, I would welcome those as well.

---

> > ### Author Response · Authors · 2024-11-25
> >
> > ```
> > 1.Regarding your response to Q1, ...
> > ```
> >
> > Thank you for your follow-up question. Below, we present the average MSE results of ICTSP (Limited) across all full data experiments. The **best value** in each experiment is **bolded**, and the *second best* is *italicized*:
> >
> > | Methods       | ICTSP | ICTSP (Limited) | Time-LLM | GPT4TS | PatchTST | FEDformer | TimesNet | DLinear |
> > |---|:-:|:-:|:-:|:-:|:-:|:-:|:-:|:-:|
> > | ETTh1 (Avg)   | **0.404** | *0.406*           | 0.408    | 0.428  | 0.413    | 0.440     | 0.458    | 0.423   |
> > | ETTh2 (Avg)   | **0.328** | *0.329*           | 0.334    | 0.355  | 0.330    | 0.437     | 0.414    | 0.431   |
> > | ETTm1 (Avg)   | *0.342* | 0.344           | *0.329*    | 0.352  | 0.351    | 0.448     | 0.400    | 0.357   |
> > | ETTm2 (Avg)   | **0.247** | *0.250*           | 0.251    | 0.267  | 0.255    | 0.305     | 0.291    | 0.267   |
> > | Weather (Avg) | **0.218** | *0.220*           | 0.226    | 0.237  | 0.226    | 0.309     | 0.259    | 0.249   |
> > | ECL (Avg)     | **0.154** | *0.156*           | 0.159    | 0.167  | 0.162    | 0.214     | 0.193    | 0.166   |
> > | Traffic (Avg) | **0.386** | 0.389           | *0.388*    | 0.414  | 0.391    | 0.610     | 0.620    | 0.434   |
> > | AvgRank       | **1.37**  | *2.20*            | 2.60     | 5.17   | 3.63     | 7.66      | 7.17     | 5.74    |
> >
> > It can be observed that even after relaxing the forecasting token formulation restrictions to limit the historical information, ICTSP (Limited) consistently outperforms previous baselines. It achieves excellent performance while requiring fewer computational resources. The full experimental results are available in Table 14 of the revised paper.
> >
> > ```
> > 2.Regarding your response to Q2, Q3, ...
> > ```
> >
> > Thank you for your valuable question. To address it, we will demonstrate the ICL capabilities learned by ICTSP through the following additional experiments.
> >
> > Building on the experiments `A. - G.` discussed earlier, we add the following three experiments, `H. - J.`. Here, we consider 4096 historical timesteps preceding each forecasting timestep. We sample our context examples from the first 25% of this historical time interval. In other words, the starting points of our context forecasting examples may range from step 1 to step 1024. Since the forecasting task length $L_b + L_P$ ranges from $512 + 96 = 608$ to $512 + 720 = 1232$, the ending points of the context forecasting examples range from step 609 to step 2256.
> >
> > Because the target token requires a lookback window of 512 timesteps, any context portion entirely unrelated to the target token lies before step 3584. In this setting:
> > - For $L_P = 720$, the **region from step 2256 to step 3584, a span of 1328 steps, is completely skipped and does not contribute to forecasting**.
> > ...
> > - For $L_P = 96$, the **region from step 609 to step 3584, a span of 2975 steps, is completely skipped and does not contribute to forecasting**.
> >
> > This pattern goes accordingly for other values of $L_P$.
> >
> > Since ICL can be viewed as dynamic supervised learning performed during inference, the above setup can be considered a few-shot context example ICL scenario. It resembles the few-shot training setup described in the original paper, where datapoints from a certain time period are skipped.
> >
> > We conduct the experiments as follows:
> >
> > - **Model Variants**
> >
> > A. We **remove all Transformer blocks** from ICTSP. In this case, the model only includes the input and output linear layers for the target token, effectively **reducing it to an NLinear model**.
> >
> > B. We **retain only the MLP feedforward layers** within the Transformer blocks of ICTSP. The model can then be considered a **univariate MLP predictor**.
> >
> > C. We **retain the Transformer blocks** in ICTSP but **drop all context tokens**. The model then reduces to a **Series-wise Transformer (iTransformer)**.
> >
> > G. We **use all context tokens**, representing the **full ICTSP model**.
> >
> > H. From the range of 1 to 1024 for context example starting points, we randomly select 10% of the steps as starting points to construct forecasting task examples. This is referred to as using **10% Distant Context Examples**.
> >
> > I. From the range of 1 to 1024 for context example starting points, we randomly select 25% of the steps as starting points to construct forecasting task examples. This is referred to as using **25% Distant Context Examples**.
> >
> > J. From the range of 1 to 1024 for context example starting points, we randomly select 75% of the steps as starting points to construct forecasting task examples. This is referred to as using **75% Distant Context Examples**.

---

> ### Author Response · Authors · 2024-11-25
>
> | Methods | A. w/o Transformer (Reduced to NLinear) | B. w/o Attention (Reduced to Univariate MLP) | C. w/o Context Tokens (Reduced to Series-wise Transformer) | H. 10% Distant Context Examples | I. 25% Distant Context Examples | J. 75% Distant Context Examples | G. Full ICTSP |
> |---|:-:|:-:|:-:|:-:|:-:|:-:|:-:|
> | Weather (96) | 0.168 | 0.148 | 0.145 | 0.146 | 0.145 | 0.142 | 0.139 |
> | Weather (192) | 0.213 | 0.191 | 0.193 | 0.192 | 0.191 | 0.189 | 0.186 |
> | Weather (336) | 0.259 | 0.245 | 0.249 | 0.246 | 0.243 | 0.242 | 0.239 |
> | Weather (720) | 0.325 | 0.319 | 0.314 | 0.314 | 0.311 | 0.310 | 0.306 |
> | **Weather (Avg)** | **0.241** | **0.226** | **0.225** | **0.225** | **0.223** | **0.221** | **0.218** |
> | ETTm2 (96) | 0.163 | 0.163 | 0.165 | 0.164 | 0.164 | 0.161 | 0.159 |
> | ETTm2 (192) | 0.219 | 0.219 | 0.220 | 0.218 | 0.219 | 0.215 | 0.212 |
> | ETTm2 (336) | 0.274 | 0.272 | 0.271 | 0.270 | 0.270 | 0.268 | 0.268 |
> | ETTm2 (720) | 0.361 | 0.354 | 0.361 | 0.355 | 0.352 | 0.351 | 0.347 |
> | **ETTm2 (Avg)** | **0.254** | **0.252** | **0.254** | **0.252** | **0.251** | **0.249** | **0.247** |
>
> It can be observed that even when introducing relatively distant context examples, they still provide better performance compared to the Series-wise Transformer, which does not use any context tokens. Additionally, when only a small portion of context examples is introduced, the performance improvement compared to having no context is relatively modest. However, as the number of context examples increases, their contribution to performance also grows. This aligns with previous observations in ICL literature regarding the impact of the number of context examples used.
>
> ### Ablation Study: Understanding the Specific Contributions of Context Examples
>
> We designed the following experiments to demonstrate how **introducing forecasting tasks as prompt tokens** specifically contributes to performance. These experiments aim to validate that ICTSP relies on context examples to work **via ICL**. Particularly in the zero-shot setting, where **no related training samples' information is in model parameters**, ICTSP’s superior performance is primarily **driven by its ability to update the forecasting predictor using context examples**. Different types of context forecasting examples provide **specific improvements** to the corresponding forecasting task targets.
>
> We use ETTm2 as the target dataset and train ICTSP under three different training set configurations: ETTm2, ETTm1, and a combined set of (ETTh1, ETTh2, ETTm1). For these configurations, we conduct the corresponding full data training and zero-shot transfer learning. **During evaluation, we independently generate context examples from series $X_1, \dots, X_7$ in ETTm2 and feed them to the context part of ICTSP for test set MSE evaluation**. We use $L_P=96$, with all other settings matching those in the main experiments.
>
> We construct the following variants:
>
> 1. **w/o Context**: No context examples are used from any series.
> 2. **Context ($X_j$)**: Only the context examples corresponding to series $X_j$ are used.
> 3. **Context (Full)**: The full ICTSP using all context examples.
>
> For each variant, we report the MSE ($X_j$) corresponding to each series $j$ in the test set forecasting results. Below are the results for the three experimental settings. For each MSE ($X_j$), the **best value** is **bolded**, and the *second best* is *italicized* for the **w/o Context** and **Context ($X_j$)** parts of experiments.
>
>
> ### ETTm2 -> ETTm2 (full data)
>
> From the results below, we can observe that using the context examples from a specific series $X_j$ improves the model's performance on the corresponding MSE ($X_j$). However, **we cannot fully attribute this improvement to the benefits of ICL. In the full data setting**, the model's parameters already contain information about the relationships between series within the dataset. As a result, the **model may still leverage the information from context examples through these fixed relationships encoded in the weights**. We can see that introducing $X_j$ context not only **improves the performance on $X_j$** but also significantly enhances the performance of **other series that are highly correlated with $X_j$**. Therefore, we **cannot really distinguish** whether the improvement arises from fixed series relationships in the parameters or from ICL.

---

> > ### Author Response · Authors · 2024-11-25
> >
> > | Method | w/o Context | Context ($X_1$) | Context ($X_2$) | Context ($X_3$) | Context ($X_4$) | Context ($X_5$) | Context ($X_6$) | Context ($X_7$) | \| | Context (Full) | Time-LLM | GPT4TS | PatchTST | TimesNet | DLinear |
> > |---|:---:|:---:|:---:|:---:|:---:|:---:|:---:|:---:|:---:|:---:|:---:|:---:|:---:|:---:|:---:|
> > | MSE ($X_1$) | 0.281 | **0.268** | 0.286 | **0.268** | *0.277* | 0.278 | 0.294 | 0.283 | \| | 0.267 | - | - | - | - | - |
> > | MSE ($X_2$) | 0.198 | *0.194* | **0.192** | 0.195 | 0.194 | 0.209 | 0.195 | 0.196 | \| | 0.192 | - | - | - | - | - |
> > | MSE ($X_3$) | **0.095** | **0.095** | 0.099 | **0.095** | 0.096 | 0.107 | 0.098 | **0.095** | \| | 0.092 | - | - | - | - | - |
> > | MSE ($X_4$) | 0.394 | 0.389 | 0.400 | *0.385* | **0.383** | 0.407 | 0.388 | **0.383** | \| | 0.378 | - | - | - | - | - |
> > | MSE ($X_5$) | 0.111 | 0.109 | 0.109 | 0.116 | *0.106* | **0.105** | 0.111 | 0.124 | \| | 0.107 | - | - | - | - | - |
> > | MSE ($X_6$) | *0.008* | 0.009 | 0.009 | 0.010 | **0.007** | 0.011 | **0.007** | 0.012 | \| | 0.008 | - | - | - | - | - |
> > | MSE ($X_7$) | 0.078 | 0.072 | 0.080 | *0.070* | 0.079 | 0.093 | 0.083 | **0.068** | \| | 0.074 | - | - | - | - | - |
> > | MSE (All) | 0.166 | 0.162 | 0.168 | 0.163 | 0.163 | 0.173 | 0.168 | 0.166 | \| | 0.160 | 0.161 | 0.173 | 0.165 | 0.187 | 0.167 |
> >
> > ### ETTm1 -> ETTm2 (zero-shot)
> >
> > We conducted **zero-shot experiments** where ICTSP was trained on ETTm1 and evaluated on ETTm2. In this scenario, the **model parameters contain no fixed effects from the target dataset**. **Without using context examples**, the model relies **solely on its fixed weights learned from the source dataset** to perform temporal forecasting inference for each target token, resulting in a **significantly worse MSE** of 0.208 compared to other results.
> >
> > We can observe the performance imporvements when **introducing context tokens from $X_j$** corresponding to series $j$. Since the model parameters **lack prior knowledge of the channel structure of the target dataset**, the **improvement is most pronounced for the corresponding $X_j$ and diminish for other series**. This **task-specific improvement** originates from ICL, where the model uses a task's question-answer prompt to enhance performance on the specific task.
> >
> > Through these experiments, we effectively **disentangled the influence of learned channel structure** from the **effectiveness of ICL**. This demonstrates the **true reason** behind ICTSP's strong performance on zero-shot tasks.
> >
> > | Method | w/o Context | Context ($X_1$) | Context ($X_2$) | Context ($X_3$) | Context ($X_4$) | Context ($X_5$) | Context ($X_6$) | Context ($X_7$) | \| | Context (Full) | Time-LLM | GPT4TS | PatchTST | TimesNet | DLinear |
> > |---|:---:|:---:|:---:|:---:|:---:|:---:|:---:|:---:|:---:|:---:|:---:|:---:|:---:|:---:|:---:|
> > | MSE ($X_1$) | 0.320 | **0.279** | 0.287 | 0.284 | 0.282 | 0.287 | *0.280* | 0.290 | \| | 0.284 | - | - | - | - | - |
> > | MSE ($X_2$) | 0.226 | *0.204* | **0.203** | 0.207 | *0.204* | *0.204* | 0.205 | 0.213 | \| | 0.198 | - | - | - | - | - |
> > | MSE ($X_3$) | 0.141 | **0.100** | 0.105 | **0.100** | 0.102 | 0.103 | *0.101* | 0.111 | \| | 0.098 | - | - | - | - | - |
> > | MSE ($X_4$) | 0.440 | 0.436 | 0.435 | 0.446 | **0.429** | 0.437 | *0.432* | *0.432* | \| | 0.391 | - | - | - | - | - |
> > | MSE ($X_5$) | 0.144 | *0.109* | 0.112 | 0.111 | **0.108** | **0.108** | *0.109* | 0.129 | \| | 0.112 | - | - | - | - | - |
> > | MSE ($X_6$) | 0.048 | *0.008* | 0.009 | *0.008* | 0.009 | *0.008* | **0.007** | 0.025 | \| | 0.008 | - | - | - | - | - |
> > | MSE ($X_7$) | 0.140 | *0.076* | 0.081 | 0.085 | 0.080 | 0.079 | 0.079 | **0.074** | \| | 0.083 | - | - | - | - | - |
> > | MSE (All) | 0.208 | 0.173 | 0.176 | 0.177 | 0.174 | 0.175 | 0.173 | 0.182 | \| | 0.168 | 0.169 | 0.217 | 0.195 | 0.222 | 0.221 |
> >
> > ### ETTh1,ETTh2,ETTm1 -> ETTm2 (zero-shot)
> >
> > We further trained ICTSP on a combined dataset (ETTh1, ETTh2, ETTm1) containing more datapoints. Compared to the previous experiment, we observed two key improvements:
> >
> > 1. The model's **ability to use different context prompt tokens** to achieve corresponding performance improvements was **further enhanced**, resulting in **better overall performance** than in the previous experiment.
> > 2. The **task-specific performance improvement effect** became more pronounced for the **series corresponding to the provided context**.
> >
> > These results highlight the dominant role of ICTSP's ICL ability in achieving strong performance on zero-shot tasks.

---

> > > ### Author Response · Authors · 2024-11-25
> > >
> > > | Method | w/o Context | Context ($X_1$) | Context ($X_2$) | Context ($X_3$) | Context ($X_4$) | Context ($X_5$) | Context ($X_6$) | Context ($X_7$) | \| | Context (Full) |
> > > |---|:---:|:---:|:---:|:---:|:---:|:---:|:---:|:---:|:---:|:---:|
> > > | MSE ($X_1$) | 0.317 | **0.279** | 0.286 | *0.284* | 0.287 | 0.290 | 0.287 | 0.285 | \| | 0.278 |
> > > | MSE ($X_2$) | 0.225 | 0.207 | **0.196** | 0.206 | 0.202 | 0.204 | *0.198* | *0.198* | \| | 0.200 |
> > > | MSE ($X_3$) | 0.116 | 0.101 | 0.107 | **0.099** | 0.105 | *0.100* | 0.101 | 0.102 | \| | 0.098 |
> > > | MSE ($X_4$) | 0.442 | 0.432 | *0.413* | 0.439 | **0.410** | 0.431 | 0.418 | 0.414 | \| | 0.392 |
> > > | MSE ($X_5$) | 0.126 | 0.110 | 0.112 | 0.109 | 0.110 | **0.107** | 0.112 | *0.108* | \| | 0.107 |
> > > | MSE ($X_6$) | 0.013 | *0.008* | 0.013 | *0.008* | 0.010 | *0.008* | **0.007** | *0.008* | \| | 0.009 |
> > > | MSE ($X_7$) | 0.085 | 0.083 | 0.088 | *0.079* | 0.083 | *0.079* | 0.084 | **0.073** | \| | 0.082 |
> > > | MSE (All) | 0.189 | 0.174 | 0.174 | 0.175 | 0.172 | 0.174 | 0.173 | 0.170 | \| | 0.167 |
> > >
> > > ### Summary
> > >
> > > Thank you once again for your questions. If you have any further inquiries, we would be more than happy to address them and look forward to receiving your positive feedback on our responses. We truly appreciate the time and effort you have invested in reviewing our work. Thank you again!

---

> ### Author Response · Authors · 2024-11-30
>
> Dear Reviewer H6bM,
>
> Thank you for your valuable follow-up questions earlier. We have provided additional responses to your concerns above. With approximately two days remaining until the end of the ICLR rebuttal period, we would like to know if the information provided resolves your concerns or if you have any further questions. If so, we would be happy to provide a quick answer.
>
> If you still have questions about why our model can demonstrate ICL capabilities without relying on very large-scale LLM parameters, we have addressed this in our responses to other reviewers. You may refer to the following reply for more details: [https://openreview.net/forum?id=dCcY2pyNIO&noteId=FORo1hmDUB](https://openreview.net/forum?id=dCcY2pyNIO&noteId=FORo1hmDUB).
>
> Once again, we deeply appreciate your thoughtful questions, which have significantly helped us better present our research. Thank you for the time and effort you’ve devoted to the review process. If you are currently on holiday, we also wish you a wonderful break.
>
> Best regards,
> Authors of Submission #12786

---

> ### Comment · Reviewer_H6bM · 2024-12-01
>
> Thanks for the authors' response. Though I am still not fully convinced that the additional experiments are sufficient to claim the benefits of ICL in time series, they indeed provide more interesting observations. I am changing my score accordingly.

---

> > ### Author Response · Authors · 2024-12-02
> >
> > Dear Reviewer H6bM,
> >
> > We sincerely appreciate your positive recognition of our work. We are also deeply grateful for the constructive questions and suggestions you provided during the review process. Your valuable insights have been very helpful in improving the quality of presenting the core content of our paper. Once again, thank you for your thoughtful feedback and the time you spent in reviewing our work!

---

### Official Review · Reviewer_NX33 · 2024-11-02

**Soundness:** 2
**Presentation:** 3
**Contribution:** 3
**Rating:** 8
**Confidence:** 4

**Summary:**

This paper presents the In-context Time Series Predictor(ICTSP), which utilizes the in-context capabilities for time series forecasting tasks. It proposes an innovative way to construct the tokens with context examples, showing promising results across multiple experimental settings.

**Strengths:**

1. The writing is clear and fluent.
2. This paper's motivation is relatively novel. It proposes utilizing the in-context learning capabilities of large language models (LLMs) for time series forecasting tasks.
3. The experiments are comprehensive, encompassing various experimental settings and datasets.

**Weaknesses:**

1. The formulation of TSF transformers is not sufficient. As far as I am concerned, there are some methods [1,2] that utilize patching embedding to form the input tokens. These methods cannot be simply categorized as Temporal-wise Transformer or Series-wise Transformer in Section 2.2

2. The baselines results for baselines are collected from source papers. However, this paper applies different input time series length from these papers, which may lead to an unfair comparison.

3. The details of Computational Costs and Table.5 are not provided.


[1] Nie, Yuqi, et al. "A Time Series is Worth 64 Words: Long-term Forecasting with Transformers." The Eleventh International Conference on Learning Representations.

[2] Jin, Ming, et al. "Time-LLM: Time Series Forecasting by Reprogramming Large Language Models." The Twelfth International Conference on Learning Representations.

**Questions:**

Based on the listed weakness,
1. Can you add more discussions for these methods based on patching embedding from an ICL perspective?
2. For the Full-data TSF setting, can you rerun the baselines under the same experimental settings?  Alternatively, you could articulate the rationale behind your experimental settings, which would make the comparisons more convincing.
3. Can you provide more details for Computational Costs? What are the number of layers and hidden dimensions of different models?

---

> ### Author Response · Authors · 2024-11-20
>
> ```
> 1. Can you add more discussions for these methods based on patching embedding from an ICL perspective?
> ```
>
> Thank you for highlighting the core content of our paper. From the ICL perspective, explaining how patch embedding improves Temporal-wise Transformers is a key part of our work, elaborated in Section 2.3. We view patch embedding as an enhancement to Temporal-wise tokenization: when applied across all series, it corresponds to Figure 3(d); when applied independently to each series, it combines the methods in Figures 3(b) and 3(d).
>
> We argue that the performance improvement of the **patching method** in Figure 3(d) stems from its ability to **introduce more learnable temporal-wise relationships directly within the tokenization**. This avoids relying solely on attention layers to construct these relationships algorithmically.
>
> #### Patch embedding enhances learnable temporal relationships in timestep embedding while preserving autoregressive token structure unlike series embedding
>
> In real-world multivariate time series forecasting, temporal-wise dependencies are generally more reliable, while series-wise relationships are often weak or even absent. Vanilla temporal-wise timestep tokenization fixes potentially non-existent series relationships in token representations and leaves critical temporal relationships to be dynamically constructed by attention layers, which is suboptimal. Patching addresses this by incorporating both local temporal-wise and series-wise relationships into tokens, naturally emphasizing stronger temporal relationships inside the token and reducing overfitting to non-existent series-wise effects in the input context.
>
> Similarly, PatchTST applies **channel-independent tokenization** alongside patching, explicitly dropping inter-series connections. This forces the model to focus only on temporal relationships, further reducing overfitting on real-world datasets.
>
> Figure 4 illustrates this with two examples. On a synthetic dataset (`Multi`) with weak intra-series temporal relationships but strong inter-series (unaligned shifted) dependencies, the vanilla Temporal-wise Transformer performs unusually well, contrasting with its behavior on real-world datasets. However, on a real-world dataset (`ETTm2`), where temporal relationships dominate and series-wise dependencies are weaker and more volatile, Temporal-wise Transformers perform poorly compared to models like Series-wise Transformer and ICTSP, which prioritize temporal relationships in tokenization.
>
> Although patch embeddings blend characteristics of Temporal-wise and Series-wise Transformers, they are categorized as Temporal-wise Transformers in the paper because their tokenization segments along the temporal dimension, preserving the **next-token prediction (autoregressive) structure**, where the **next token is the "answer" to the previous token**. **Adding patch embeddings does not disrupt this relationship between tokens, whereas series embedding methods completely break this structure**.
>
> ```
>     2. For the Full-data TSF setting, can you rerun the baselines under the same experimental settings? Alternatively, you could articulate the rationale behind your experimental settings, which would make the comparisons more convincing.
> ```
>
> Thank you for this meaningful question. First, the choice of input length \(L_I\) is widely discussed in the literature, with different models often requiring different optimal lengths depending on the dataset. For example:
>
> - **GPT4TS** [1] (Section H.10) suggests using the optimal \(L_I\) reported in original papers rather than a single \(L_I\) for all baselines, as longer lookbacks often cause overfitting in some models, resulting in unfair comparisons.
> - Studies like **TiDE** [2] (Section 5.1) selects the best \(L_I\) from multiple tested lengths to ensure fair comparisons.
> - **DLinear** [3] (Figure 6), **PatchTST** [4] (Table 9), and **iTransformer** [5] (Figure 6) show that for many real-world datasets, shorter \(L_I\) often performs better as longer lookbacks can lead to overfitting.
>
> Thus, effectively utilizing longer \(L_I\) while avoiding overfitting remains a common challenge.
>
> ICTSP addresses this by treating segments of length \(L_b + L_P\) as forecasting task tokens and creating \(N = L_I - (L_b + L_P)\) task tokens to incorporate lookback information. With \(L_b = 512\), each token represents a forecasting task used by models like PatchTST or GPT4TS. ICTSP can be viewed as leveraging past forecasting task datapoints to optimize the predictor for the current forecasting task, enhancing generalization performance.

---

> ### Author Response · Authors · 2024-11-20
>
> While we believe this rationale supports our approach, **we are happy to conduct an ablation study using only 512-step input information**. However, this configuration disrupts ICTSP's **forecasting tasks as context** structure and is **not recommended for practical use**. ICTSP requires \(L_I > L_b + L_P\) to split the input into multiple ground truth forecasting tasks. Given \(L_b = 512\), when \(L_P = 720\), \(L_I > 512 + 720 = 1232\) is necessary to construct at least one ground truth forecasting task context token.
>
> ### Ablation Study: ICTSP with limited lookback information
>
> To fully align with the baseline models using \(L_I = 512\), we **relax ICTSP's design**, which originally required all context tokens to be ground truth forecasting tasks. Instead, we **allow some forecasting tasks to include incomplete information**. Specifically, we retain \(L_I = 1440\) but **mask the first \(1440 - 512 = 928\) timesteps in the input**, ensuring **only the last 512 steps contain true information**. The masked portion is **filled with the mean value** of each series token **from the last 512 steps**. To maintain the presence of some true forecasting task tokens for shorter \(L_P\), we shorten \(L_b\) to 336. All other hyperparameters remain unchanged. This adjusted structure is denoted as **ICTSP (Limited)**.
>
> We repeated experiments on the ETTm2 and Weather datasets, covering full data, few-shot 10\% (FS10\%), few-shot 5\% (FS05\%), and zero-shot (ZS) settings for the ETT datasets. Full results are available in Table 11 of the revised paper. Below is the summary of average MSE and average rank compared to six strong baselines.
>
> | Methods | ICTSP | ICTSP (Limited) | Time-LLM | GPT4TS | PatchTST | TimesNet | DLinear |
> |---|:-:|:-:|:-:|:-:|:-:|:-:|:-:|
> | Full ETTm2 (Avg) | **0.247** | 0.250  | 0.251 | 0.267 | 0.255 | 0.291 | 0.267 |
> | Full Weather (Avg) | **0.218** | 0.220 | 0.226 | 0.237 | 0.226 | 0.259 | 0.249 |
> | FS10% ETTm2 (Avg) | 0.275 | **0.274** | 0.277 | 0.293 | 0.296 | 0.320 | 0.316 |
> | FS10% Weather (Avg) | 0.237 | 0.238 | **0.234** | 0.238 | 0.242 | 0.279 | 0.241 |
> | FS05% ETTm2 (Avg) | 0.285 | 0.281 | **0.274** | 0.308 | 0.315 | 0.399 | 0.345 |
> | FS05% Weather (Avg)  | **0.256** | **0.256** | 0.261 | 0.264 | 0.270 | 0.264 | 0.298 |
> | ZS m1→m2 (Avg) | **0.255** | 0.258 | 0.265 | 0.314 | 0.297 | 0.322 | 0.335 |
> | ZS h1→h2 (Avg) | **0.337** | 0.339 | 0.352 | 0.407 | 0.381 | 0.421 | 0.494 |
> | AvgRank | **1.66** | 1.97 | 2.66 | 4.56 | 4.34 | 6.31 | 6.19 |
>
> From the average rankings, we observe that even with masking applied to information outside the 512-step window, **ICTSP (Limited) still significantly outperforms all baselines**. In full data and zero-shot experiments, ICTSP (Limited) performs slightly weaker than the original ICTSP but remains superior to all baselines. However, in few-shot training experiments, ICTSP (Limited) surpasses the original ICTSP. It is important to note that this improvement is **due to an additional constraint in the original ICTSP**, which, despite using $L_I=1440$, imposes **zero-filling** restrictions outside of $L_b=512$ to ensure fair comparison with baselines using $L_I=512$ (see Section A.3.3 Discussion of fair comparison for details on **few-shot experiments**).
>
> In the full data and zero-shot experiments, all data can be used with any $L_I$ without causing data leakage. However, in few-shot experiments, due to skipping in data sampling, a longer \(L_I\) accesses more lookback data during validation/testing, giving an unfair advantage. To address this, the original ICTSP applied **zero-masking** to restrict access to only 512 valid timesteps at the start of validation/testing, gradually expanding to \(L_I = 1440\) as more valid data became available.
>
> Nonetheless, zero-filling in forecasting task tokens caused issues because of the **last-value demeaning operation** used on the forecasting task tokens, resulting in meaningless negative last-value placeholders and reduced model performance. **ICTSP (Limited)** resolves this by using **mean-value filling**, ensuring more reasonable values in forecasting task tokens. This adjustment allows ICTSP (Limited) to outperform the original ICTSP in few-shot experiments.

---

> ### Author Response · Authors · 2024-11-20
>
> To clarify this, we **replaced the zero-filling used for few-shot fair comparison in the original ICTSP with mean-filling** and present the results below. ICTSP with mean-filling **significantly outperforms** all models, including Time-LLM, across all experiments. This highlights that ICTSP’s few-shot capability is inherently **stronger** than original results presented in the paper when proper mean-filling strategies are applied. Full results can be found in Table 12 of the revised paper.
>
> | Methods | ICTSP (Zero-filling) | ICTSP (Mean-filling) | ICTSP (Limited) | Time-LLM | GPT4TS | PatchTST | TimesNet | DLinear |
> |---|:-:|:-:|:-:|:-:|:-:|:-:|:-:|:-:|
> | FS10% ETTm2 (Avg) | 0.275 | **0.264** | 0.274 | 0.277 | 0.293 | 0.296 | 0.320 | 0.316 |
> | FS10% Weather (Avg) | 0.237 | **0.234** | 0.238 | **0.234** | 0.238 | 0.242 | 0.279 | 0.241 |
> | FS05% ETTm2 (Avg) | 0.285 | **0.272** | 0.281 | 0.274 | 0.308 | 0.315 | 0.399 | 0.345 |
> | FS05% Weather (Avg)  | 0.256 | **0.248** | 0.256 | 0.261 | 0.264 | 0.270 | 0.264 | 0.298 |
>
>     3. Can you provide more details for Computational Costs? What are the number of layers and hidden dimensions of different models?
>
> Thank you for your attention to the details of our paper. In Table 5, we used the reported default settings from previous studies for the baselines. For Autoformer, Informer, PatchTST, and iTransformer, we set \(L_I = 512\) with 3 Transformer layers (or two encoder layers and one decoder layer). We used the number of series \(C = 7\) and dimension \(d = 128\). For ICTSP, we followed the settings described in the first paragraph of Section 3, which include the same number of layers and hidden dimensions with \(L_I=1440\). We calculated the computational costs using the Python `ptflops` package.
>
> Lastly, we greatly appreciate your feedback on our paper. If you have any more questions, we are more than happy to provide further explanations and look forward to engaging in more discussions with you. Thank you very much!
>
> ### References
>
> [1] Zhou, Tian, et al. "One fits all: Power general time series analysis by pretrained lm." Advances in neural information processing systems 36 (2023): 43322-43355.
>
> [2] Das, Abhimanyu, et al. "Long-term Forecasting with TiDE: Time-series Dense Encoder." Transactions on Machine Learning Research.
>
> [3] Zeng, Ailing, et al. "Are transformers effective for time series forecasting?." Proceedings of the AAAI conference on artificial intelligence. Vol. 37. No. 9. 2023.
>
> [4] Nie, Yuqi, et al. "A Time Series is Worth 64 Words: Long-term Forecasting with Transformers." The Eleventh International Conference on Learning Representations.
>
> [5] Liu, Yong, et al. "iTransformer: Inverted Transformers Are Effective for Time Series Forecasting." The Twelfth International Conference on Learning Representations.

---

> > ### Comment · Reviewer_NX33 · 2024-11-26
> >
> > Thanks for the detailed response, which addressed most of my concerns. However, I noticed that the model used in this paper is relatively lightweight, with 3 layers and dimension h=128. We generally take large language models as examples for in-context learning, which are pre-trained on extensive datasets.  In this paper, the model is trained from scratch solely on TSF data and has a relatively small number of parameters, how can you ensure that it possesses in-context learning capabilities?

---

> > > ### Author Response · Authors · 2024-11-26
> > >
> > > ```
> > > However, I noticed that the model used in this paper is relatively lightweight...
> > > ```
> > > Thank you for raising this valuable question. I believe this is a common point of confusion for any researcher looking to explore this direction further.
> > >
> > > First, our work is part of a series of studies focused on how Transformers can be trained from scratch on specific problem subclasses, function classes, or data types and show ICL capabilities to effectively demonstrate generalization on new data with covariate shifts by leveraging provided context examples.  While ICL was originally discovered and discussed in the context of trained large LLMs, these more recent studies explore the ICL capabilities of Transformers in specific problem settings, function classes, or dataset types. This is reflected in the ICL formulation mentioned in line 106 of our paper.
> > >
> > > In the references cited in our paper and below, [1,2] train Transformers from scratch on OLS data, [3,4] train Transformers dynamical system data, [5,6] train Transformers on linear classification data, [7] train Transformer on tabular classification data, and [8] train Transformer on univariate time series data, among others.
> > >
> > > In this context, the hyperparameter settings for ICL Transformers are more dependent on the specific data type or problem setting being explored. For example:
> > >
> > > - [1] includes settings such as `#Layers=3, #Heads=2, and #EmbDim=64`, labeled as `Tiny`. The study demonstrates that even with these `Tiny` settings, Transformers can achieve performance comparable to larger models across many task types and even reach optimum error on certain tasks.
> > > - [3] shows that using a `Tiny` setting with `#Layers=3` does not alter the Transformer’s behavior in ICL generalization.
> > > - [4] demonstrates that Transformers with different #Layers can approximate the behavior of different linear models: `single-layer` Transformers closely match a single step of gradient descent, `2-4 layer` Transformers are closer to Ridge Regression, and Transformers with `more layers` approximate OLS.
> > > - [2] explicitly proves that for ICL to perform Ridge Regression, a mild bound of `#Layers ≤ 3` is required. Similarly, for LASSO, `#Layers ≤ 2 and #EmbDim ≤ 2d` are required, where \(d\) is the number of input features in the regression problem.
> > > - During model training, [5] employs `#Layers=3` as the default setting. And in [9], which explore the ICL univariate time series forecasting which is close to our domain, even use a `#Layers=2` setting.
> > >
> > > In summary, previous ICL Transformer literature highlights the importance of selecting an appropriate depth for the specific problem and even suggests that certain ICL behaviors may require an upper bound on depth. Therefore, for multivariate time series data, which demands significant regularization, we believe that using the default setting of `#Layers=3` is a reasonable choice.
> > >
> > > Additionally, I would like to provide **more evidence of the ICL abilities exhibited by ICTSP**. To avoid taking up too much space in this page, you might consider, if you have enough time, reading my response to another reviewer ([https://openreview.net/forum?id=dCcY2pyNIO&noteId=nlTS1YoSgK](https://openreview.net/forum?id=dCcY2pyNIO&noteId=nlTS1YoSgK)), specifically the section titled `Ablation Study: Understanding the Specific Contributions of Context Examples`.
> > >
> > > If you have further questions, we would be happy to provide additional clarification. Once again, thank you for the time and effort in reviewing our paper.
> > >
> > > ```
> > > References
> > > [1] Garg, Shivam, et al. "What can transformers learn in-context? a case study of simple function classes." Advances in Neural Information Processing Systems 35 (2022): 30583-30598.
> > > [2] Bai, Yu, et al. "Transformers as statisticians: Provable in-context learning with in-context algorithm selection." Advances in neural information processing systems 36 (2024).
> > > [3] Li, Yingcong, et al. "Transformers as algorithms: Generalization and stability in in-context learning." International Conference on Machine Learning. PMLR, 2023.
> > > [4] Akyürek, Ekin, et al. "​​ What learning algorithm is in-context learning? Investigations with linear models." The Eleventh International Conference on Learning Representations. 2022.
> > > [5] Müller, Samuel, et al. "Transformers can do bayesian inference." arXiv preprint arXiv:2112.10510 (2021).
> > > [6] Hollmann, Noah, et al. "Tabpfn: A transformer that solves small tabular classification problems in a second." arXiv preprint arXiv:2207.01848 (2022).
> > > [7] Dooley, Samuel, et al. "Forecastpfn: Synthetically-trained zero-shot forecasting." Advances in Neural Information Processing Systems 36 (2024).
> > > [8] Edelman, Benjamin L., et al. "Inductive biases and variable creation in self-attention mechanisms." International Conference on Machine Learning. PMLR, 2022.
> > > [9] Dooley, Samuel, et al. "Forecastpfn: Synthetically-trained zero-shot forecasting." Advances in Neural Information Processing Systems 36 (2024).
> > > ```

---

> > > > ### Comment · Reviewer_NX33 · 2024-11-26
> > > >
> > > > Thanks for the authors' feedback. My concerns are addressed, and I am happy to raise my score. I believe these discussions would make the paper more comprehensive.

---

> > > > > ### Author Response · Authors · 2024-12-02
> > > > >
> > > > > Dear Reviewer NX33,
> > > > >
> > > > > We are truly grateful for your high evaluation of our work. We sincerely appreciate the thoughtful questions and suggestions you provided during the review process. We are pleased to have incorporated these valuable inputs into our paper. Once again, thank you for your valuable feedback and the time you dedicated to reviewing our paper!

---

### Official Review · Reviewer_hkWk · 2024-11-02

**Soundness:** 2
**Presentation:** 2
**Contribution:** 2
**Rating:** 3
**Confidence:** 4

**Summary:**

The paper introduces a model named "In-context Time Series Predictor" (ICTSP), designed specifically for time series forecasting (TSF). Unlike traditional Transformer-based models, ICTSP adopts a novel input token structure, resulting in improved performance.

**Strengths:**

ICTSP model transforms time series forecasting tasks into input tokens, aligning closely with the inherent mechanisms of the Transformer model and efficiently leveraging its contextual learning capabilities.

**Weaknesses:**

1. The innovation appears limited. While the paper extensively discusses the approach from an in-context learning perspective, the methodology itself seems simplistic and more akin to a heuristic trick due to excessive textual explanation without solid theoretical support.

2. The paper does not address whether this model could be applied to other time series tasks, such as classification, interpolation, or anomaly detection. The utility of the model appears confined to the forecasting tasks, necessitating stronger performance to justify its specialized use.

3. The significant improvements in zero-shot tasks over full-data tasks lack detailed explanations. The paper does not sufficiently clarify why the ICTSP demonstrates a pronounced advantage in few-shot and zero-shot learning scenarios than in full-data tasks

**Questions:**

The provided link to the code repository is inaccessible, which hinders the reproducibility of the results. Could you please update or fix the link to the source code?

---

> ### Author Response · Authors · 2024-11-20
>
> The provided link to the code repository is inaccessible, which hinders the reproducibility of the results. Could you please update or fix the link to the source code?
>
> We sincerely hope the reviewer can confirm again whether the code repository we provided is accessible. We have contacted several individuals for testing, and the feedback we received consistently indicates that the link [https://anonymous.4open.science/r/ICTSP-C995](https://anonymous.4open.science/r/ICTSP-C995) is working properly. Since we uploaded the repository to this link on August 7, 2024, we have not made any changes to it. Therefore, if there were any instances where it was inaccessible, it may have been due to temporary website maintenance or network fluctuations.
>
> We would like to highlight that `anonymous.4open.science` offers a highly useful modification tracking feature. When you open the link and check the bottom left corner of the website, you will see `Last Update: Aug 7, 2024`. This indicates that we have not made any modifications to the repository after this date (e.g., temporarily closing the repository). If any changes were made to the link, such as rewriting code, adding, or deleting files, the `Last Update` date would reflect the most recent change. Moreover, if the anonymized link is closed and reopened under the same name, the `Last Update` would display the new link’s creation date (not the last commit date from the original GitHub repository). This platform is widely used for anonymizing repositories, and we encourage you to verify these features yourself.
>
> Therefore, from August 7, 2024, before we submitted the paper, to the discussion period, we have not made any modifications to the anonymized repository, and it should have remained accessible. If you still find the link inaccessible, we have provided an alternative link [https://anonymous.4open.science/r/ICTSP-ALT-6BE4](https://anonymous.4open.science/r/ICTSP-ALT-6BE4), which contains the same code as the original repository for your reference. Additionally, we have provided a repository for your questions regarding Weakness 2, which demonstrates ICTSP’s application to other time series tasks: [https://anonymous.4open.science/r/ICTSP-MultiTask-53B3](https://anonymous.4open.science/r/ICTSP-MultiTask-53B3). We will explain its contents in our response to Weakness 2 below.

---

> ### Author Response · Authors · 2024-11-20
>
> ```
> 1. The innovation appears limited. While the paper extensively discusses the approach from an in-context learning perspective, the methodology itself seems simplistic and more akin to a heuristic trick due to excessive textual explanation without solid theoretical support.
> ```
>
> We appreciate the reviewer’s valuable feedback. While the reviewer noted a lack of solid theoretical support in our paper, we want to clarify that our work and experiments are grounded in prior ICL studies with well-established theoretical foundations. Here are key proven conclusions from the cited ICL research that directly support applying ICL to time series forecasting in our paper:
>
> 1. Transformers can learn linear predictors ([1,2,3,4])
>    This enables Transformers to perform fundamental linear time series forecasting predictors through ICL, such as NLinear [5] and RLinear [8].
>
> 2. A single Transformer layer can implement simple operations, including shifting input features, computing differences between input features, and transferring values from one token to a target token ([2,4,6])
>    This allows Transformers to process the autocorrelated structure of multivariate time series inputs appropriately.
>
> 3. Transformers can learn function classes, including shallow MLPs and decision trees ([3,4,6])
>    This enables Transformers to perform the core MLP predictor used like recent MLP-based time series forecasting models [7,9,10] through ICL.
>
> 4. Transformers can select the best-performing algorithm based on input context pairs ([3,4,6])
>    This allows Transformers to dynamically choose the most optimal predictor type based on the properties of the input data, such as the strength of temporal relationships, aligned series-wise relationships, and unaligned series-wise relationships.
>
> These proven theoretical results underpin all discussions in our paper. In the revision, we highlight more about the links of these findings to statements in our work.
>
> ```
> References 1
>
> [1] Zhang, Ruiqi, Spencer Frei, and Peter Bartlett. "Trained Transformers Learn Linear Models In-Context." R0-FoMo: Robustness of Few-shot and Zero-shot Learning in Large Foundation Models.
> [2] Akyürek, Ekin, et al. "​​ What learning algorithm is in-context learning? Investigations with linear models." The Eleventh International Conference on Learning Representations. 2022.
> [3] Li, Yingcong, et al. "Transformers as algorithms: Generalization and stability in in-context learning." International Conference on Machine Learning. PMLR, 2023.
> [4] Bai, Yu, et al. "Transformers as statisticians: Provable in-context learning with in-context algorithm selection." Advances in neural information processing systems 36 (2024).
> [5] Zeng, Ailing, et al. "Are transformers effective for time series forecasting?." Proceedings of the AAAI conference on artificial intelligence. Vol. 37. No. 9. 2023.
> [6] Guo, Tianyu, et al. "How Do Transformers Learn In-Context Beyond Simple Functions? A Case Study on Learning with Representations." The Twelfth International Conference on Learning Representations.
> [7] Ekambaram, Vijay, et al. "Tsmixer: Lightweight mlp-mixer model for multivariate time series forecasting." Proceedings of the 29th ACM SIGKDD Conference on Knowledge Discovery and Data Mining. 2023.
> [8] Li, Zhe, et al. "Revisiting long-term time series forecasting: An investigation on linear mapping." arXiv preprint arXiv:2305.10721 (2023).
> [9] Das, Abhimanyu, et al. "Long-term Forecasting with TiDE: Time-series Dense Encoder." Transactions on Machine Learning Research.
> [10] Wang, Shiyu, et al. "Timemixer: Decomposable multiscale mixing for time series forecasting." arXiv preprint arXiv:2405.14616 (2024).
> ```

---

> ### Author Response · Authors · 2024-11-20
>
> However, we respectfully disagree with the reviewer’s comment that *"the methodology itself seems simplistic and more akin to a heuristic trick due to excessive textual explanation without solid theoretical support."* We believe this view may not reflect a consensus in the AI/ML community. Below, we provide examples from AI research, including time series and ICL domains, to illustrate this point.
>
> While the ICL field includes studies with rigorous theoretical proofs, as we highlighted earlier, key papers we cite, such as [1,2,3], do not include theoretical derivations or proofs. Instead, they rely on textual explanations and discussions of empirical results. Despite this, they have had significant impact, offering valuable insights for further theoretical and empirical research.
>
> AI researchers often prioritize simple, intuitive, and effective methods that provide meaningful insights. In time series forecasting, works like DLinear [4], PatchTST [5], GPT4TS [6], and iTransformer [7] show how existing structures can achieve broad impact by being transfered and applied to time series tasks. If these works are considered to lack novelty and solid support, even the methods they build upon, like ViT [8] and GPT-2 [9], could be similarly critiqued. Yet, these foundational contributions are widely recognized.
>
> In the following section, we aim to help identify the novelty of our research more clearly.
>
> ```
> References 2
>
> [1] Min, Sewon, et al. "Rethinking the Role of Demonstrations: What Makes In-Context Learning Work?." EMNLP. 2022.
> [2] Garg, Shivam, et al. "What can transformers learn in-context? a case study of simple function classes." Advances in Neural Information Processing Systems 35 (2022): 30583-30598.
> [3] Shi, Zhenmei, et al. "Why Larger Language Models Do In-context Learning Differently?." R0-FoMo: Robustness of Few-shot and Zero-shot Learning in Large Foundation Models.
> [4] Zeng, Ailing, et al. "Are transformers effective for time series forecasting?." Proceedings of the AAAI conference on artificial intelligence. Vol. 37. No. 9. 2023.
> [5] Nie, Yuqi, et al. "A Time Series is Worth 64 Words: Long-term Forecasting with Transformers." The Eleventh International Conference on Learning Representations.
> [6] Zhou, Tian, et al. "One fits all: Power general time series analysis by pretrained lm." Advances in neural information processing systems 36 (2023): 43322-43355.
> [7] Liu, Yong, et al. "iTransformer: Inverted Transformers Are Effective for Time Series Forecasting." The Twelfth International Conference on Learning Representations.
> [8] Alexey Dosovitskiy, et al. "An Image is Worth 16x16 Words: Transformers for Image Recognition at Scale." International Conference on Learning Representations. 2021.
> [9] Radford, Alec, et al. "Language models are unsupervised multitask learners." OpenAI blog 1.8 (2019): 9.
> ```
>
> ### About the novelty of this research
>
> First, we would like to mention that the other 3 reviewers identified the novel ideas in our paper as key strengths. To clarify the innovations, we summarize our contributions below:
>
> - **Novel Purpose**
>   We explore activating the ICL capabilities of Transformers for TSF solely through model design, without relying on large-scale LLM pretrained weights. This approach enhances generalization and improves few-shot and zero-shot performance, a direction not previously explored for TSF Transformers.
>
> - **Novel Findings**
> Using the ICL perspective, we systematically analyze Transformer-based TSF models, categorize them, identify key issues, and provides a series of findings and insights:
>   a) **Temporal-wise Transformers** underperform due to overemphasizing channel-wise dynamics in the input context while neglecting temporal-wise relationships, leading to poor generalization on real-world datasets with weak series dependencies.
>   b) **Series-wise Transformers** can model both temporal-wise and aligned series-wise relationships. However, they lack context inputs with shifting structures, making them less effective at handling unaligned series-wise relationships.
>   c) Components like **channel independence**, **random dropping**, and **patching** enhance Temporal-wise Transformers by encourage tokenization focusing more on temporal-wise relationships.
>
> These findings, supported by evidence from prior work and our experiments, have not been described in prior literature and offer valuable directions for future research.

---

> ### Author Response · Authors · 2024-11-20
>
> - **Solving Unresolved Problems**
>
> Building on the issues identified in previous models, we developed ICTSP, a structure capable of addressing all these problems, which are crucial for constructing a multivariate TSF model with strong generalization capabilities. ICTSP effectively resolves the following challenges:
>
> 1. Modeling **temporal-wise dependencies** (tokenization along the temporal dimension).
> 2. Modeling **aligned series-wise dependencies** (input series in tokenization).
> 3. Modeling **unaligned series-wise dependencies** (introducing a shifted forecasting task tokenization structure in the context examples).
>    (These first three points are supported by the main experiments and Figure 4.)
> 4. Avoiding **overfitting to unstable channel-wise effects** caused by context timestep tokens (segment tokenization ensures temporal relationships outweigh weak series relationships in the representation for real-world data).
> 5. Offering a **flexible input channel structure** that accommodates any number of input series without requiring modifications to the model structure (forecasting tasks as tokens).
> 6. Enabling seamless transfer to **datasets from different domains** without additional training (using historical forecasting tasks as context examples, the model can automatically and implicitly fit the most suitable predictor).
>
> Previous Transformer-based TSF models generally fail to address at least two of these challenges:
> iTransformer: (3, 6), Time-LLM: (2, 3), GPT4TS: (2, 3), ForecastPFN: (2, 3), Crossformer: (5, 6), PatchTST: (2, 3, 6), FEDFormer: (4, 5, 6), Autoformer: (1, 4, 5, 6), Informer: (1, 4, 5, 6).
>
> - **Insightful Presentation**
>
> Reviewer uZY9 noted that our research offers a novel perspective for categorizing and summarizing prior models while showcasing ICTSP's innovations. We fully agree. To clarify further, we propose summarizing the content of Section 2 in the following table, which clearly shows how the ICL perspective identifies issues in previous models and addresses them with appropriate ICTSP structure.
>
> | **Methods** | **Tokenization** | **ICL (x, y) Pair** | **Token Representation for** | **Attention to Learn** | **Suited For** | **Not Suited For** | **ICL Context Examples Describing** | **Channel Structure** | **Problem** |
> |:-:|:-:|:-:|:-:|:-:|:-:|:-:|:-:|:-:|:-:|
> | **Temporal-wise Transformer** | Timestep embedding | (Timestep index, Multi-channel values) | Single-step inter-series dependencies | Temporal-wise dynamics of series dependencies | Strong and consistent inter-series dependencies | Weak inter-series dependencies in real-world datasets | Underlying inter-series effects vary with the time indices in the context lookback (likely nonexistent or weak) | Fixed | Only series relationships in tokenization; risks overfitting to series-wise effects from the input context |
> | **Temporal-wise Transformer (with Channel Independence)** | Scalar embedding | (Timestep index, Single-channel value) | Scalar value at a single timestep | Temporal-wise dependencies | Strong and fast-changing temporal dependencies | Strong inter-series dependencies | Temporal relationships within the context for one series | Fixed | Unable to model series-wise relationships |
> | **Temporal-wise Transformer (with Patching)** | Patch embedding | (Timestep index, Multi-channel values in a temporal interval) | Temporal-wise and series-wise relationships in a local interval | Temporal-wise relationships between local intervals | Temporal and series relationships can be automatically balanced by this structure | Few-shot/zero-shot cases with different series relationship characteristics | Temporal-wise relationships between local intervals | Fixed | Fixed channel structure in tokenization prevents generalization to datasets with different structures |
> | **Series-wise Transformer** | Series embedding | (Input series, Output series) | Temporal-wise relationships within a single series input | Aligned inter-series relationships | Real-world datasets with strong temporal-wise and aligned series-wise relationships (e.g., traffic) | Unaligned series-wise relationships, such as shifting effects between series | No context examples | Flexible but requires additional training to adapt to datasets with new channel structure | Restricted to specific dataset channel structures; unable to learn unaligned series-wise effects |
> | **ICTSP** | Embedding of forecasting tasks | (Input series, Output series) | Temporal-wise relationships in a forecasting task | Forecasting task predictors, considering both aligned and unaligned inter-series relationships in shifted context examples | Datasets with fast-changing temporal relationships, aligned or unaligned series relationships; transfering to new datasets with different channel structures | / | Historical ground truth forecasting tasks as references | Flexible; additional training is not necessary since context examples exist  | / |

---

> ### Author Response · Authors · 2024-11-20
>
> ```
> 2. The paper does not address whether this model could be applied to other time series tasks, such as classification, interpolation, or anomaly detection. The utility of the model appears confined to the forecasting tasks, necessitating stronger performance to justify its specialized use.
> ```
>
> Thank you for raising this point. The ICTSP structure is highly transferable to multi-task time series analysis. By simply replacing the forecasting task’s `(lookback, future)` structure with `(lookback, new_target)`, it can adapt to various tasks. In our follow-up work, we pretrained an ICL model capable of solving multiple tasks by dynamically identifying the task type from context examples. For your reference, we provide the following link: [https://anonymous.4open.science/r/ICTSP-MultiTask-53B3](https://anonymous.4open.science/r/ICTSP-MultiTask-53B3). This repository demonstrates ICTSP’s application to tasks like forecasting, classification, imputation, cropping, shifting, hyper-resolution (up-sampling), moving average, differencing, and exponential smoothing. It includes data generation scripts (`ts_generation/`) and in-context tokenization for multi-task learning (`data_provider/`). However, we must clarify that discussing multi-task learning is beyond the main focus of this paper.
>
> ### Why doesn’t this paper discuss time series tasks beyond TSF?
>
> As outlined above, the primary goal of this paper is to diagnose existing TSF Transformers from an in-context perspective, identify their limitations, and demonstrate how ICTSP addresses these issues to improve generalization ability. A key distinction between the forecasting task and other tasks (like classification and imputation) lies in the **autoregressive structure**, where the next step is the prediction answer of the previous step. This creates a unique **shifting structure** between `(lookback, future)`, unlike classification `(lookback, y)`, which is more akin to a traditional supervised learning task, which has been widely studied in prior ICL research [1,2].
>
> In this paper, we focus on how previous Transformer structures handle temporal, aligned series, and unaligned series dependencies, which are tightly tied to the **causal autoregressive nature** of forecasting. For example, ICTSP uses **context examples** to introduce a **shifted structure**, helping the model consider unaligned series-wise relationships, which Series-wise Transformers (like iTransformer) lack. In standard supervised learning settings of general ICL, features are treated independently before training, but in time series, **tokens inherently exhibit shifted autocorrelation**, making shifted structure of context examples essential. This necessity is supported by our experiments (Figure 4).
>
> Thus, the key contributions of this paper is to **highlight these unique considerations for using Transformers in TSF tasks** to the TSF community. Therefore, given space constraints, the paper focuses its core content on **ICL for TSF**, as summarized above.
>
> ```
> References 3
>
> [1] Müller, Samuel, et al. "Transformers Can Do Bayesian Inference." International Conference on Learning Representations.
> [2] Hollmann, Noah, et al. "TabPFN: A Transformer That Solves Small Tabular Classification Problems in a Second." The Eleventh International Conference on Learning Representations.
> ```
>
> We respectfully disagree with the reviewer’s comment that our results are not strong enough. Our work activates the in-context learning abilities of Transformers **without relying on LLM pretrained weights**, using a model with fewer than **10M parameters**, compared to the **7B parameters** in LLaMA used by Time-LLM. Despite the drastically reduced computational cost, our model achieves bettern performance, which we believe demonstrates **stronger results**. Below, we summarize the results for full-data, few-shot (10%, 5%), and zero-shot experiments:
>
> | Methods | ICTSP | Time-LLM | GPT4TS | PatchTST | DLinear |
> |-|-|-|-|-|-|
> | AvgRank (Full) | **1.36** | 1.93 | 4.29 | 2.79 | 5.00 |
> | #Rank1 (Full) | **18** | 8 | 0 | 2 | 0 |
> | AvgRank (10%) | **1.57** | 1.89 | 3.21 | 4.07 | 4.93 |
> | #Rank1 (10%) | **15** | 10 | 1 | 1 | 1 |
> | AvgRank (5%) | **1.56** | 2.36 | 3.48 | 3.96 | 6.04 |
> | #Rank1 (5%) | **12** | 7 | 1 | 2 | 0 |
> | AvgRank (0%) | **1.13** | 1.88 | 4.54 | 3.29 | 5.33 |
> | #Rank1 (0%) | **21** | 3 | 0 | 0 | 0 |

---

> ### Author Response · Authors · 2024-11-20
>
> ```
> 3. The significant improvements in zero-shot tasks over full-data tasks lack detailed explanations. The paper does not sufficiently clarify why the ICTSP demonstrates a pronounced advantage in few-shot and zero-shot learning scenarios than in full-data tasks
> ```
>
> Thank you for recognizing the significant improvements ICTSP achieves in few-shot and zero-shot scenarios. These results are critical to our paper, showcasing ICTSP’s key advantage over previous structures. By introducing forecasting tasks as context tokens, ICTSP activates in-context learning abilities that dynamically selects the most suitable in-context predictor \(f^*\) based on the provided examples.
>
> Unlike **Temporal-wise Transformers**, ICTSP does not use a fixed channel structure in token representation. Instead, it dynamically infers series relationships from input tokens, allowing it to adapt to new series structure when transferred to new dataset. This contrasts with Temporal-wise Transformers, which often encode series relationships from training data as fixed latent structures in the token representation. Compared to channel-independent Temporal-wise Transformers like Time-LLM or PatchTST that do not consider cross-channel information, ICTSP excels at dynamically inferring series relationships.
>
> Without the requirement of additional training, ICTSP adapts to multivariate TSF data with varying distributions through context forecasting examples. This adaptability is a unique capability, enabling ICTSP’s superior performance in **few-shot scenarios (handling temporal distribution shifts)** and **zero-shot scenarios (handling different channel structures)**.
>
> ### Appreciation
>
> Thank you once again for your highly valuable feedback on our paper. Your suggestions have been instrumental in improving the presentation of our work. If you have any further concerns, we would be more than happy to address them.

---

> ### Comment · Reviewer_hkWk · 2024-11-24
>
> Thank you for your thoughtful response. The provided link is accessible, though I encountered multiple access failures on November 2-3, 2024. To be clear, I fully believe that no changes were made to the code during the submission period. I raised this issue solely to bring it to your attention.
>
> However, the reasons I find the novelty of this paper insufficient are as follows:
>
> 1. I fully understand and acknowledge that a well-trained Transformer has ICL capabilities. However, as stated in line 212, the authors’ motivation for proposing ICTSP is "to fully exploit the Transformer's ICL capabilities." We must recognize that utilizing such capabilities assumes the model is already well-trained and inherently possesses them (line 24-27). Only then can these capabilities be further exploited. Since ICTSP is trained from scratch (as indicated in line 352), how can you claim to be leveraging the Transformer’s ICL capabilities? Moreover, the analysis in Section 2.2, which uses an ICL perspective to evaluate other methods, also seems to misrepresent causality. For models trained from scratch, what they learn aligns with their training paradigm. Applying an ICL perspective to interpret these training outcomes is unreasonable. This further highlights what I consider to be a flawed motivation.
>
> 2. While I acknowledge the substantial theoretical support for ICL in prior research, this is not directly relevant to the contributions of this paper. The value of ICL does not automatically validate all work claiming to utilize it.
>
> The most important thing is the authors need to demonstrate their utilization of LIC capabilities from the perspective of the model itself rather than relying solely on experimental results.
>
> Moreover, there an another question: According to Equation 2, whether the final output sequence length should match the label length in the example.

---

> > ### Author Response · Authors · 2024-11-25
> >
> > We sincerely appreciate your additional questions and are happy to provide answers to address your concerns.
> >
> > ```
> > 1. I fully understand and ...
> > 2. While I acknowledge ...
> > ```
> >
> > Sorry for making you confuse. There seems to be some **unclear gaps** regarding the formulation of the ICL problem, and we apologize for not making it clearer. While ICL was originally discovered and discussed **in the context of trained large LLMs**, many recent studies focus on **investigating the ICL capabilities of Transformers in specific problem settings, function classes, or dataset types**. This is reflected in the **ICL formulation** mentioned in line 106 of our paper:
> >
> > - ICL involves each datapoint $(\mathbf{x}\_i, \mathbf{y}\_i) \in \mathbb{R}^a \times \mathbb{R}^b$ from a dataset $\{( \mathbf{x}\_i, \mathbf{y}\_i) \} \_{i=1}^N$. Unlike traditional supervised learning that learns direct mappings from $\mathbf{x}\_i$ to $\mathbf{y}\_i$, ICL predicts $\mathbf{y}\_i$ using both historical observations $\{(\mathbf{x}\_j, \mathbf{y}_j)\}\_{j < i}$ and the current input $\mathbf{x}\_i$.
> >
> > Our work is a follow-up to a series of studies that use this formulation. The basic setup of these studies involves **training a Transformer from scratch on datasets corresponding to a specific problem subclass and then evaluating its inference performance on test data with covariate shifts**. By treating the ground truth example pairs $(\mathbf{x}_i, \mathbf{y}_i)$ from the test data as context example tokens, these studies observe how well Transformers can **handle covariate shifts through context prompts**.
> >
> > In the references cited in our paper and below, [1,2] train Transformers **from scratch on OLS data**, [3,4] train Transformers **from scratch on dynamical system data**, [5,6] train Transformers **from scratch on linear classification data**, [7] train Transformer **from scratch on tabular classification data**, and [8] train Transformer **from scratch on univariate time series data**, among others. These literatures were a major source of **motivation** for us to develop an **in-context multivariate time series Transformer** that does not rely on pretrained LLM weights.
> >
> > - Motivation
> >
> > Since you mentioned motivation, we would like to provide more details about the motivation behind our model design.
> >
> > After reviewing the background literature, we aimed to design an **ICL-based Transformer** capable of effectively **handling covariate shifts in multivariate time series data**. To this end, we first **revisited previous** Transformer-based multivariate TSF models.
> >
> > We found that recent studies indicate that vanilla Transformers using timestep tokenization perform poorly on real-world datasets, showing severe overfitting. However, after applying patching tokenization or series tokenization, their performance becomes significantly more robust. The **TSF community currently lacks explanations** for the mechanisms behind these methods.
> >
> > Given our goal of constructing a multivariate TSF model that effectively leverages series-wise dependencies, we conducted experiments with these structures on multivariate TSF datasets with strong (unaligned) series dependencies (e.g., `Multi`). Surprisingly, we discovered that a vanilla Transformer with timestep tokenization significantly outperformed other tokenization methods, prompting **further investigation**.
> >
> > We attempted to interpret this phenomenon through the perspective of ICL and found that **this framework perfectly explains** the underlying mechanism behind these experimental results. A vanilla Transformer with timestep tokenization uses (historical timesteps, multi-channel values) as context examples within the ICL framework. This approach causes the model to perceive its task as inferring future dynamics based on multi-channel dynamics from the context input. However, when these multi-channel effects do not consistently exist in real-world datasets, this setup leads to significant overfitting. On the other hand, when such multi-channel effects consistently exist, the vanilla Transformer outperforms other further modified structures.

---

> > > ### Author Response · Authors · 2024-11-25
> > >
> > > Initially, we attempted to build an in-context multivariate TSF Transformer using timestep tokenization. However, we found that this approach not only overemphasized multi-channel effects in the input but also **lacked flexibility**. Its token representation construction relies on a linear input projection of multi-channel values. When generalizing the trained model to other datasets, this structure causes token representations to change if the positions of two series are swapped. Moreover, it struggles to adapt to datasets with varying numbers of series. The overlap of different series in the latent space leads to differences of scale in token vectors, making it difficult to distinguish series information when number of series is large. This fixed channel structure is suitable for univariate ICL TSF Transformers (like ForecastPFN [8]) but cannot be effectively extended to multivariate tasks.
> > >
> > > We quickly identified series tokenization, as used in iTransformer, as a good baseline. Its channel structure is highly flexible, allowing it to adapt to any number of series. Unlike timestep or patch tokenization, iTransformer embeds temporal relationships entirely within the token’s latent space and leave the modeling of series relationships to the attention mechanism. This approach aligns with our goal of using **ICL** in multivariate TSF to **build algorithms that manage relationships between multiple series**. However, iTransformer still lacks explicit representation of the (x, y) pairs that serve as context examples in ICL, and performs poorly in modeling unaligned series relationships.
> > >
> > > To address this, we explicitly added (x, y) pairs to iTransformer, extending series tokenization into forecasting task tokenization. We analyzed that iTransformer’s inability to effectively handle unaligned series relationships stems from the absence of context examples with shifted structures, which teach the model how to process such relationships. This **need for shifted structures is unique to multivariate time series** because the different features of different datapoints exhibit **shifting relationships**, which are **not common in other forms of supervised learning problems**.
> > >
> > > As a result, we designed **ICTSP with overlapping forecasting task tokens featuring shifted structures**, rather than non-overlapping tokenization. This design gives ICTSP the flexibility to **adapt to varying numbers of series, different types of series relationships, and new series structures in different datasets**. By leveraging context examples, ICTSP **avoids the limitations of previous Transformer-based TSF models and effectively adapts to diverse multivariate tasks.**
> > >
> > >
> > > - Highlight
> > >
> > > We apologize, as we understand that you may not prefer to read some lengthy explanation. However, we hope that the motivation outlined above helps convey why the **structure of this paper is designed** as this. The **primary focus** of this paper is to propose an **effective structure for training in-context TSF Transformers** by **diagnosing limitations in previous multivariate TSF Transformers**.
> > >
> > > Our intent in structuring the paper this way is to **maximize its impact for the TSF community**. We hope that our **analysis of previous models** and the **effective in-context TSF formulation** can serve as a valuable resource for future researchers, providing them with **clearer directions for training larger TSF Transformers that can better capture multivariate relationships**.
> > >
> > > We hope you understand that **our aim was to provide more insights for the TSF community**, which is why the paper presents itself somewhat like a review and diagnostic of existing models. Our goal was to use ICL as a tool to **maximize its impact on the TSF community**, **rather than to prioritize its contribution to the ICL community**.
> > >
> > > We noticed that the **other three reviewers** were able to identify that **our writing logic was intended to better assist the TSF community and considered this a strength**. We respectfully hope that you can also understand our purpose in this context. Thank you very much for your understanding.

---

> > > > ### Author Response · Authors · 2024-11-25
> > > >
> > > > ```
> > > > References 1
> > > >
> > > > [1] Garg, Shivam, et al. "What can transformers learn in-context? a case study of simple function classes." Advances in Neural Information Processing Systems 35 (2022): 30583-30598.
> > > > [2] Bai, Yu, et al. "Transformers as statisticians: Provable in-context learning with in-context algorithm selection." Advances in neural information processing systems 36 (2024).
> > > > [3] Li, Yingcong, et al. "Transformers as algorithms: Generalization and stability in in-context learning." International Conference on Machine Learning. PMLR, 2023.
> > > > [4] Akyürek, Ekin, et al. "​​ What learning algorithm is in-context learning? Investigations with linear models." The Eleventh International Conference on Learning Representations. 2022.
> > > > [5] Müller, Samuel, et al. "Transformers can do bayesian inference." arXiv preprint arXiv:2112.10510 (2021).
> > > > [6] Hollmann, Noah, et al. "Tabpfn: A transformer that solves small tabular classification problems in a second." arXiv preprint arXiv:2207.01848 (2022).
> > > > [7] Dooley, Samuel, et al. "Forecastpfn: Synthetically-trained zero-shot forecasting." Advances in Neural Information Processing Systems 36 (2024).
> > > > ```
> > > >
> > > > ```
> > > > The most important thing is the authors need to demonstrate their utilization of LIC capabilities from the perspective of the model itself rather than relying solely on experimental results.
> > > > ```
> > > >
> > > > We are more than happy to further explain how our model specifically leverages ICL capabilities to assist with forecasting under covariate shift.
> > > >
> > > >
> > > > ### Ablation Study: Adaptive Model Reduction and Effects of Context Examples
> > > >
> > > > (If you’re short on time, feel free to skip this section and move on to the next ablation study section.)
> > > >
> > > > We designed the following progressive experiments (`A. - G.`) to compare the performance differences resulting from using only a subset of the model structure. These experiments follow the same problem settings as the main experiments in the original paper with $L_I=1440, L_b=512, L_P \in \{96,192,336,720\}$.
> > > >
> > > > To further investigate the impact of distant context tokens to distinguish it from the effects of the consistent temporal relationship between nearer context and target, we added experiments `H. - J.` to demonstrate that even context examples far from the target can effectively improve performance. Here, we consider a 4096-timestep historical window before each forecasting timestep. We sample context examples from the first 25% of this historical interval, meaning the starting points of our context forecasting examples range from step 1 to 1024. Since the forecasting task length $L_b + L_P$ ranges from $512 + 96 = 608$ to $512 + 720 = 1232$, the context forecasting examples end between steps 609 and 2256.
> > > >
> > > > Because the target token requires a lookback window of 512 timesteps, any context portion entirely unrelated to the target token lies before step 3584. In this setting:
> > > > - For $L_P = 720$, the region from step 2256 to step 3584, a span of 1328 steps, is completely skipped and does not contribute to forecasting.
> > > > ...
> > > > - For $L_P = 96$, the region from step 609 to step 3584, a span of 2975 steps, is completely skipped and does not contribute to forecasting.
> > > >
> > > > This pattern goes accordingly for other values of $L_P$.
> > > >
> > > > Since ICL can be viewed as dynamic supervised learning performed during inference, the above setup can be considered a few-shot context example ICL scenario. It resembles the few-shot training setup described in the original paper, where datapoints from a certain time period are skipped.
> > > >
> > > > We conduct the experiments as follows:
> > > >
> > > > - **Model Variants**
> > > >
> > > > A. We remove all Transformer blocks from ICTSP. In this case, the model only includes the input and output linear layers for the target token, effectively reducing it to an NLinear model.
> > > >
> > > > B. We retain only the MLP feedforward layers within the Transformer blocks of ICTSP. The model can then be considered a univariate MLP predictor.
> > > >
> > > > C. We retain the Transformer blocks in ICTSP but drop all context tokens. The model then reduces to a Series-wise Transformer (iTransformer).
> > > >
> > > > D. We retain the full ICTSP structure but randomly keep only 25% of the context tokens in the context examples. The model sees randomly incomplete in-context forecasting task samples.
> > > >
> > > > E. We increase the random retention rate of context tokens to 50%.
> > > >
> > > > F. We further increase the random retention rate of context tokens to 75%.
> > > >
> > > > G. We use all context tokens, representing the full ICTSP model.

---

> > > > > ### Author Response · Authors · 2024-11-25
> > > > >
> > > > > H. From the range of 1 to 1024 for context example starting points, we randomly select 10% of the steps as starting points to construct forecasting task examples. This is referred to as using **10% Distant Context Examples**.
> > > > >
> > > > > I. From the range of 1 to 1024 for context example starting points, we randomly select 25% of the steps as starting points to construct forecasting task examples. This is referred to as using **25% Distant Context Examples**.
> > > > >
> > > > > J. From the range of 1 to 1024 for context example starting points, we randomly select 75% of the steps as starting points to construct forecasting task examples. This is referred to as using **75% Distant Context Examples**.
> > > > >
> > > > > | Methods | A. w/o Transformer (Reduced to NLinear) | B. w/o Attention (Reduced to Univariate MLP) | C. w/o Context Tokens (Reduced to Series-wise Transformer) | D. w/ 25% Random Selected Context Tokens | E. w/ 50% Random Selected Context Tokens | F. w/ 75% Random Selected Context Tokens | H. 10% Distant Context Examples | I. 25% Distant Context Examples | J. 75% Distant Context Examples | G. Full ICTSP |
> > > > > |---|:---:|:---:|:---:|:---:|:---:|:---:|:---:|:---:|:---:|:---:|
> > > > > | Weather (96) | 0.168 | 0.148 | 0.145 | 0.146 | 0.140 | 0.140 | 0.146 | 0.145 | 0.142 | 0.139 |
> > > > > | Weather (192) | 0.213 | 0.191 | 0.193 | 0.190 | 0.188 | 0.185 | 0.192 | 0.191 | 0.189 | 0.186 |
> > > > > | Weather (336) | 0.259 | 0.245 | 0.249 | 0.244 | 0.241 | 0.240 | 0.246 | 0.243 | 0.242 | 0.239 |
> > > > > | Weather (720) | 0.325 | 0.319 | 0.314 | 0.311 | 0.312 | 0.309 | 0.314 | 0.311 | 0.310 | 0.306 |
> > > > > | **Weather (Avg)** | **0.241** | **0.226** | **0.225** | **0.223** | **0.220** | **0.219** | **0.225** | **0.223** | **0.221** | **0.218** |
> > > > > | ETTm2 (96) | 0.163 | 0.163 | 0.165 | 0.162 | 0.161 | 0.161 | 0.164 | 0.164 | 0.161 | 0.159 |
> > > > > | ETTm2 (192) | 0.219 | 0.219 | 0.220 | 0.216 | 0.214 | 0.215 | 0.218 | 0.219 | 0.215 | 0.212 |
> > > > > | ETTm2 (336) | 0.274 | 0.272 | 0.271 | 0.267 | 0.268 | 0.268 | 0.270 | 0.270 | 0.268 | 0.268 |
> > > > > | ETTm2 (720) | 0.361 | 0.354 | 0.361 | 0.358 | 0.355 | 0.349 | 0.355 | 0.352 | 0.351 | 0.347 |
> > > > > | **ETTm2 (Avg)** | **0.254** | **0.252** | **0.254** | **0.251** | **0.250** | **0.248** | **0.252** | **0.251** | **0.249** | **0.247** |
> > > > >
> > > > > It can be observed that even when introducing relatively distant context examples, they still provide better performance compared to the Series-wise Transformer, which does not use any context tokens. Additionally, when only a small portion of context examples is introduced, the performance improvement compared to having no context is relatively modest. However, as the number of context examples increases, their contribution to performance also grows. This aligns with previous observations in ICL literature regarding the impact of the number of context examples used.
> > > > >
> > > > > **However, you may still feel that this does not fully explain how context tokens are used in the form of ICL to improve performance. To address this, we conduct the following additional experiments to help clarify this point for you.**
> > > > >
> > > > > ## Ablation Study: Understanding the Specific Contributions of Context Examples
> > > > >
> > > > > We designed the following experiments to demonstrate how **introducing forecasting tasks as prompt tokens** specifically contributes to performance. These experiments aim to validate that ICTSP relies on context examples to work **via ICL**. Particularly in the zero-shot setting, where **no related training samples' information is in model parameters**, ICTSP’s superior performance is primarily **driven by its ability to update the forecasting predictor using context examples**. Different types of context forecasting examples provide **specific improvements** to the corresponding forecasting task targets.
> > > > >
> > > > > We use ETTm2 as the target dataset and train ICTSP under three different training set configurations: ETTm2, ETTm1, and a combined set of (ETTh1, ETTh2, ETTm1). For these configurations, we conduct the corresponding full data training and zero-shot transfer learning. **During evaluation, we independently generate context examples from series $X_1, \dots, X_7$ in ETTm2 and feed them to the context part of ICTSP for test set MSE evaluation**. We use $L_P=96$, with all other settings matching those in the main experiments.
> > > > >
> > > > > We construct the following variants:
> > > > >
> > > > > 1. **w/o Context**: No context examples are used from any series.
> > > > > 2. **Context ($X_j$)**: Only the context examples corresponding to series $X_j$ are used.
> > > > > 3. **Context (Full)**: The full ICTSP using all context examples.
> > > > >
> > > > > For each variant, we report the MSE ($X_j$) corresponding to each series $j$ in the test set forecasting results. Below are the results for the three experimental settings. For each MSE ($X_j$), the **best value** is **bolded**, and the *second best* is *italicized* for the **w/o Context** and **Context ($X_j$)** parts of experiments.

---

> > > > > > ### Author Response · Authors · 2024-11-25
> > > > > >
> > > > > > ### ETTm2 -> ETTm2 (full data)
> > > > > >
> > > > > > From the results below, we can observe that using the context examples from a specific series $X_j$ improves the model's performance on the corresponding MSE ($X_j$). However, **we cannot fully attribute this improvement to the benefits of ICL. In the full data setting**, the model's parameters already contain information about the relationships between series within the dataset. As a result, the **model may still leverage the information from context examples through these fixed relationships encoded in the weights**. We can see that introducing $X_j$ context not only **improves the performance on $X_j$** but also significantly enhances the performance of **other series that are highly correlated with $X_j$**. Therefore, we **cannot really distinguish** whether the improvement arises from fixed series relationships in the parameters or from ICL.
> > > > > >
> > > > > > | Method | w/o Context | Context ($X_1$) | Context ($X_2$) | Context ($X_3$) | Context ($X_4$) | Context ($X_5$) | Context ($X_6$) | Context ($X_7$) | \| | Context (Full) | Time-LLM | GPT4TS | PatchTST | TimesNet | DLinear |
> > > > > > |---|:---:|:---:|:---:|:---:|:---:|:---:|:---:|:---:|:---:|:---:|:---:|:---:|:---:|:---:|:---:|
> > > > > > | MSE ($X_1$) | 0.281 | **0.268** | 0.286 | **0.268** | *0.277* | 0.278 | 0.294 | 0.283 | \| | 0.267 | - | - | - | - | - |
> > > > > > | MSE ($X_2$) | 0.198 | *0.194* | **0.192** | 0.195 | 0.194 | 0.209 | 0.195 | 0.196 | \| | 0.192 | - | - | - | - | - |
> > > > > > | MSE ($X_3$) | **0.095** | **0.095** | 0.099 | **0.095** | 0.096 | 0.107 | 0.098 | **0.095** | \| | 0.092 | - | - | - | - | - |
> > > > > > | MSE ($X_4$) | 0.394 | 0.389 | 0.400 | *0.385* | **0.383** | 0.407 | 0.388 | **0.383** | \| | 0.378 | - | - | - | - | - |
> > > > > > | MSE ($X_5$) | 0.111 | 0.109 | 0.109 | 0.116 | *0.106* | **0.105** | 0.111 | 0.124 | \| | 0.107 | - | - | - | - | - |
> > > > > > | MSE ($X_6$) | *0.008* | 0.009 | 0.009 | 0.010 | **0.007** | 0.011 | **0.007** | 0.012 | \| | 0.008 | - | - | - | - | - |
> > > > > > | MSE ($X_7$) | 0.078 | 0.072 | 0.080 | *0.070* | 0.079 | 0.093 | 0.083 | **0.068** | \| | 0.074 | - | - | - | - | - |
> > > > > > | MSE (All) | 0.166 | 0.162 | 0.168 | 0.163 | 0.163 | 0.173 | 0.168 | 0.166 | \| | 0.160 | 0.161 | 0.173 | 0.165 | 0.187 | 0.167 |
> > > > > >
> > > > > > ### ETTm1 -> ETTm2 (zero-shot)
> > > > > >
> > > > > > We conducted **zero-shot experiments** where ICTSP was trained on ETTm1 and evaluated on ETTm2. In this scenario, the **model parameters contain no fixed effects from the target dataset**. **Without using context examples**, the model relies **solely on its fixed weights learned from the source dataset** to perform temporal forecasting inference for each target token, resulting in a **significantly worse MSE** of 0.208 compared to other results.
> > > > > >
> > > > > > We can observe the performance imporvements when **introducing context tokens from $X_j$** corresponding to series $j$. Since the model parameters **lack prior knowledge of the channel structure of the target dataset**, the **improvement is most pronounced for the corresponding $X_j$ and diminish for other series**. This **task-specific improvement** originates from ICL, where the model uses a task's question-answer prompt to enhance performance on the specific task.
> > > > > >
> > > > > > Through these experiments, we effectively **disentangled the influence of learned channel structure** from the **effectiveness of ICL**. This demonstrates the **true reason** behind ICTSP's strong performance on zero-shot tasks.
> > > > > >
> > > > > > | Method | w/o Context | Context ($X_1$) | Context ($X_2$) | Context ($X_3$) | Context ($X_4$) | Context ($X_5$) | Context ($X_6$) | Context ($X_7$) | \| | Context (Full) | Time-LLM | GPT4TS | PatchTST | TimesNet | DLinear |
> > > > > > |---|:---:|:---:|:---:|:---:|:---:|:---:|:---:|:---:|:---:|:---:|:---:|:---:|:---:|:---:|:---:|
> > > > > > | MSE ($X_1$) | 0.320 | **0.279** | 0.287 | 0.284 | 0.282 | 0.287 | *0.280* | 0.290 | \| | 0.284 | - | - | - | - | - |
> > > > > > | MSE ($X_2$) | 0.226 | *0.204* | **0.203** | 0.207 | *0.204* | *0.204* | 0.205 | 0.213 | \| | 0.198 | - | - | - | - | - |
> > > > > > | MSE ($X_3$) | 0.141 | **0.100** | 0.105 | **0.100** | 0.102 | 0.103 | *0.101* | 0.111 | \| | 0.098 | - | - | - | - | - |
> > > > > > | MSE ($X_4$) | 0.440 | 0.436 | 0.435 | 0.446 | **0.429** | 0.437 | *0.432* | *0.432* | \| | 0.391 | - | - | - | - | - |
> > > > > > | MSE ($X_5$) | 0.144 | *0.109* | 0.112 | 0.111 | **0.108** | **0.108** | *0.109* | 0.129 | \| | 0.112 | - | - | - | - | - |
> > > > > > | MSE ($X_6$) | 0.048 | *0.008* | 0.009 | *0.008* | 0.009 | *0.008* | **0.007** | 0.025 | \| | 0.008 | - | - | - | - | - |
> > > > > > | MSE ($X_7$) | 0.140 | *0.076* | 0.081 | 0.085 | 0.080 | 0.079 | 0.079 | **0.074** | \| | 0.083 | - | - | - | - | - |
> > > > > > | MSE (All) | 0.208 | 0.173 | 0.176 | 0.177 | 0.174 | 0.175 | 0.173 | 0.182 | \| | 0.168 | 0.169 | 0.217 | 0.195 | 0.222 | 0.221 |

---

> > > > > > > ### Author Response · Authors · 2024-11-25
> > > > > > >
> > > > > > > ### ETTh1,ETTh2,ETTm1 -> ETTm2 (zero-shot)
> > > > > > >
> > > > > > > We further trained ICTSP on a combined dataset (ETTh1, ETTh2, ETTm1) containing more datapoints. Compared to the previous experiment, we observed two key improvements:
> > > > > > >
> > > > > > > 1. The model's **ability to use different context prompt tokens** to achieve corresponding performance improvements was **further enhanced**, resulting in **better overall performance** than in the previous experiment.
> > > > > > > 2. The **task-specific performance improvement effect** became more pronounced for the **series corresponding to the provided context**.
> > > > > > >
> > > > > > > These results highlight the dominant role of ICTSP's ICL ability in achieving strong performance on zero-shot tasks.
> > > > > > >
> > > > > > > | Method | w/o Context | Context ($X_1$) | Context ($X_2$) | Context ($X_3$) | Context ($X_4$) | Context ($X_5$) | Context ($X_6$) | Context ($X_7$) | \| | Context (Full) |
> > > > > > > |---|:---:|:---:|:---:|:---:|:---:|:---:|:---:|:---:|:---:|:---:|
> > > > > > > | MSE ($X_1$) | 0.317 | **0.279** | 0.286 | *0.284* | 0.287 | 0.290 | 0.287 | 0.285 | \| | 0.278 |
> > > > > > > | MSE ($X_2$) | 0.225 | 0.207 | **0.196** | 0.206 | 0.202 | 0.204 | *0.198* | *0.198* | \| | 0.200 |
> > > > > > > | MSE ($X_3$) | 0.116 | 0.101 | 0.107 | **0.099** | 0.105 | *0.100* | 0.101 | 0.102 | \| | 0.098 |
> > > > > > > | MSE ($X_4$) | 0.442 | 0.432 | *0.413* | 0.439 | **0.410** | 0.431 | 0.418 | 0.414 | \| | 0.392 |
> > > > > > > | MSE ($X_5$) | 0.126 | 0.110 | 0.112 | 0.109 | 0.110 | **0.107** | 0.112 | *0.108* | \| | 0.107 |
> > > > > > > | MSE ($X_6$) | 0.013 | *0.008* | 0.013 | *0.008* | 0.010 | *0.008* | **0.007** | *0.008* | \| | 0.009 |
> > > > > > > | MSE ($X_7$) | 0.085 | 0.083 | 0.088 | *0.079* | 0.083 | *0.079* | 0.084 | **0.073** | \| | 0.082 |
> > > > > > > | MSE (All) | 0.189 | 0.174 | 0.174 | 0.175 | 0.172 | 0.174 | 0.173 | 0.170 | \| | 0.167 |
> > > > > > >
> > > > > > > The above experiments also address your initial question: why does ICTSP exhibit such outstanding performance on zero-shot tasks? In fact, the more a task relies on the model's ICL capabilities to handle covariate shifts, the more ICTSP outperforms previous models. As a result, ICTSP's improvement in zero-shot settings is significantly greater than its improvement on full-data tasks.
> > > > > > >
> > > > > > > ```
> > > > > > > Moreover, there an another question: According to Equation 2, whether the final output sequence length should match the label length in the example.
> > > > > > > ```
> > > > > > > Yes, our ICL formulation is based on [1-4]. The number of target tokens corresponds to the number of output series.
> > > > > > >
> > > > > > > To ensure we fully understand your question, we provide more clarification related to the length in the model formulation. The number of target tokens matches the number of output series. As for context tokens, there is no strict limitation on their quantity. In our default settings, context tokens are generated through overlapped slicing on each series, as described in lines 228-259.
> > > > > > >
> > > > > > > If your question pertains to the length of the forecasting task within a token, each token consists of a `(lookback, future)` pair with a total length of $L_b + L_P$. The output projection transforms the final target token length into $L_P$. As we previously mentioned, ICTSP can handle inputs with any number of series. Moreover, when we initialize both the default $L_b$ and $L_P$ with sufficiently large values, the model can adapt to varying lengths of forecasting tasks.
> > > > > > >
> > > > > > > During training, we can randomly truncate the lookback and future portions, modifying them to `(lookback[-l_b:], future[:l_P])`. This allows the model to train on a new forecasting length setting $f: \mathbb{R}^{C \times l_b} \rightarrow \mathbb{R}^{C \times l_P}$. With this flexible training method, ICTSP can handle any input-output configurations within the bounds of $L_b$ and $L_P$, making it feasible to train larger and more versatile multivariate TSF models. If you’re interested, you can refer to [https://anonymous.4open.science/r/ICTSP-MultiTask-53B3](https://anonymous.4open.science/r/ICTSP-MultiTask-53B3) under `data_provider/ictsp_tokenizer.py`. It includes the implementation details of this flexible training method and its extensions to additional tasks.
> > > > > > >
> > > > > > > ```
> > > > > > > References 2
> > > > > > > [1] Bai, Yu, et al. "Transformers as statisticians: Provable in-context learning with in-context algorithm selection." Advances in neural information processing systems 36 (2024).
> > > > > > > [2] Zhang, Ruiqi, Spencer Frei, and Peter L. Bartlett. "Trained transformers learn linear models in-context." arXiv preprint arXiv:2306.09927 (2023).
> > > > > > > [3] Hollmann, Noah, et al. "Tabpfn: A transformer that solves small tabular classification problems in a second." arXiv preprint arXiv:2207.01848 (2022).
> > > > > > > [4] Guo, Tianyu, et al. "How do transformers learn in-context beyond simple functions? a case study on learning with representations." arXiv preprint arXiv:2310.10616 (2023).
> > > > > > > ```

---

> > > > > > > > ### Author Response · Authors · 2024-11-25
> > > > > > > >
> > > > > > > > ### Summary
> > > > > > > >
> > > > > > > > We sincerely thank you for raising additional questions, and we are more than happy to address them. While we hope you understand the highlights we have provided regarding the intended impact of the paper, we also deeply respect your perspective. If the explanations above have addressed your concerns, we wonder if you might consider slightly lowering the confidence level of your reservations or potentially adjusting your score.
> > > > > > > >
> > > > > > > > Once again, we truly appreciate the time and effort you have dedicated to reviewing our paper. We are always open to providing further clarification or answering any additional questions you may have. Thank you very much.

---

> > > ### Author Response · Authors · 2024-11-26
> > >
> > > ### Additional Response: On ICL Abilities and Large-Scale Parameters in LLMs
> > >
> > > Dear Reviewer hkWk,
> > >
> > > We recently responded to Reviewer NX33 regarding concerns about whether using `#Layers=3` to train ICTSP makes the model size too small for ICL. This prompted me to wonder if a potential gap in background knowledge regarding recent developments in ICL might have led to your additional questions above about ICL abilities. We would like to provide further clarification.
> > >
> > > You may believe that ICL abilities is kind of an emergence solely as a result of training LLMs on large-scale data, which was perhaps the initial impression when the concept of ICL was introduced. However, many recent ICL studies do not approach the problem from this perspective. These studies follow a pattern of proofs framed as "Transformer can do xxx," demonstrating that the ICL formulation (line 106) provides a method to achieve ICL for Transformer. They mathematically validate that the core algorithms performed during ICL are achieved through the softmax self-attention layers within Transformers.
> > >
> > > ICL is not an emergent phenomenon dependent on large-scale pretraining but a capability explicitly proven to exist within Transformers and activatable under the right formulation. Moreover, **ICL abilities are independent of large-scale parameters and can exist in very small Transformers. In fact, for certain ICL behaviors that approximate linear model functionality, Transformers may even require restrictions of number of layers to perform effectively, descibed as follows.**
> > >
> > > Below is the content from our response to Reviewer NX33 regarding the number of layers:
> > >
> > > [Start]
> > >
> > > The hyperparameter settings (model size) for ICL Transformers are more dependent on the specific data type or problem setting being explored. For example:
> > >
> > > - [1] includes settings such as `#Layers=3, #Heads=2, and #EmbDim=64`, labeled as `Tiny`. The study demonstrates that even with these `Tiny` settings, Transformers can achieve performance comparable to larger models across many task types and even reach optimum error on certain tasks.
> > > - [3] shows that using a `Tiny` setting with `#Layers=3` does not alter the Transformer’s behavior in ICL generalization.
> > > - [4] demonstrates that Transformers with different #Layers can approximate the behavior of different linear models: `single-layer` Transformers closely match a single step of gradient descent, `2-4 layer` Transformers are closer to Ridge Regression, and Transformers with `more layers` approximate OLS.
> > > - [2] explicitly proves that for ICL to perform Ridge Regression, a mild bound of `#Layers ≤ 3` is required. Similarly, for LASSO, `#Layers ≤ 2 and #EmbDim ≤ 2d` are required, where \(d\) is the number of input features in the regression problem.
> > > - During model training, [5] employs `#Layers=3` as the default setting. And in [9], which explore the ICL univariate time series forecasting which is close to our domain, even use a `#Layers=2` setting.
> > >
> > > In summary, previous ICL Transformer literature highlights the importance of selecting an appropriate depth for the specific problem and even suggests that certain ICL behaviors may require an upper bound on depth. Therefore, for multivariate time series data, which demands significant regularization, we believe that using the default setting of `#Layers=3` is a reasonable choice.
> > >
> > > [End]
> > >
> > > Thank you once again for the time and effort you have dedicated to reviewing our paper, and we apologize for taking up your time.
> > >
> > > ```
> > > References
> > >
> > > [1] Garg, Shivam, et al. "What can transformers learn in-context? a case study of simple function classes." Advances in Neural Information Processing Systems 35 (2022): 30583-30598.
> > > [2] Bai, Yu, et al. "Transformers as statisticians: Provable in-context learning with in-context algorithm selection." Advances in neural information processing systems 36 (2024).
> > > [3] Li, Yingcong, et al. "Transformers as algorithms: Generalization and stability in in-context learning." International Conference on Machine Learning. PMLR, 2023.
> > > [4] Akyürek, Ekin, et al. "​​ What learning algorithm is in-context learning? Investigations with linear models." The Eleventh International Conference on Learning Representations. 2022.
> > > [5] Müller, Samuel, et al. "Transformers can do bayesian inference." arXiv preprint arXiv:2112.10510 (2021).
> > > [6] Hollmann, Noah, et al. "Tabpfn: A transformer that solves small tabular classification problems in a second." arXiv preprint arXiv:2207.01848 (2022).
> > > [7] Dooley, Samuel, et al. "Forecastpfn: Synthetically-trained zero-shot forecasting." Advances in Neural Information Processing Systems 36 (2024).
> > > [8] Edelman, Benjamin L., et al. "Inductive biases and variable creation in self-attention mechanisms." International Conference on Machine Learning. PMLR, 2022.
> > > [9] Dooley, Samuel, et al. "Forecastpfn: Synthetically-trained zero-shot forecasting." Advances in Neural Information Processing Systems 36 (2024).
> > > ```

---

> ### Author Response · Authors · 2024-12-02
>
> Dear Reviewer hkWk,
>
> Thank you very much for raising a series of insightful questions during the review process. These questions have been incredibly helpful in improving the presentation quality of the core content of our work. With only about one day left until the end of the ICLR rebuttal period, we kindly wish to know if the responses we provided have addressed your concerns. If you have any additional questions, we would be more than happy to provide a quick response.
>
> Once again, we deeply appreciate your thoughtful questions, which have greatly enhanced the quality of our presentation. Thank you for the time and effort you have dedicated to reviewing our paper.

---

### Public Comment · ~Jiecheng_Lu1 · 2025-05-26

We apologize for forgetting to replace the temporary anonymous code repository link in the main text with the permanent one. The correct GitHub repository for this paper is: [https://github.com/LJC-FVNR/In-context-Time-Series-Predictor](https://github.com/LJC-FVNR/In-context-Time-Series-Predictor).

---

### Meta-Review · Area_Chair_CVHY · 2024-12-20

**Metareview:**

This paper proposes an in-context time-series predictor, a unified formulation integrating temporal-wise and series-wise Transformers for time-series forecasting. This general formulation encompasses several existing models as special cases, thereby offering insights into their limitations. The experiments demonstrate superior performance across full-data, few-shot, and zero-shot settings.

One limitation of this work, raised by Reviewer H6bM, is that it does not sufficiently isolate the contribution of in-context learning capability and longer lookback window ot its strong performance. AC think this is acceptable, because, unlike in NLP, context samples from distinct time series often differ significantly in distribution, potentially hindering in-context learning. While Reviewer hkWk also holds a similar concern, he/she arguments that a Transformer trained from scratch, rather than pretrained on large-scale data, should not exhibit ICL capability. This stance contradicts a growing consensus in the community, and the authors provide extensive literature support for their position.

Given that in-context learning remains a particularly important but underexplored topic in time-series forecasting, and considering this paper’s systematic literature discussion and solid experimental results, I recommend acceptance.

**Additional Comments On Reviewer Discussion:**

A common concern (Reviewer NX33, H6bM) raises from the setting of input sequence length, which potentially leads to unfair comparison. This is ideally addressed by additional results with limited lookback information. Another more important concern shared by Reviewer hkWk and H6bM is on the in-context learning capability, as explained above. Although hkWk remains firmly unconvinced and advocates for rejection, H6bM provides a slightly positive assessment. Additionally, the authors provides lengthy responses during rebuttal. The authors’ rebuttal was quite lengthy, which made it difficult to extract the most critical information and may have hindered effective discussion with hkWk’s concerns.

---

### Decision · Program_Chairs · 2025-01-22

Accept (Poster)